# PBEBench: A Multi-Step Programming by Examples Reasoning Benchmark inspired by Historical Linguistics

## Abstract

Although many benchmarks evaluate the reasoning abilities of Large Language Models (LLMs) within domains such as mathematics, coding, or data wrangling, few abstract away from domain specifics to examine reasoning as a capability in and of itself. We contribute a novel type of benchmark evaluating the inductive reasoning capabilities of LLMs that is inspired by the forward reconstruction task from historical linguistics but is formulated in an extremely simple, general way (in the form of Programming by Examples or PBE). The task involves generating a cascade of simple string rewrite programs to transform a given list of input strings into a list of desired output strings. We present a fully automated pipeline that programmatically generates problems of this type with controllable difficulty (varying, for example, ground truth cascade lengths), enabling scalable evaluation of reasoning models while avoiding contamination. Using this approach, we construct two benchmarks: *PBEBench-Lite*, which efficiently stratifies models of varying capabilities, and *PBEBench*, which requires models to induce programs more similar (in complexity) to those constructed by historical linguists. Our experiments reveal a substantial performance gap between models that leverage test-time compute or LCoT (long chain-of-thought) reasoning and those that do not. Moreover, although recent models such as gpt-oss-120b and GPT-5 show promise, the solve rate for both of them drops below 5% for hard instances of the PBEBench dataset (ground truth cascade lengths of 20 and 30, respectively), falling well short of realistic historical linguistics requirements even with computationally expensive, popular scaling techniques from the PBE and reasoning literature. Additionally, we also study the effectiveness of different scaling strategies and the impact of various hyperparameters on the difficulty of the generated data using gpt-oss-120b, the best-performing open-source model. We plan to open-source our code and benchmark snapshots to enable reproducibility and future work.

## 1 Introduction

You are trying to refactor a large codebase, implementing changes like replacing all instances of `.foo` with `.bar`, `.bar` with `.baz`, and `.baz` with `.foo`. When you tell your senior colleague, a seasoned Unix hacker, what you are doing, he smiles and says that he can (after inspecting the originals and your examples) write a `sed` (stream editor) script that will update the entire codebase. His task involves not only constructing the appropriate regular expression replacements, but also placing them in the correct sequence. For example `s/ḃar/ḃaz/` has to be ordered before `s/ḟoo/ḃar/` (otherwise, all instances that were originally `.foo` would become `.baz`). At a deep level, this problem is the same as one from historical linguistics: given a set of pairs consisting of a word in an ancestral language and a word in a modern language, define a sequence of local string transformations that convert the ancestral words to the modern words (a task called forward reconstruction). In both cases, one is given a collection of inputs $\vec{i}$ (input files or ancestral words) and a collection of outputs $\vec{o}$ to which the inputs should be mapped by a program $\vec{p}$ consisting of a sequence of incremental transformations. Both are clear instances of programming example, Naik et al. (2025b) (the inspiration for the current work) having made this explicit for forward reconstruction.

An important observation of Naik et al. (2025b) is that this class of tasks is challenging for LLMs. Because of this, and because it can be reduced to the barest formal machinery, this task provides an ideal framework for comparing the ability of language models to construct (by induction) sequential plans. The principle contribution of the paper is a benchmark of this type. It has a number of desirable properties underrepresented in current benchmarks for reasoning: (1) Unlike most reasoning benchmarks (see 2), it is not associated with a specific domain and does not benefit from non-trivial domain knowledge. (2) Its complexity and difficulty emerge from a very simple framework with provable logic. (3) This emergent complexity includes four fundamental challenges (to capture the relationships Kiparsky (1968; 1971) called feeding, bleeding, counter-feeding, and counter-bleeding) which can be manipulated and quantified. (4) The benchmark is immune to data contamination and resistant to saturation because new test instances can be generated, with controlled difficulty, very efficiently. (5) It compares the ability of models to perform a class of practically and scientifically interesting tasks rather than focusing on comparisons between model and human abilities.

Our study addresses the following questions:

**RQ1: Which models are good at inductive reasoning?** We benchmark 13 reasoning and 8 strong non-reasoning models on PBEBench-Lite (a lightweight version of the proposed benchmark) to determine whether long chain of thought (LCoT) reasoning is beneficial in this task.

**RQ2: What makes a PBE instance hard?** We use logistic regression and factorial analysis to determine how much factors like ground truth cascade length, types of relations between rewrites, etc., impact the solution difficulty of dataset problems, based on solution success across all models. We also create a snapshot similar to PBEBench-Lite but with 50 examples per PBE instance (PBEBench-Lite-MoreEg) to evaluate the impact of having more examples on solution success rate.

**RQ3: Could the strongest LLMs solve problems requiring long programs (like real-world forward reconstruction problems)?** We pick the strongest open and closed-source models from the PBEBench-Lite results and analyze them on PBEBench (the full benchmark).

**RQ4: What scaling strategies work best for solution search?** We investigate which of the two solution search scaling strategies discussed in Section 3.2 yields more "bang for the buck" by holding the sampling budget constant and varying the maximum sequence length and vice-versa.

## 2 RELATED WORK

**Programming By Example.** Programming by Example (PBE) (Gulwani, 2010) is a well-known and intuitive paradigm in program synthesis research, where programs are inferred solely from a small set of input-output pairs. Early symbolic approaches relied on domain-specific languages (DSLs) and constraint solving: FlashFill uses a string-transformation DSL to automate spreadsheet tasks (Gulwani, 2011), while Syntax-Guided Synthesis (SyGuS) constrains program search via grammar specifications (Alur et al., 2013). Later, DeepCoder (Balog et al., 2017) guided search using learned function predictions, and RobustFill (Devlin et al., 2017) trained sequence-to-sequence models to directly emit DSL programs. More recently, Large Language Models have shown few-shot capabilities in code generation (Chen et al., 2021a; Guo et al., 2024) and test case generation (Li & Yuan, 2024), but on traditional PBE tasks they often struggle with out-of-distribution examples, and improve only after fine-tuning on the target distribution (Li & Ellis, 2024; Naik et al., 2025b).

**Inducing Context-Sensitive Grammars in LLMs.** Attempts to induce string-rewrite rules from data have a long history (Gildea & Jurafsky, 1995); discussions of the formal properties of these rules in linguistics go back farther, including (pivotal to this study) the discovery of feeding and bleeding relationships between such rules Kiparsky (1968; 1971). More recently, formal language benchmarks have shown that RNNs can outperform transformers on formal language tasks involving certain classes of grammars (Butoi et al., 2025). Morphophonological probes (Borenstein, 2024) reveal that while models fit the training data, they often default to heuristics, highlighting limitations in true rule induction. Recent work analyzing LLM-based hypothesis search further documents systematic errors in rule induction, showing that models often produce oversimplified hypotheses that fail to generalize (Parab et al., 2025). Naik et al. (2024) demonstrate that LLMs can induce sound laws, generalizing the method to full context-sensitive program synthesis (Naik et al., 2025b). However, none of these prior works have proposed provably correct algorithms for detecting feeding and bleeding relations (a unique contribution of this work; see Appendix C).

**Reasoning and Induction Benchmarks.** A range of benchmarks test reasoning and PBE skills. Code-centric suites include HumanEval and MBPP (Chen et al., 2021a; Austin et al., 2021), while PBE-style tasks appear in FlashFill datasets (Gulwani, 2011). Newer benchmarks include TESTGENEVAL (Jain et al., 2025), that targets unit tests generation. CODE2BENCH (Zhang et al., 2025) has been proposed to improve scalability in benchmark generation for software programs, while Case2Code (Shao et al., 2025) introduces a scalable synthetic dataset for program synthesis (requiring models to infer code from input-output examples). However, these approaches are tied to software engineering, where datasets demand both domain knowledge and reasoning, introducing unequal prior exposure across models and preventing contamination-free evaluation of the knowledge-free reasoning ability we aim to study. Linguistic and reasoning benchmarks such as HotpotQA (Yang et al., 2018), DROP (Dua et al., 2019), and GSM8K (Cobbe et al., 2021) stress multi-step and mathematical reasoning. System-2 reasoning is evaluated in BIG-Bench Hard (Suzgun et al., 2022), where chain-of-thought prompting improves performance significantly. Compositional generalization splits like SCAN (Lake & Baroni, 2018) and CFQ (Keysers et al., 2020) reveal extrapolation gaps. Visual PBE in ARC (Chollet, 2019) is a non-language analog to our task. However, our benchmark is significantly more resistant to leakage, easily scalable, and offers a mechanism to modulate task difficulty by simple adjustment of the generation parameters. To our knowledge, no benchmark targets multi-step synthesis of string-rewriting programs; PBEBench fills this niche by unifying PBE and compositional generalization in a single benchmark. The Illusion of Thinking (Shojaee et al., 2025) highlights reasoning collapse and out-of-token issues in complex tasks, with subsequent work (Lawsen, 2025) noting methodological limitations. We encounter similar issues during evaluation, but counter them by scaling chain-of-thought budgets, adding additional sampling when needed, and extending timeouts with parallelized inference for efficiency.

## 3 METHODOLOGY

Our approach consists of two components: (1) a problem proposer and (2) a problem solver. The proposer is a programmatic system that generates PBE instances of controllable difficulty, forming the core of our dynamic benchmarking framework. It scalably and reliably produces novel, contamination-free data of the desired complexity. The solver is any system under evaluation. In this work, we focus on state-of-the-art open and closed-source reasoning LLMs to examine the limits of their inductive reasoning capabilities and to assess whether LLM-based methods can be deployed for realistic historical linguistics sound law induction tasks.

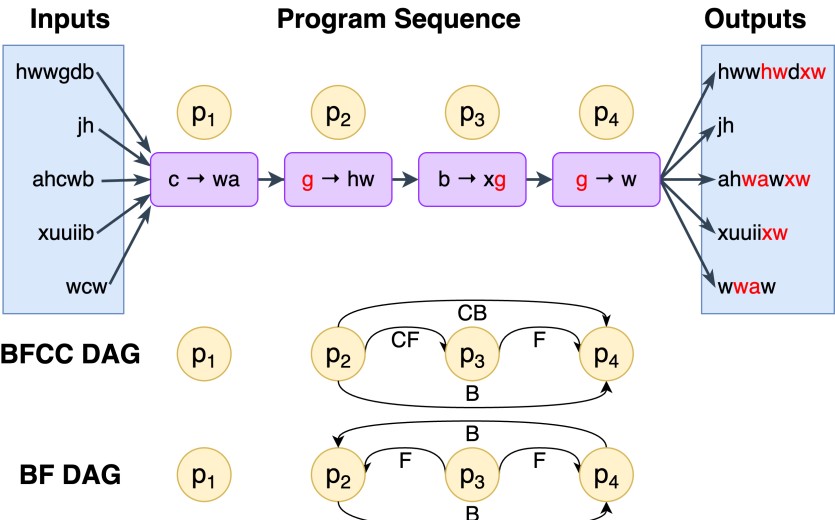

Figure 1: Structure of the $\langle \vec{i}, \vec{p}, \vec{o} \rangle$ triples dataset. The relations between the programs in the sequence are shown by the BFCC DAG, and the simplified BF DAG replaces the counterfactual relations with a reversed simple link.

### 3.1 PROBLEM PROPOSER

We construct our benchmark programmatically by sampling inputs from a distribution of strings $\mathcal{I}$ and a sequence of string rewrite programs from a distribution $\mathcal{P}$. Each string rewrite program acts like a find and replace function and substitutes all occurrences of substring $\alpha$ with a substring $\beta$ (details in Section 3.1.2). We apply the sampled *cascade* of string rewrite programs $\vec{p} = \langle p_1, \ldots, p_m \rangle \in \mathcal{P}$ to the sampled inputs $\vec{i} = \langle i_1, \ldots, i_n \rangle \in \mathcal{I}$, producing outputs $\vec{o} = \{p_m(\ldots p_1(i_1)), \ldots, p_m(\ldots p_1(i_n))\}$. For convenience, we write this transformation as $\vec{o} = \vec{p}(\vec{i})$. Each instance in our dataset is defined by the triplet $\langle \vec{i}, \vec{p}, \vec{o} \rangle$, consisting of the inputs, the program cascade, and the corresponding outputs. Additionally, we record metadata such as the interactions between programs (e.g., BLEEDING, FEEDING, etc.).

Our data generation algorithm is parameterized by: the number of examples $n$ ($|\vec{i}| = |\vec{o}| = n$); the input alphabet $\Sigma$ used to generate both the inputs and the string rewrite programs; the minimum and maximum input string lengths $l_{min}$ and $l_{max}$; the minimum and maximum cascade lengths $L_{min}$ and $L_{max}$ ($L_{min} \leq |\vec{p}| \leq L_{max}$); the minimum and maximum length of strings $s_{min}$ and $s_{max}$ ($s_{min} \leq |\alpha|, |\beta| \leq s_{max}$) used by the string rewrite programs and the desired dataset size $D$. We denote a benchmark snapshot generated with specific parameter values as $\mathcal{D}(n, \Sigma, L_{min}, L_{max}, l_{min}, l_{max}, s_{min}, s_{max}, D)$.

#### 3.1.1 INPUT SAMPLING

Our input sampling procedure is parameterized by $(n, \Sigma, l_{min}, l_{max})$ as defined in Section 3.1. We build the input sequence $\vec{i} = \langle i_1, \ldots, i_n \rangle$ by independently sampling each input $i_k$ for $1 \leq k \leq n$. For each $i_k$, we first sample its length from a uniform discrete distribution, $|i_k| \sim \text{Unif}\{l_{min}, \ldots, l_{max}\}$, where $|i_k|$ denotes the length of the string $i_k$. We then generate the string itself by sampling $|i_k|$ characters independently with replacement from the alphabet $\Sigma$, i.e., $i_k \sim \text{Unif}(\Sigma)^{|i_k|}$.

#### 3.1.2 PROGRAM SAMPLING AND OUTPUT GENERATION

Program sampling is parameterized by $(\Sigma, L_{min}, L_{max}, s_{min}, s_{max})$. We begin by programmatically selecting a cascade length $L$ such that $L_{min} \leq L \leq L_{max}$. Next, we construct a sequence of $L$ string rewrite programs $\vec{p} = \langle p_1, \ldots, p_L \rangle$, where each program is of the form $p_k = \text{replace}(\alpha_k, \beta_k)$. Here, replace has the same semantics as Python's built-in `replace()` method for strings: the substring $\alpha_k$ is replaced by $\beta_k$, with the restriction that $\alpha_k$ is non-empty. To sample each program $p_k$, we generate $\alpha_k$ and $\beta_k$ independently, following a procedure analogous to input sampling:

1. Sample the lengths $|\alpha_k|$ and $|\beta_k|$ from a discrete uniform distribution: $|\alpha_k|, |\beta_k| \sim \text{Unif}\{s_{min}, \ldots, s_{max}\}$
2. Sample $\alpha_k$ uniformly from the set of substrings of length $|\alpha_k|$ in the intermediate input vector $\vec{i}_{k-1} = p_{k-1}(\ldots p_1(\vec{i}))$ (where $\text{Substr}_s(\vec{i}_{k-1})$ denotes the set of all substrings of length $s$ in $\vec{i}_{k-1}$): $\alpha_k \sim \text{Unif}(\text{Substr}_{|\alpha_k|}(\vec{i}_{k-1}))$ (note that $\vec{i}_0 = \vec{i}$).
3. Sample $\beta_k$ as a sequence of $|\beta_k|$ characters, each drawn independently from the uniform character distribution over the alphabet $\Sigma$ (same as in Section 3.1.1): $\beta_k \sim \text{Unif}(\Sigma)^{|\beta_k|}$

The outputs are then generated by executing the program cascade $\vec{p}$ over the inputs $\vec{i}$ as $\vec{o} = \vec{p}(\vec{i})$

#### 3.1.3 ENFORCING COMPLEXITY CONSTRAINTS

We control the complexity of cascades of string rewrite programs $\vec{p}$ using rejection sampling guided by the classifiers developed in Section C. Complexity is balanced along two dimensions: (1) the cascade length $|\vec{p}|$, and (2) the set of relation types present between programs in $\vec{p}$. The second dimension is encoded as a binary category string $c_p \in \{0, 1\}^4$. Each digit corresponds to the binary presence or absence of feeding, bleeding, counter-feeding, and counter-bleeding relations. Instances are balanced across all 16 possible category strings, from $c_p = 0000$ (no relations present, arbitrary ordering possible) to $c_p = 1111$ (all relations present, ordering highly constrained).

**Computing Instance Complexity.** We hypothesize that the difficulty of a PBE problem $\langle \vec{i}, \vec{p}, \vec{o} \rangle$ is governed by (1) cascade length and (2) the types of relations between program pairs in $\vec{p}$ (relations

formulated by phonologists and historical linguists; see Kiparsky (1968; 1971)). One key contribution is a provably correct and automatic classification of these relations (Appendix C). For any ordered pair of programs $(p_i, p_j)$ in $\vec{p}$, the possible relations are:

- **Feeding (F($p_i, p_j$)):** $p_i$ creates substrings that enable $p_j$ to apply.
- **Bleeding (B($p_i, p_j$)):** $p_i$ removes substrings required by $p_j$.
- **Counter-Feeding (CF($p_i, p_j$)):** $p_i$ could have fed $p_j$, but $p_j$ precedes $p_i$.
- **Counter-Bleeding (CB($p_i, p_j$)):** $p_i$ could have bled $p_j$, but $p_j$ precedes $p_i$.
- **No Relation:** $p_i$ and $p_j$ can be ordered arbitrarily.

Counter-relations need not be stored explicitly, since $CF(p_i, p_j)$ and $CB(p_i, p_j)$ are implied if $p_j$ precedes $p_i$ and $F(p_j, p_i)$ and $B(p_j, p_i)$, respectively. These relations can be visualized using a Directed Acyclic Graph (DAG), as illustrated in Figure 1. We show both the full DAG with all relations and a simplified DAG that indirectly encodes counter-relations via this symmetry.

**Rejection Sampling to Control Complexity.** To enforce balanced complexity, we use rejection sampling (Section 3.1.2). Each instance is assigned a category string $c_p$, and data is generated to approximate balance across the $2^4 = 16$ relation-type categories. We also aim to balance cascade lengths between $L_{min}$ and $L_{max}$, though these constraints can conflict. For example, with $L = 2$, randomly sampled program pairs rarely exhibit three or four relation types, making some category strings unattainable. Prior work on sound law induction (Naik et al., 2025a) showed that fine-tuning on randomly sampled programs yielded greater improvements in real-world forward reconstruction than retrieving from attested sound laws, likely due to higher diversity. Following this insight, we retain random sampling (rather than deterministic generation or database retrieval) to promote diversity and avoid contamination. This leads to high rejection rates, often requiring constraint relaxation. We therefore introduce a patience parameter $\tau$, which is the number of sampling steps carried out with both constraints. Once $\tau(= 100000)$ is exceeded, we accept any instance satisfying at least one constraint (either $c_p$ or cascade length). In early experiments, relation-type balance was prioritized, but later analysis (Appendix G.3, G.4) showed cascade length to be a stronger predictor of difficulty. Accordingly, in subsequent experiments, we relaxed relation-type constraints while maintaining strict control over cascade length. We also study the effect of using different values of $\tau$ on the balance of relation types and efficiency of the data generation process (Appendix G.2).

## 3.2 PROBLEM SOLVER

**Prompting Strategy:** Once a benchmark snapshot $\mathcal{D}$ is generated, we evaluate each LLM $M$ by prompting it (see appendix E.2) to produce a candidate program cascade $\hat{\vec{p}} = M(\vec{i}, \vec{o})$ mapping inputs $\vec{i}$ to outputs $\vec{o}$. The prompt enforces the following constraints: each program must be a Python `replace` function; both arguments $\alpha_k$ and $\beta_k$ must be strings with $|\alpha_k|, |\beta_k| \leq s_{max}$; $\alpha_k$ must be non-empty; the cascade may contain at most $L_{max}$ programs (matching the length of the longest ground-truth cascade); and the output must follow a strict Markdown format for reliable extraction. If a predicted program $\hat{p}_j$ has more than $L_{max}$ programs, we consider the first $L_{max}$ programs for evaluation. If the predicted program violates any other constraint, it is replaced by an identity program $p^I$, which leaves inputs unchanged. Given the predicted cascade $\hat{\vec{p}}$, we compute the predicted outputs as $\hat{\vec{o}} = \hat{\vec{p}}(\vec{i})$. Since we evaluate all kinds of LLMs, including reasoning models that may first produce intermediate outputs and then iteratively refine or improve them by reflection, we account for this by evaluating both the first and last code block produced by each model.

**Scaling Solution Search:** We explore two approaches to scaling the solution search, inspired by prior work on LLMs for PBE (Li & Ellis, 2024) and general reasoning (Muennighoff et al., 2025): (1) sampling budget and (2) test-time thinking-budget scaling. The first approach gives the LLM multiple attempts to solve a problem by successively sampling solutions $K$ times and selecting any solution consistent with the input-output examples. If no correct solution is found in the $K$ samples, we pick the one with the highest edit similarity reward (defined in Section 4.3). The second approach is inspired by Muennighoff et al. (2025). We constrain the thinking budget to $N$ tokens and scale it by increasing this limit. However, we observed that gpt-oss-120b does not strictly adhere to thinking budget constraint, even when employing strategies such as forcing it to emit chain-of-thought termination tokens (experimental details are available in Appendix F.3), so we ultimately control the overall maximum sequence length.

## 4 EXPERIMENTS

### 4.1 BENCHMARK CREATION

Using the problem proposer described in Section 3.1, we construct two benchmarks. *PBEBench-Lite* is a simplified dataset with longer cascades and 5 input–output pairs per instance, synthesized as $\mathcal{D}(n = 5, \Sigma = \Sigma_{\text{lite}}, L_{min} = 2, L_{max} = 5, l_{min} = 2, l_{max} = 6, s_{min} = 1, s_{max} = 3, D = 1008)$, with exactly 63 examples per relation category. The alphabet is restricted $\Sigma_{\text{lite}} = \{a, \ldots, k, u, v, w, x, y, z\}$ (lowercase letters excluding $l$–$t$). *PBEBench* is a larger dataset with 50 input-output pairs per instance, synthesized as $\mathcal{D}(n = 50, \Sigma, L_{min} = 2, L_{max} = 20, l_{min} = 2, l_{max} = 6, s_{min} = 1, s_{max} = 3, D = 1216)$, with 64 instances for each cascade length. Here $\Sigma = \{a, \ldots, z, A, \ldots, Z\}$ (the full alphabet in both cases). While PBEBench does not reach the complexity of real-world forward reconstruction, where cascades may exceed 40 programs, it remains challenging for the strongest open-source models. For evaluation of advanced closed-source systems such as GPT-5, we additionally synthesize cases with cascades up to length 30.

### 4.2 MODELS EVALUATED

We evaluate a broad range of state-of-the-art LLMs spanning multiple model families and design choices. Our evaluation covers models from leading developers including OpenAI (GPT, o-series, gpt-oss), Anthropic (Claude series), Google DeepMind (Gemini), Qwen (Qwen2.5, Qwen3, QwQ), DeepSeek (R1-Distill-Qwen), Mistral (Codestral). The models differ in scale, reasoning specialization (thinking vs. non-thinking), architectural choices (dense vs. MoE), and source availability (open vs. closed). A full list of models with their attributes is provided in Table 2, which demonstrates the breadth of model families and capabilities we consider.

### 4.3 EVALUATION METRICS

Since a given list of inputs $(\vec{i})$ could be transformed into the outputs $(\vec{o})$ by multiple program cascades $\vec{p}$ we utilize metrics based on functional equivalence. We execute the model-generated solution $\hat{\vec{p}}$ on the inputs, treating them like test cases. We compare the predicted outputs $\hat{\vec{o}} = \hat{\vec{p}}(\vec{i})$ with the ground truth outputs $\vec{o}$ at two levels of granularity: (1) coarse-grained evaluation (`Pass@1` or solve rate) and (2) fine-grained evaluation (`Edit_Sim` or normalized edit similarity). Both metrics do element-wise comparisons on the strings in the output vectors:

**Coarse-grained Metric** (`Pass@1`): This metric is `Pass@1` from Chen et al. (2021b).

$$\text{pass@1} = \frac{1}{|\mathcal{D}|} \sum_{\vec{o}, \vec{i} \in \mathcal{D}} 1_{\hat{\vec{p}}(\vec{i}) = \vec{o}}$$

Here $1_{\hat{\vec{p}}(\vec{i}) = \vec{o}}$ is an indicator variable which is 1 when $\hat{\vec{p}}(\vec{i}) = \vec{o}$ is true and 0 if it is false.

**Fine-grained Metric** (`Edit_Sim`): This metric is the same as Reward@1 used by Naik et al. (2025b).

$$\text{Edit\_Sim} = \frac{1}{|\mathcal{D}|} \sum_{\vec{o}, \vec{i} \in \mathcal{D}} 1 - \frac{\text{dist}(\hat{\vec{p}}(\vec{i}), \vec{o})}{\text{dist}(\vec{i}, \vec{o})}$$

Here dist denotes the total Levenshtein edit distance summed across the corresponding inputs and outputs. Additionally, we evaluate the proportion of programs an LLM generates that are valid (follow all instructions mentioned in 3.2) which we term as the `Valid_Rate`.

## 5 RESULTS

### 5.1 PBEBENCH-LITE PERFORMANCE

Table 1 reports `Pass@1`, `Edit_Sim`, and `Valid_Rate` for all models (Section 4.2) on both the first and last code block from each response (as explained in Section 3.2). Since the last block usually performs as well or better, we use it for instance difficulty analysis. Hyperparameters used for model evaluations are listed in Table 6. Claude-4 Opus (Thinking) was run on only 20% of data due to

Table 1: **PBEBench-Lite Performance:** We compute the `Pass@1` and `Edit_Sim` as the coarse and fine-grained evaluation, respectively, for each model. ■ indicates mixture-of-experts (or MoE) model. ★ indicates a reasoning model. * indicates evaluated on 20% of the dataset due to cost.

| Model | First Code Block | | | Last Code Block | | |
|---|---|---|---|---|---|---|
| | Pass@1 | Edit Sim | Valid Rate | Pass@1 | Edit Sim | Valid Rate |
| Codestral-22B | 0.0109 | -0.0130 | 0.8280 | 0.0109 | -0.0107 | 0.8254 |
| Qwen2.5-32B-Instruct | 0.0298 | 0.1232 | 0.8265 | 0.0298 | 0.1252 | 0.8288 |
| Qwen2.5Coder-32B-Instruct | 0.0397 | 0.1836 | 0.6883 | 0.0407 | 0.1884 | 0.6890 |
| Qwen3-32B | 0.0179 | 0.0964 | 0.7645 | 0.0179 | 0.0964 | 0.7645 |
| Qwen3-Coder-30B-A3B-Instruct ■ | 0.0377 | 0.0857 | 0.8048 | 0.0377 | 0.0864 | 0.8134 |
| DeepSeek-R1-Distill-Qwen-32B ★ | 0.2242 | 0.3486 | 0.8709 | 0.2242 | 0.3486 | 0.8709 |
| Qwen3-30B-A3B ★■ | 0.2887 | 0.3360 | **0.9905** | 0.2887 | 0.3360 | **0.9905** |
| QwQ-32B ★ | 0.3591 | 0.4072 | 0.9500 | 0.3601 | 0.4092 | 0.9493 |
| Qwen3-32B (with CoT) ★ | 0.3938 | 0.4803 | 0.9182 | 0.4187 | 0.5026 | 0.9676 |
| gpt-oss-20b ★■ | 0.4058 | 0.4619 | 0.9900 | 0.4058 | 0.4619 | 0.9900 |
| **gpt-oss-120b ★■** | **0.6250** | **0.6985** | 0.9254 | **0.6250** | **0.6985** | 0.9254 |
| Claude-3.5-Sonnet | 0.1845 | 0.4430 | 0.8208 | 0.1845 | 0.4434 | 0.8209 |
| Claude-3.7-Sonnet | 0.2212 | 0.4825 | 0.8409 | 0.2321 | 0.4996 | 0.8409 |
| Claude-4 Sonnet | 0.2956 | 0.5832 | 0.7720 | 0.2966 | 0.5870 | 0.7719 |
| Claude-3.7-Sonnet (Thinking) ★ | 0.3343 | 0.5953 | 0.8127 | 0.3661 | 0.6154 | 0.8192 |
| Claude-4 Sonnet (Thinking) ★ | 0.3571 | 0.6085 | 0.7788 | 0.3581 | 0.6079 | 0.7821 |
| Claude-4 Opus (Thinking)* ★ | 0.5389 | 0.7497 | 0.8577 | 0.5389 | 0.7521 | 0.8561 |
| o3-mini ★ | 0.5278 | 0.5954 | 0.9179 | 0.5278 | 0.5954 | 0.9179 |
| o4-mini ★ | 0.6329 | 0.6907 | 0.9153 | 0.6329 | 0.6907 | 0.9153 |
| Gemini 2.5 Flash ★ | 0.5833 | 0.6533 | 0.7911 | 0.5863 | 0.6562 | 0.7901 |
| **GPT-5 ★** | **0.7242** | **0.7645** | 0.9286 | **0.7242** | **0.7645** | 0.9286 |

high API costs (details for costs of experiment runs are added in Table-E.5. Reasoning models, open and closed-source, consistently outperform non-reasoning ones, with a larger gap for open-source models. The top models are gpt-oss-120b (open) and GPT-5 (closed). Factorial analysis of QwQ-32B and GPT-5 shows cascade length as the strongest negative predictor of `Pass@1`. We further find that feeding and bleeding decrease `Pass@1`, while counter-feeding and counter-bleeding have no significant effect. (Appendix G.3). Finally, we compare the distributions of cascade lengths and relation types in model predictions to the ground truth using confusion matrices (Appendix G.5, G.6), showing that models tend to predict simpler cascades with fewer relations and programs, often succeeding only when a functionally equivalent solution exists.

## 5.2 PBEBENCH PERFORMANCE

We evaluate the best open and closed-source models, gpt-oss-120b and GPT-5, on PBEBench, which includes cascades of length 2–20 and harder snapshots of length 25 and 30. On cascades of length 2–20, gpt-oss-120b achieves `Pass@1` of 0.67, average `Edit_Sim` of 0.95, and `Valid_Rate` of 0.96. Per cascade `Pass@1` are values added in (Table 11). GPT-5, being stronger, achieves `Pass@1` of 0.94 on cascades of length 10 with just 1 sample (versus `Pass@1` of 0.84 with 32 for gpt-oss-120b) and 0.44 on cascades of length 20 with 4 samples (versus `Pass@1` of 0.05 with 32 for gpt-oss-120b). Since GPT-5 still performs meaningfully at length 20, unlike gpt-oss-120b which collapses to 5%, we evaluate it on harder cascades of length 20, 25, and 30, where it averages `Pass@1` of 0.175 (Table 13). `Pass@1` trends are shown in Fig. 2a (gpt-oss-120b, 2–20) and Fig. 2b (GPT-5, 20–30). Both models follow the scaling strategies in Section 3.2: gpt-oss-120b uses sampling budget 32 with max sequence length 16384, while GPT-5 uses sampling budget 4 (to mitigate linear cost scaling), a completion limit of 65536 tokens (including reasoning and output tokens), and moderate reasoning effort to make sure the reasoning terminates. Performance declines nearly monotonically with cascade length: gpt-oss-120b drops to about 50% and 5% at lengths 15 and 20, while GPT-5 drops to 14% and 5% at 25 and 30, highlighting reasoning limits in both. Logistic regression on gpt-oss-120b shows cascade length as the strongest negative predictor of `Pass@1`, with feeding, counter-feeding, and counter-bleeding also significant and bleeding associated with success (Table 10).

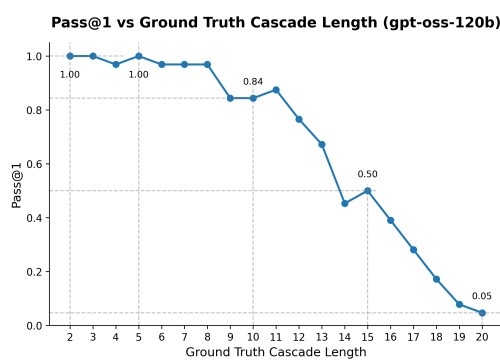

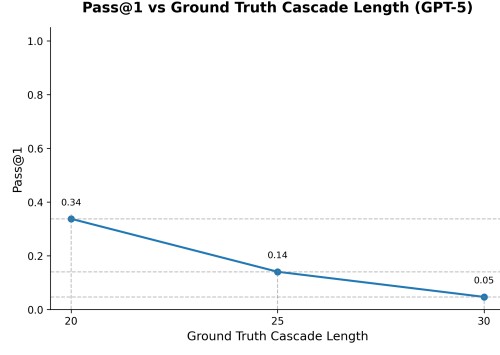

(a) gpt-oss-120b (sampling budget: 32, max sequence length: 16384, cascade: 2-20)

(b) GPT-5 (sampling budget: 4, max completion tokens: 65536, reasoning (medium), cascade: 20, 25, 30)

Figure 2: **Performance across Cascade Lengths on PBEBench:** `Pass@1` for gpt-oss-120b and GPT-5 for various ground truth cascade lengths, a key difficulty measure, shows where inductive reasoning fails and problems become nearly unsolvable despite high compute budgets.

## 5.3 ABLATIONS

We evaluate gpt-oss-20b and gpt-oss-120b on the PBEBench-Lite-MoreEg snapshot (Section 1) and show the results in Table 12. The results show similar to slightly lower performance, which is counterintuitive because historical linguists anecdotally perform better with more examples. We analyze the fraction of examples modified by each program for both PBEBench-Lite and PBEBench-Lite-MoreEg and find that it is 1.56/5 (31%) and 7.07/50 (14%) for each dataset, meaning that while there is greater absolute effect of each program in the latter dataset, the relative effect if lesser (ratio of changed to total examples). This might explain the difficulty for PBEBench-Lite-MoreEg. We also evaluate the effect of scaling the sampling budget for a constant max sequence length and vice versa in Fig 3a (Table 16) and Fig 3b (Table 18) respectively for gpt-oss-120b and Fig 4a (Table 14) and Fig 4b (Table 15) respectively for GPT-5 (we compute k-fold average for GPT-5 to account for variance, exact details of which are provided in E.7). All the plots show rapid initial gains followed by a regime of diminishing returns and eventual saturation.

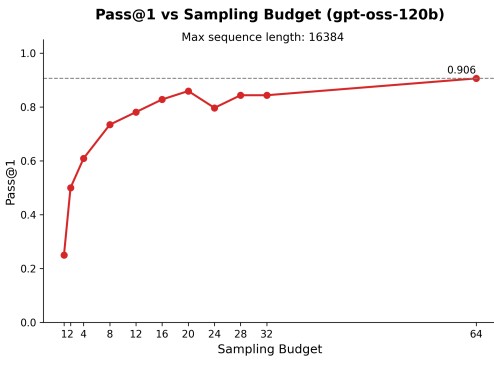

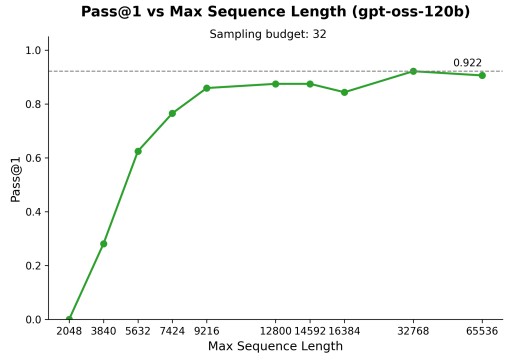

(a) Pass@1 vs Sampling Budget (max seq len: 16384)

(b) Pass@1 vs Max Seq Len (sampling budget: 32)

Figure 3: **Effect of Scaling Strategies on PBEBench (gpt-oss-120b):** Comparison of successive sampling and increased thinking budgets (via max sequence length) on PBEBench instances with ground-truth cascades of length 10, the most complex balanced subset, unsolved yet nearly solvable under greater scaling, allowing a meaningful strategy comparison.

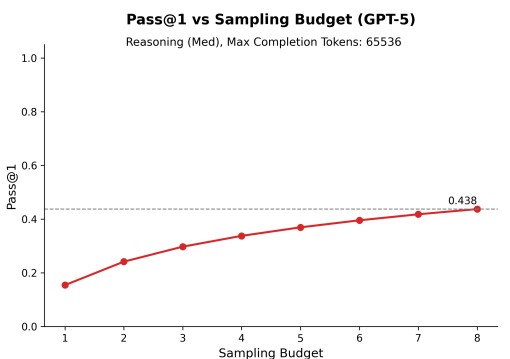 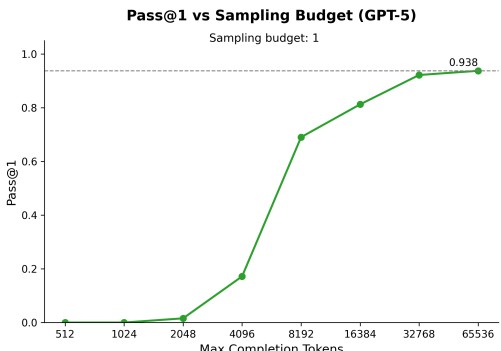

(a) Pass@1 vs Sampling Budget (reasoning (med), max completion tokens: 65536)

(b) Pass@1 vs Max Completion Tokens (sampling budget: 1)

Figure 4: **Effect of Scaling Strategies on PBEBench (GPT-5):** Comparison of successive sampling and larger thinking budgets (via max completion tokens and reasoning effort) on PBEBench with ground-truth cascade lengths 20 and 10 respectively. Cascade 20 is used for the sampling budget experiment as we observe high `Pass@1` (0.938) for cascade 10 with just 1 sample, in the max completion tokens scaling experiment.

## 6   DISCUSSION

After analyzing the results, we address the research questions from Section 1. **RQ1: Which models are good at inductive reasoning?**: Long chain-of-thought models consistently perform better across both open and closed-source categories on PBEBench-Lite, with gpt-oss-120b and GPT-5 being the best in each. **RQ2: What makes a PBE instance hard?**: Factorial and regression analysis (Appendix G.3, G.4) show cascade length as the most influential factor, followed by certain BFCC relations (especially feeding and counter-feeding). Counterintuitively, more examples reduce performance, as the relative density of updates per program decreases with increasing examples (see Sec. 5.3). **RQ3: Can the strongest LLMs solve problems requiring long programs (e.g., real-world forward reconstruction)?**: Even the strongest LLMs struggle on PBEBench, despite large scaling budgets. Real world forward reconstruction problems are even harder, with higher cascade lengths and more examples. **RQ4: What scaling strategies work best for solution search?**: A minimum value for max sequence length is critical for reasoning models, and this requirement grows with model capability. Without it, performance drops to zero. GPT-5, being more capable, has a higher minimum requirement than gpt-oss-120b. Increasing reasoning_effort for GPT-5 further raises this requirement, and with a fixed max sequence length, performance declines due to more unterminated outputs (Fig. 18). Beyond this threshold, either scaling strategy (sampling budget and max sequence length) is roughly equivalent for gpt-oss-120b (e.g., 32768 max sequence length with sampling budget 32 ≈ sampling budget 64 at 16384). However, scaling sampling budget is slower if not parallelized and more redundant, as attempts cannot share steps, whereas a single long chain of thought can. A balanced use of both strategies is therefore necessary.

## 7   CONCLUSION AND FUTURE WORK

PBEBench-Lite is already challenging for many models, especially non-reasoning ones, indicating that inductive reasoning requires long chains of thought at test time. Experiments with the strongest open and closed-source models on PBEBench show that while they handle moderate complexity, performance steadily declines as complexity, mainly driven by ground-truth cascade length, increases. Even with intensive scaling strategies like larger sampling and token budgets, performance eventually saturates, highlighting the reasoning limits of current state-of-the-art models. Nonetheless, results are promising and surpass prior work (Naik et al., 2024; 2025b), pointing to a bright future for solving forward reconstruction problems with LLMs. Moreover, our automated, scalable data generation with controllable complexity could enable increasingly complex synthetic training data, potentially enhancing PBE and inductive reasoning in open-weight models via curriculum-style learning, an avenue we plan to pursue in future work.

## 8 ETHICS STATEMENT

Our work introduces a provably correct, fully automated, and scalable pipeline for generating multi-step Programming by Example (PBE) problems, enabling the evaluation of inductive reasoning in Large Language Models without human supervision. As no human subjects are involved, we do not anticipate ethical concerns. Moreover, since the dataset focuses solely on assessing abstract inductive reasoning in LLMs, it is unlikely to be misused for harmful purposes.

## 9 REPRODUCIBILITY STATEMENT

To facilitate reproduction of our results, we thoroughly document all components of our work. The BFCC relation types, their detectors (Algorithm 2), and proofs of correctness are presented in Appendix C. The rejection-sampling-based data generation process is both described (Section 3.1) and formally specified (Algorithm 1). Detailed statistics and parameters for all generated datasets appear in Section 4.1 and Appendix D.2. Our prompting and program extraction strategy, including scaling techniques, are reported in Section 3.2, while the prompt template is provided in Appendix E.2, and inference parameters for each model are given in Appendix F.2. For closed source models we use specific dated snapshots/checkpoints of models as recorded in Appendix E.6 to aid with reproducibility. To estimate the amount of variance between runs/experiments, we do a small controlled experiment for gpt-oss-120b (Table 17), and compute k-fold average for GPT-5 (Appendix E.7) The evaluation procedure and metrics are described in Section 4.3 and Section 3.2. Finally, we discuss limitations of our work in Appendix B.

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

## A APPENDIX

The Appendix is organized as follows:

1. Appendix B presents the limitations.
2. Appendix C provides the theory and proofs for the relation type classifiers used to control the distribution.
3. Appendix D details the benchmark, including licensing, statistics, and related information.
4. Appendix E gives additional methodological details, including prompts, model selection, licensing, run costs, algorithm pseudocode, snapshots for closed-source models, and the K-fold evaluation used to simulate GPT-5's sampling budget.
5. Appendix F outlines experimental details such as computational environment, inference parameters, and strategies for CoT truncation/control with gpt-oss-120b.
6. Appendix G presents additional results, including data generation efficiency, factorial and logistic regression analyses of instance difficulty, confusion matrices comparing dataset ground truth and model predictions, and extended tables for scaling experiments, effects of additional examples, and performance breakdowns by cascade length.

## B LIMITATIONS

Despite this paper's significant contributions, such as a scalable and provably correct dynamic data generation pipeline for instances of controllable complexity and extensive benchmarking of state-of-the-art reasoning models under popular scaling strategies, several limitations remain:

1. Due to cost, we could not use very large sampling budgets with GPT-5, nor evaluate it on every cascade length, restricting experiments to the most challenging cases where performance drops to nearly zero.
2. Due to randomness in the sampling procedure, it cannot fully balance constraints such as relationship types and cascade lengths, especially for very long cascades, where suitable examples are increasingly rare, leading to high rejection rates and exhaustion of patience $\tau$. Future work could study the impact of targeted sampling, using algorithmic approaches to balance constraints simultaneously.
3. In PBEBench, programs modify on average only two examples, which is less reflective of real sound law induction problems, where many more words may be affected by each rule.

4. We evaluate only the strongest open and closed-source models on PBEBench. While this partly reflects limited time and computing resources, it also ensures the analysis is meaningful, as weaker models plateau much earlier and fail to match gpt-oss-120b and GPT-5 on longer cascades.

## C  THEORETICAL FRAMEWORK

This section provides the proof of correctness of our proposed method for automatically classifying the type of relation between any pair of string-rewrite programs, which is one of our novel contributions.

We propose the function $\text{feeds}(\cdot, \cdot)$, which classifies pairs of rules as feeding or not feeding.

$$\text{feeds}(s_i \to t_i, s_j \to t_j) = \begin{cases} \top & t_i = \varepsilon \wedge |s_j| > 1 \\ \top & t_i \in \text{Substr}(s_j) \wedge t_i \notin \text{Substr}(s_i) \\ \top & t_j \in (\text{Substr}(t_i) \setminus \text{Substr}(s_i)) \\ \top & \text{Pref}(t_i) \setminus \text{Substr}(s_i) \cap \text{Suff}(s_j) \neq \emptyset \\ \top & \text{Suff}(t_i) \setminus \text{Substr}(s_i) \cap \text{Pref}(s_j) \neq \emptyset \\ \bot & \text{otherwise} \end{cases} \tag{1}$$

where $\text{Pref}(s)$, $\text{Suff}(s)$, and $\text{Substr}(s)$ are the multisets of prefixes, suffixes, and substrings of $s$, respectively.

**Definition C.1** (Feeding). Feeding is a relation between pairs of rules $p_i = s_i \to t_i$ and $p_j = s_j \to t_j$, such that $\exists s, t \in \Sigma^*$ such that $s \xrightarrow{p_i} t$ and $t$ includes a string $w$ that meets the structural description of $p_j$ but is not present in $s$.

**Definition C.2** (Bleeding). Bleeding is a relation between pairs of rules $p_i = s_i \to t_i$ and $p_j = s_j \to t_j$, such that $\exists s, t' \in \Sigma^*$ such that $s_i \xrightarrow{p_i} t_i$ and $s_i$ includes a string $w$ that meets the structural description of $p_j$ but is not present in $t_i$.

**Definition C.3** (Substr). $\text{Substr}(s)$ denotes the multiset of substrings of $s$, counting multiple occurances separately.

**Lemma 1:** If $\text{feeds}(p_i, p_j)$ then $p_i$ feeds $p_j$.

*Proof.* Given $u, v, o, s_i, t_i, s_j, t_j \in \Sigma^*$, $s_i \xrightarrow{p_i} t_i$, and $s_j \xrightarrow{p_j} t_j$ there are four types of transformations of $u$ by applying $p_i$ that will yield $v$ such that $s_j \sqsubseteq v$ (where $\sqsubseteq$ indicates "is a substring of"). (1) **Deletion.** Assume that $t_i = \varepsilon$. $\exists wx \in \Sigma^+$ such that $ws_ix \xrightarrow{p_i} xw$. If $s_j = xw$ then $p_i$ feeds $p_j$. (2) **Containment.** $t_i \sqsubseteq s_j \wedge t_i \not\sqsubseteq s_i$, $\exists w, x \in \Sigma+$ such that $w \xrightarrow{p_i} x \wedge s_j \sqsubseteq x \wedge s_j \not\sqsubseteq x$. (3) **Subsumption.** Assume that $s_j \in \text{Substr}(t_i) \setminus \text{Substr}(s_i)$. Given $s_i \xrightarrow{p_i} t_i$, $t_i$ will always contain instances of $s_j$ not present in $s_i$, entailing that $p_i$ feeds $p_j$. (4) **Completion.** Assume that $t_i = uo$ and $s_j = ov$ (so that $o$ is a suffix of $t_i$ and a prefix of $s_j$). $s_iov \xrightarrow{p_i} t_iov = uov = us_j$, entailing that $p_i$ feeds $p_j$ (as with $t_i = ou$ and $s_j = vo$, *mutatis mutandis*). $\square$

**Lemma 2:** If $\neg\text{feeds}(p_i, p_j)$ then $p_i$ does not feed $p_j$

*Proof.* Given $s_i, t_i, s_j, t_ju \in \Sigma^*$, assume for the sake of contradiction two rewrite rules $s_i \xrightarrow{p_i} t_i$ and $s_j \xrightarrow{t}_j$ such that $p_i$ feeds $p_j$ but $s_i, t_i$, and $s_j$ do not satisfy any of the following conditions: **Deletion.** $s_i \neq \varepsilon \vee s_j \neq wx \forall w, x \in \Sigma^+$, **Containment.** $t \neg \sqsubseteq s_j \vee t_i \sqsubseteq s_i$ **Subsumption.** $s_j$ does not occur in $t_i$ except where it occurs in $s_i$. **Completion.** $\nexists u, o, v$ such that $(t_i = ou \wedge s_j = vo) \vee t_i = uo \wedge s_j = ov$. Either $t_i$ is a non-empty string neither containing nor being contained by $s_j$ and sharing no prefix or suffix with $s_j$ or replacing $s_i$ with $t_i$ derives no instances of $s_j$. The first case must be false, since the conditions exhaust the transformations that could yield a string containing $s_j$. The second case must be false, because it contradicts the definition of feeding. $\square$

**Theorem 3** (Feeding)**:** A rule $s_i \to t_i$ feeds a rule $s_j \to t_j$ iff $\text{feeds}(s_i \to t_i, s_j \to t_j)$

*Proof.* Given two rules $p_i = s_i \to t_i$ and $s_j \to t_j$, Lemma 1 proves by enumerating cases that each of the conditions defined for $\text{feed}(p_i, p_j)$ are sufficient for establishing that $p_i$ feeds $p_j$. Lemma 2 proves by enumerating cases than $p_i$ does not feed $p_j$ if none of these conditions are satisfied. $\square$

**Theorem 4** (Bleeding)**:** A rule $p_i = s_i \rightarrow t_i$ bleeds a rule $p_j = s_j \rightarrow t_j$ iff feeds$(t_i \rightarrow s_i, s_j \rightarrow t_j)$

*Proof.* if $\exists u, v \in \Sigma^*, u \xrightarrow{t_i \rightarrow s_i} v$ such that $s_j \sqsubseteq v \wedge s_j \not\sqsubseteq u$, it follows that mapping $s_i \xrightarrow{p_i} t_i$ bleeds $p_j$ (where $s_j \xrightarrow{p_j} t_j$). $\square$

# D   BENCHMARK DETAILS

In this section, we discuss issues such as licensing and the distributional statistics of all the data snapshots created and used in our work.

## D.1   LICENSING

We create a provably correct, fully automated, and scalable pipeline for generating multi-step Programming by Example (PBE) problems, enabling the evaluation of inductive reasoning in LLMs. We plan to release all the benchmark snapshots (PBEBench-Lite, PBEBench-Lite-MoreEg, PBEBench, PBEBench (25, 30)) under the CC BY-SA 4.0 license. Additionally, we produce code that allows you to generate more snapshots, which we also release under the MIT license.

## D.2   BENCHMARK STATISTICS

We report the distributional statistics (like distribution of ground truth cascade lengths or relation types) of all the benchmark snapshots used in this paper below:

**PBEBench-Lite:** We generate a relation type balanced dataset with 1008 instances, 5 examples per PBE problem, and 63 instances per relation type category with cascade lengths ranging from 2 to 5. The alphabet spans $\Sigma_{\text{lite}} = \{a, \ldots, k, u, v, w, x, y, z\}$, each input example contains 2 to 6 letters, and each rule has 1-3 characters on either side of the replace function. The distribution of cascades and relation types is shown in Fig. 5 and Fig. 6.

**PBEBench-Lite-MoreEg:** We generate a relation type balanced dataset with 240 instances, 50 examples per PBE problem and 15 instances per relation type category with cascade lengths ranging from 1 to 5. The alphabet spans $\Sigma_{\text{lite}} = \{a, \ldots, k, u, v, w, x, y, z\}$, each input example contains 2 to 6 letters and each rule has 1-3 characters on either side of the replace function. The distribution of cascades and relation types is shown in Fig. 7 and Fig. 8.

**PBEBench:** We generate a cascade balanced dataset with 1216 instances, 50 examples per PBE problem, and 64 instances per cascade length ranging from 2 to 20. The alphabet spans $\Sigma = \{a, \ldots, z, A, \ldots, Z\}$, each input example contains 2 to 6 letters and each rule has 1-3 characters on either side of the replace function. The distribution of cascades and relation types is shown in Fig. 9 and Fig. 10.

**PBEBench (25, 30):** We generate a cascade balanced dataset with 128 instances, 50 examples per PBE problem, and 64 instances per cascade for cascade length of 25 and 30. The alphabet spans $\Sigma = \{a, \ldots, z, A, \ldots, Z\}$, each input example contains 2 to 6 letters and each rule has 1-3 characters on either side of the replace function. The distribution of cascades and relation types is shown in Fig. 11 and Fig. 12.

# E   METHOD DETAILS

This section provides additional details on the problem proposer and problem solver to support reproducibility. Specifically, it covers:

- **Problem proposer:** the algorithm used for data generation.
- **Problem solver:** the prompt template for inference, the models used in our work (including details, licensing, and snapshots for closed-source models, as well as costs of running them), and the K-Fold analysis employed to efficiently simulate different sampling budgets with expensive closed-source models like GPT-5 while reducing variance.

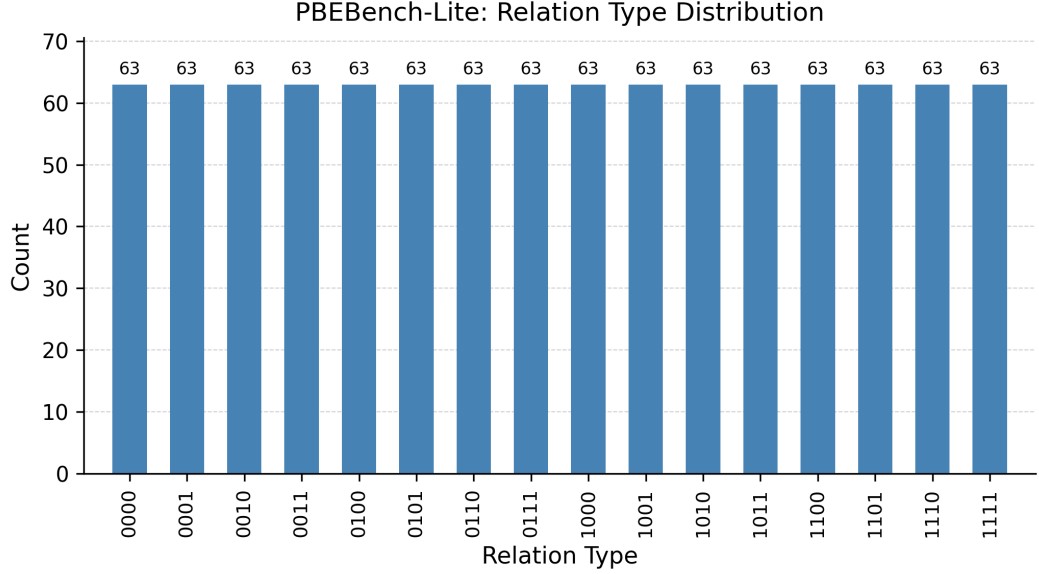

Figure 5: Cascade length distribution for PBEBench-Lite.

Figure 6: Relation type distribution for PBEBench-Lite.

## E.1 ALGORITHM PSEUDOCODE

This section formally expresses the pseudocode/logic behind the data generation algorithm proposed in our work, breaking it down into the rejection sampling subroutine (Algorithm 1) and the relation type classification subroutine (Algorithm 2), respectively.

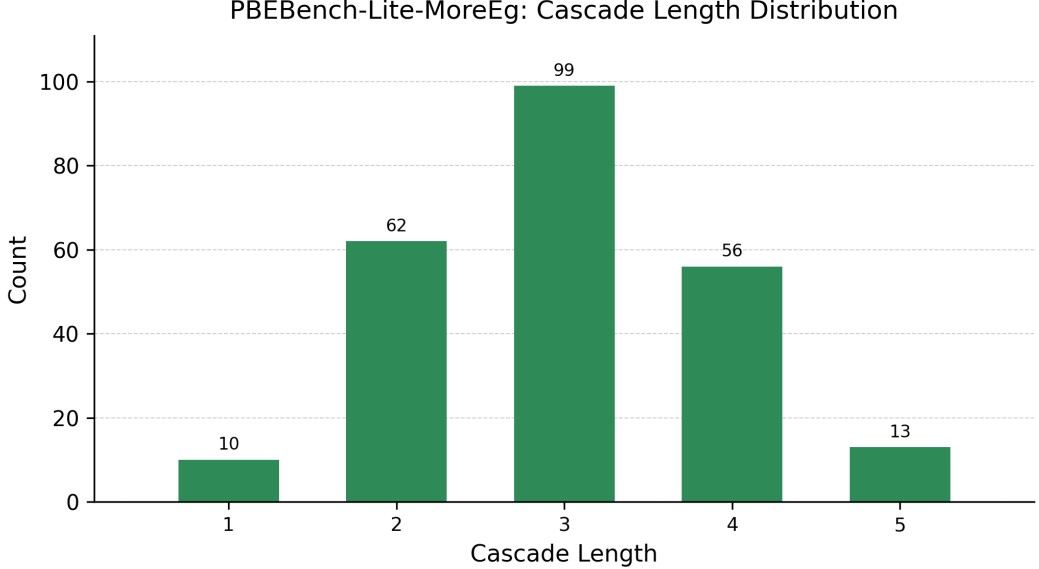

Figure 7: Cascade length distribution for PBEBench-Lite-MoreEg.

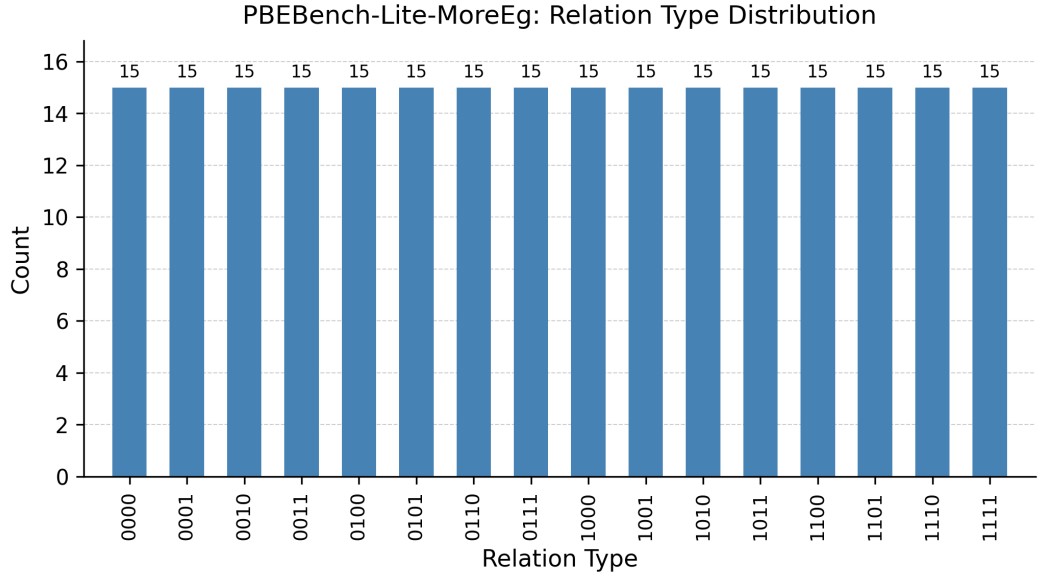

Figure 8: Relation type distribution for PBEBench-Lite-MoreEg.

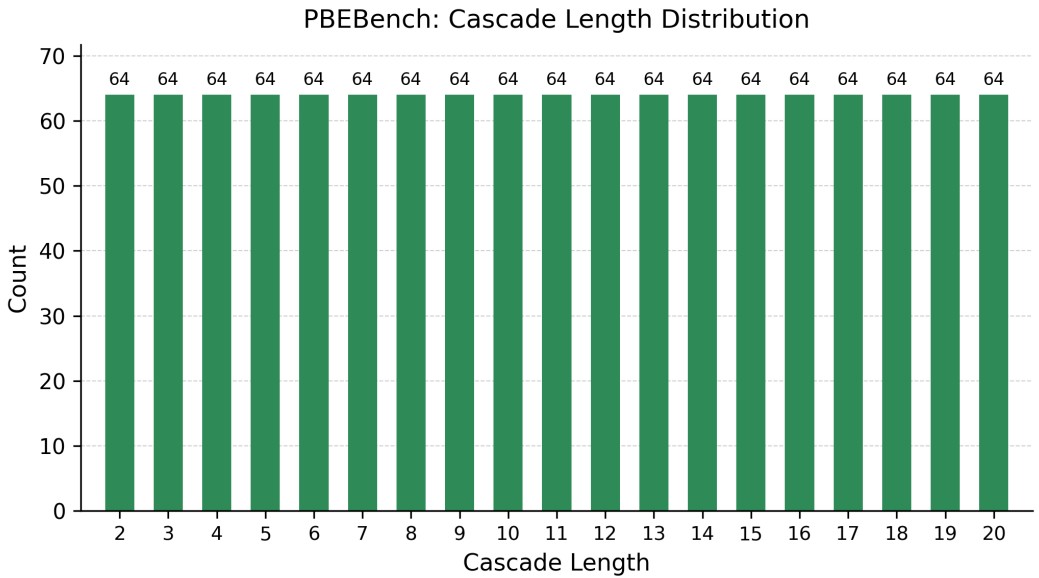

Figure 9: Cascade length distribution for PBEBench.

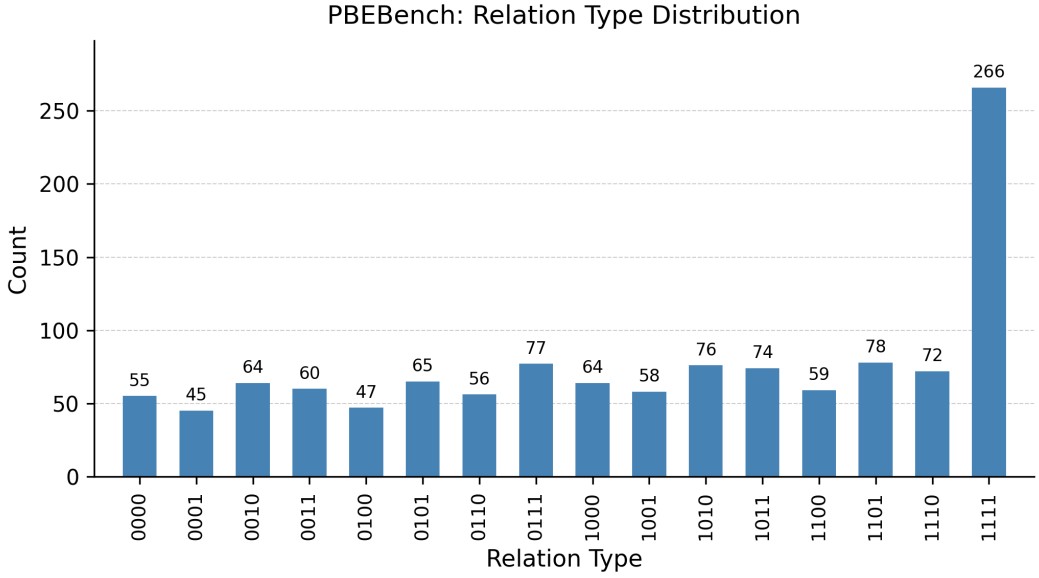

Figure 10: Relation type distribution for PBEBench.

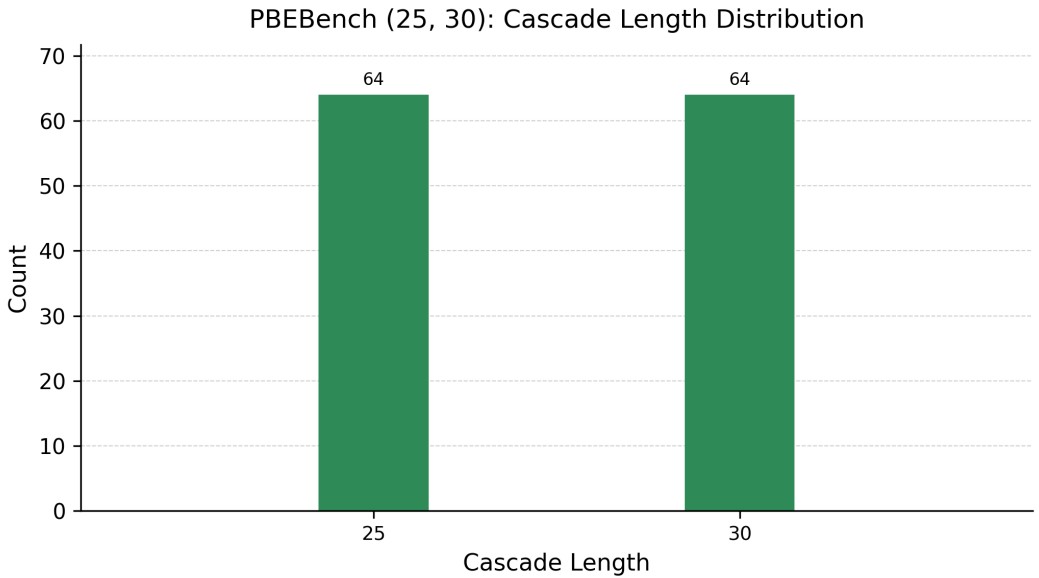

Figure 11: Cascade length distribution for PBEBench (25, 30).

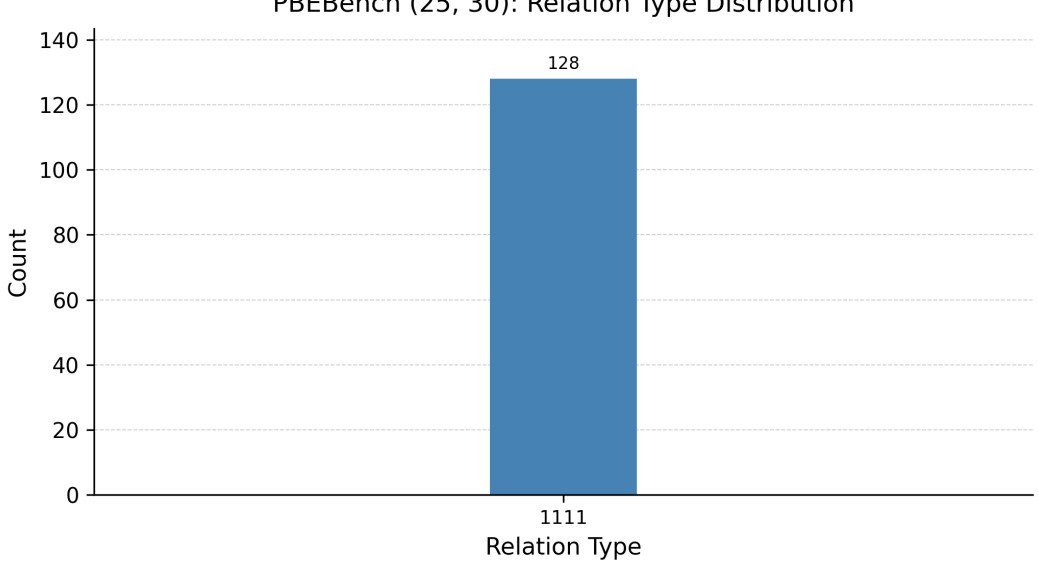

Figure 12: Relation type distribution for PBEBench (25, 30).

---

**Algorithm 1** Rejection Sampling for BFCC Dataset Generation

---

**Require:** Target size $n$; cascade bounds $[\ell_{\min}, \ell_{\max}]$; patience $\tau$
**Require:** Input sampler $p_{\mathcal{I}}$: generates $k$ random strings from vocabulary $\Sigma$
**Require:** Cascade sampler $p_{\mathcal{P}}(\cdot \mid X, \ell)$: generates $\ell$ programs where each replace$(a, b)$ has $a$ sampled from substrings of $X$ and $b$ sampled from $\Sigma$
**Ensure:** Dataset $\mathcal{D}$ balanced over 16 BFCC categories
 1: Initialize quotas $q_c \leftarrow \lfloor n/16 \rfloor$ for all $c \in \{0, 1\}^4$
 2: Initialize $\mathcal{D} \leftarrow \emptyset$, seen signatures $\mathcal{S} \leftarrow \emptyset$, steps $t \leftarrow 0$
 3: **while** $|\mathcal{D}| < n$ **do**
 4: $\quad t \leftarrow t + 1$
 5: $\quad$ Sample length $\ell \sim \text{Uniform}\{\ell_{\min}, \ldots, \ell_{\max}\}$
 6: $\quad$ Sample inputs $X = \{x_1, \ldots, x_k\} \sim p_{\mathcal{I}}$
 7: $\quad$ Sample cascade $\pi = (f_1, \ldots, f_\ell) \sim p_{\mathcal{P}}(\cdot \mid X, \ell)$
 8: $\quad \hat{\pi} \leftarrow []; Y \leftarrow X$
 9: $\quad$ **for** each program $f \in \pi$ **do**
10: $\quad\quad$ **if** $f$ changes at least one element in $Y$ **then**
11: $\quad\quad\quad \hat{\pi} \leftarrow \hat{\pi} \cdot f; Y \leftarrow f(Y)$
12: $\quad\quad$ **end if**
13: $\quad$ **end for**
14: $\quad$ **if** $|\hat{\pi}| < \ell_{\min}$ or $Y = X$ **then**
15: $\quad\quad$ **continue** $\hfill \triangleright$ Reject: insufficient transformation
16: $\quad$ **end if**
17: $\quad \sigma \leftarrow (X, Y, \hat{\pi}, |\hat{\pi}|)$
18: $\quad$ **if** $\sigma \in \mathcal{S}$ **then**
19: $\quad\quad$ **continue** $\hfill \triangleright$ Reject: duplicate
20: $\quad$ **end if**
21: $\quad c \leftarrow \text{ClassifyBFCC}(\hat{\pi})$
22: $\quad$ **if** $t < \tau$ and $q_c > 0$ **then** $\hfill \triangleright$ Before patience: enforce quotas
23: $\quad\quad \mathcal{D} \leftarrow \mathcal{D} \cup \{(X, \hat{\pi}, Y, c)\}$
24: $\quad\quad q_c \leftarrow q_c - 1; \mathcal{S} \leftarrow \mathcal{S} \cup \{\sigma\}$
25: $\quad$ **else if** $t \geq \tau$ **then** $\hfill \triangleright$ After patience: accept any valid instance
26: $\quad\quad \mathcal{D} \leftarrow \mathcal{D} \cup \{(X, \hat{\pi}, Y, c)\}$
27: $\quad\quad \mathcal{S} \leftarrow \mathcal{S} \cup \{\sigma\}$
28: $\quad$ **end if**
29: **end while**
30: **return** $\mathcal{D}$

---

**Algorithm 2** BFCC Classification

---

 1: **function** CLASSIFYBFCC(cascade $\hat{\pi} = [\text{replace}(a_1, b_1), \ldots, \text{replace}(a_m, b_m)]$)
 2: $\quad$ Initialize category vector $(c_F, c_B, c_{CF}, c_{CB}) \leftarrow (0, 0, 0, 0)$
 3: $\quad$ **for** each ordered pair $(i, j)$ where $i \neq j$ **do**
 4: $\quad\quad$ **if** program $i$ feeds program $j$ **then** $\hfill \triangleright b_i$ creates instances of $a_j$
 5: $\quad\quad\quad$ Set $c_F \leftarrow 1$ if $i < j$; set $c_{CF} \leftarrow 1$ if $i > j$
 6: $\quad\quad$ **end if**
 7: $\quad\quad$ **if** program $i$ bleeds program $j$ **then** $\hfill \triangleright a_i$ removes instances of $a_j$
 8: $\quad\quad\quad$ Set $c_B \leftarrow 1$ if $i < j$; set $c_{CB} \leftarrow 1$ if $i > j$
 9: $\quad\quad$ **end if**
10: $\quad$ **end for**
11: $\quad$ **return** $(c_F, c_B, c_{CF}, c_{CB})$
12: **end function**

---

## E.2 PROMPT TEMPLATE

We show the prompt template used for the PBE forward reconstructions task below. This prompt includes the exact instructions and examples given to all the LLMs for performing this task.

---

**PBE Forward Reconstruction Prompt**

Follow the instructions below to solve the code completion task:
We will provide the input corpus and corresponding output corpus. Each element in the corpus is a string, and the output is transformed from the corresponding input using an ordered sequence of "replace" programs. You need to find the correctly constructed and ordered sequence of "replace" programs to transform the entire input corpus into the output corpus. Note that the programs can interact with each other in a way that reduces or increases the number of times they are applied on a given input based on where they are ordered in the sequence. This makes it very important to apply them in the correct order.
The programs should be written using only the Python `replace` function. For example, for a program that replaces all occurrences of "ab" with "bc" it should be written as:
`replace('ab', 'bc')`
Here is an example of the full task:

```
### Inputs
["abc", "ebc", "aba"]

### Outputs
["edc", "edc", "aba"]

### Program Sequence
```python
["replace('bc','dc')", "replace('ad','ed')"]
```
```

While generating the program sequence, you need to abide by the following restrictions:

1. Each program in the sequence should have the form `replace(A, B)`, where `A` and `B` are both strings.

2. Both argument strings `A` and `B` in `replace(A, B)` should have length $\leq$ $\{program\_length\}$. `A` must have length $\geq 1$, while `B` may be empty (i.e., `""`).

3. The maximum number of programs in a sequence is {program_num}.

4. You should only consider the Python `replace` function for specifying programs (each program is a Python `replace` function). You cannot use any other Python modules or functions.

5. Strictly follow the markdown style convention while presenting your final program sequence, and make sure to enclose it in the ```python markdown style code block.

Now, please generate the sequence of programs corresponding to the following input corpus and output corpus:

```
### Inputs
{inputs_list}

### Outputs
{outputs_list}

### Program Sequence
```

---

### E.3 Model Selection Details

Table 2 details all the models chosen for our benchmark and their various attributes to ensure we evaluate a diverse and representative set of LLMs to evaluate which of them excel at inductive reasoning.

Table 2: **Model Selection:** This table details the characteristics of the models benchmarked on PBEBench-Lite. The columns discuss cover the model name, reasoning ability, citation, parameter count, architecture style (MoE vs Dense), and open/closed soruce nature of the chosen models to showcase the diversity of the evaluated models.

| Model Name | Reasoning | Citation | Parameters | MoE | Source |
|---|---|---|---|---|---|
| QwQ-32B | Yes | Team (2025b) | 32B | No | Closed |
| DeepSeek-R1-Distill-Qwen-32B | Yes | DeepSeek-AI (2025) | 32B | No | Closed |
| o3-mini | Yes | OpenAI (2024) | – | No | Closed |
| o4-mini | Yes | OpenAI (2025) | – | No | Closed |
| Qwen3-30B-A3B (Thinking) | Yes | Team (2025a) | 30B | Yes | Open |
| Qwen3-32B | Yes | Team (2025a) | 32B | No | Open |
| Gemini 2.5 Flash | Yes | Comanici et al. (2025) | – | No | Closed |
| Claude-3.7-Sonnet | Yes | Anthropic (2025) | – | No | Closed |
| Claude-4 Sonnet (Thinking) | Yes | Anthropic (2025a) | – | No | Closed |
| Claude-4 Opus (Thinking) | Yes | Anthropic (2025b) | – | No | Closed |
| gpt-oss-20b | Yes | OpenAI (2025b) | 20B | No | Open |
| gpt-oss-120b | Yes | OpenAI (2025b) | 120B | No | Open |
| GPT-5 (Thinking) | Yes | OpenAI (2025a) | – | No | Closed |
| Qwen2.5-32B-Instruct | No | Team (2024) | 32B | No | Open |
| Claude-3.5-Sonnet | No | Anthropic (2024) | – | No | Closed |
| GPT-5 (Non-Thinking) | No | OpenAI (2025a) | – | No | Closed |
| Codestral-22B | No | AI (2024) | 22B | No | Open |
| Qwen2.5Coder-32B-Instruct | No | Team (2024) | 32B | No | Open |
| Qwen3-Coder-30B-A3B-Instruct | No | Team (2025a) | 30B | Yes | Open |

### E.4 Licenses for Evaluated Models

We list the licenses used for each evaluated open and closed source models in Table 3.

### E.5 Costs for Experiment Runs

We document the costs of the expensive experiments carried out for closed source models in Table 4.

### E.6 Snapshots used for closed source models

We document the Snapshots used for closed source models Table 5.

### E.7 K-Fold analysis for GPT-5

We observed a variance of up to 10% in Pass@1 for GPT-5. To account for this variance, we report the aggregated score over all available samples when computing Pass@1 values. For instance, in the sampling experiment for GPT-5 shown in Fig. 4a, we perform 8 independent runs and compute the average score over all possible k-run combinations, yielding scores for sampling budgets $1 \leq k \leq 8$.

## F Experimental Details

In this section, we provide additional experimental details, including the computational environment and the inference and sampling parameters used for all the LLMs evaluated in our work. We also briefly discuss the strategies explored to achieve finer-grained control over the thinking budget of

Table 3: Licenses for open and closed source models.

| Model | License |
|---|---|
| Codestral-22B | Mistral Non-Production License (MNPL) |
| Qwen2.5-32B-Instruct | Apache 2.0 |
| Qwen2.5Coder-32B-Instruct | Apache 2.0 |
| Qwen3-32B | Apache 2.0 |
| Qwen3-Coder-30B-A3B-Instruct | Apache 2.0 |
| QwQ-32B | Apache 2.0 |
| Qwen3-32B | Apache 2.0 |
| Qwen3-30B-A3B | Apache 2.0 |
| Qwen3-Coder-30B-A3B-Instruct | Apache 2.0 |
| DeepSeek-R1-Distill-Qwen-32B | MIT |
| o3-mini | API (OpenAI EULA) |
| o4-mini | API (OpenAI EULA) |
| GPT-5 | API (OpenAI EULA) |
| gpt-oss-20b | Apache 2.0 |
| gpt-oss-120b | Apache 2.0 |
| Gemini 2.5 Flash Preview 04-17 | API (Google EULA) |
| Claude-3.5-Sonnet | API (Anthropic EULA) |
| Claude-3.7-Sonnet | API (Anthropic EULA) |
| Claude-4-Sonnet | API (Anthropic EULA) |
| Claude-4-Opus | API (Anthropic EULA) |

Table 4: Documented costs for select experiment runs.

| Model | Experiment | Cost |
|---|---|---|
| Claude-4.1-Opus | PBE-Bench Lite Performance | $40 (20% of dataset) |
| GPT-5 | Cascade Length Experiment | $190 |
| GPT-5 | Sampling Experiment | $165 |
| GPT-5 | CoT Experiment | $50 |
| GPT-5 | PBE-Bench Lite Performance | $50 |
| Claude Sonnet Thinking 3.7 | PBE-Bench Lite Performance | $30 |
| Claude Sonnet Thinking 4 | PBE-Bench Lite Performance | $30 |
| o3-mini | PBE-Bench Lite Performance | $65 |
| o4-mini | PBE-Bench Lite Performance | $65 |

Table 5: Exact snapshots used for closed source models.

| Model | Snapshot |
|---|---|
| o3-mini | o3-mini-2025-01-31 |
| o4-mini | o4-mini-2025-04-16 |
| Claude 3.5 Sonnet | claude-3-5-sonnet-20241022 |
| Claude 3.7 Sonnet | claude-3-7-sonnet-20250219 |
| Claude 4 Sonnet | claude-sonnet-4-20250514 |
| Claude 4.1 Opus | claude-opus-4-1-20250805 |
| GPT-5 | gpt-5-2025-08-07 |
| Gemini 2.5 | gemini-2.5-flash |

gpt-oss-120b for scaling experiments. These strategies were ultimately unsuccessful due to gpt-oss-120b's test-time behavior, leading us to instead study the effect of varying the maximum sequence length directly.

Table 6: Sampling parameters used for inference across all models runs. "Max tokens" refers to the total number of tokens (output + thinking tokens) for models that support it. "Top-p" controls nucleus sampling. "Temperature" sets the randomness of token selection. "Thinking budget" is the number of thinking tokens, applicable only to models that support this feature. GPT-5, o3-mini, and o4-mini support "reasoning_effort" parameter, which is a qualitative measure of "Thinking Budget". These models also do not support temperature and top_p parameters.

| Model | Max Tokens | Top P | Temperature | Thinking Budget(s) |
|---|---|---|---|---|
| Codestral-22B | 2048 | 0.95 | 0.7 | - |
| Qwen2.5-32B-Instruct | 512 | 0.95 | 0.7 | - |
| Qwen2.5Coder-32B-Instruct | 512 | 0.95 | 0.7 | - |
| QwQ-32B | 8192 | 0.95 | 0.7 | - |
| Qwen/Qwen3-32B (with CoT) | 8192 | 0.95 | 0.7 | - |
| Qwen/Qwen3-32B | 8192 | 0.95 | 0.7 | - |
| Qwen3-30B-A3B | 8192 | 0.95 | 0.7 | - |
| DeepSeek-R1-Distill-Qwen-32B | 8192 | 0.95 | 0.7 | - |
| o3-mini | 8192 | - | - | reasoning_effort="medium" |
| o4-mini | 8192 | - | - | reasoning_effort="medium" |
| Gemini 2.5 Flash | dynamic | 0.95 | 0.7 | dynamic |
| Claude-3.5-Sonnet | 8192 | 0.95 | 0.7 | - |
| Claude-3.7-Sonnet | 10000 | 0.95 | 0.7 | - |
| Claude-3.7-Sonnet (Thinking) | 10000 | 0.95 | 1 (default) | 2048 |
| gpt-oss-20b | 8192 | 0.95 | 0.7 | - |
| gpt-oss-120b | 8192 | 0.95 | 0.7 | - |
| GPT-5 | 8192 | - | - | reasoning_effort="medium" |
| Claude-4 sonnet | 8192 | 0.95 | 0.7 | - |
| Claude-4 sonnet (Thinking) | 8192 | 0.95 | 1 (default) | 2048 |
| Claude-4 opus (Thinking) (20% of dataset) | 8192 | 0.95 | 1 (default) | 2048 |
| Qwen/Qwen3-Coder-30B-A3B-Instruct | 2048 | 0.95 | 0.7 | - |

## F.1 COMPUTATIONAL ENVIORNMENT

We conduct experiments on a Linux server equipped with NVIDIA A100 80GB GPUs (Ampere architecture), CUDA 12.9, and driver version 575.51.03. Each job had access to 100 GB of CPU memory and up to 16 CPU cores. The GPU allocation varied with model size with gpt-oss-120b and most 32B models requiring 2 A100 GPUs. The experiments on PBEBench-Lite took multiple hours, while each cascade on PBEBench took nearly 8 hours for sampling budget of 32 and 16384 max sequence length, prompting use to parally run multiple cascades across several GPUs. We used vLLM for inference of all the open weight models and multi-threading for inference of closed source models like GPT-5 to speed up all inference experiments.

## F.2 INFERENCE/SAMPLING PARAMETERS

We show the sampling parameters used for all the models in Table 6. The max tokens are the total output tokens the model can generate (including thinking tokens), while the thinking budget(s) captures only the chain-of-thought or reasoning related tokens. The top-p is the cumulative probability cutoff used for nucleus sampling, while the temperature is for controlling the degree of randomness in the sampling. We report the max tokens and thinking tokens wherever possible based on the providers (for some models you can only control the total tokens, while for some you can only control thinking tokens). For some models like Gemini 2.5 Flash Preview, the model has a mode where it first reasons about how much thinking is required based on how complex it determines the problem to be. We use this setting for the experiments in Table 1. However, we also do experiments comparing the effect of varying token budgets (2048, 4096, 8192) for QwQ and Gemini 2.5 Flash Preview, hence we highlight the default setting used for Table 1 for these models in bold.

## F.3 COT TRUNCATION EXPERIMENT DETAILS

For gpt-oss-120b, we attempt to reduce the model's *thinking budget* and introduce it as a parameter independent of *max tokens*. To achieve this, we run two inferences:

1. In the first pass, we set *max tokens* equal to the desired thinking budget. We then check whether the Chain-of-Thought Truncation token appears in the response.

2. If it does not appear, we run a second inference, appending the following string as assistant context to the model's input:

```
early_stop_instruction = "Considering the thinking token budget,
I will not generate any more reasoning tokens, and provide the
final answer '.\n"
```

In the second generation, however, the model begins with "We need to produce final answer" and then continues reasoning as usual, ignoring our instruction.

In variations, we appended THINKING_END_TOKEN, and <FINAL_OUTPUT_START_TOKEN> to the *early_stop_instruction*, but observed the same behavior. We also tried placing a modified version of *early_stop_instruction* in the *user* role instead of the assistant role, again without effect. Finally, we reduced *max tokens* to 300 in the second generation. This did not prompt the model to produce a final answer either; instead, It significantly increased the rate of null outputs, rising from 11% to 77%. We therefore conclude that for gpt-oss-120b, it is not possible to enforce the truncation of the chain-of-thought budget independently of the total output token budget.

## G  ADDITIONAL RESULTS

This section presents some additional detailed results over the PBEBench-Lite and PBEBench, and PBEBench (25, 30) snapshots, such as the detailed tables for the performance vs ground truth cascade length, scaling ablations, etc. It also contains the results of analyzing the effect of changing the number of examples (PBEBench-Lite-MoreEg snapshot). It also contains the results of logistic regression analysis on factors affecting instance difficulty on PBEBench with gpt-oss-120b and factorial analysis on reasoning-capable models, as well as some representative models on PBEBench-Lite. Finally, we also present results for two types of confusion matrices that visualize the distributional differences between cascade lengths and relation types of the ground truth and predicted cascades.

### G.1  EFFECT OF MORE EXAMPLES

Table 12 illustrates the effect of varying the number of examples per PBE instance while keeping other factors, such as cascade distribution and relation type balance, constant using the PBEBench-Lite-MoreEg snapshot. The results for gpt-oss-20b and gpt-oss-120b show largely similar performance for gpt-oss-20b, but a decrease for gpt-oss-120b. This suggests that increasing the number of examples can sometimes make the task harder for LLMs, which is counterintuitive, as more examples would ideally simplify the task. Analysis of the average number of changes per program reveals that PBEBench-Lite has fewer absolute changes (1.56 words out of 5 on average) compared to PBEBench-Lite-MoreEg (7.07 words out of 50 on average). However, the relative number of changes is lower in PBEBench-Lite-MoreEg (14% vs. higher in PBEBench-Lite), indicating lower information density, which may explain why the task becomes harder despite having more examples.

### G.2  EFFICIENCY OF PROBLEM PROPOSER

For each desired ground-truth cascade length, our data generation procedure is designed to yield a balanced distribution across relation type categories, where each category is represented by a binary vector. Let $U$ denote the ideal uniform distribution over all categories, and let $Q$ denote the empirical distribution obtained from the generated data. To quantify the deviation of $Q$ from the ideal $U$, we use the Kullback–Leibler (KL) divergence:

$$D_{\mathrm{KL}}(U||Q) = \sum_x U(x) \log \frac{U(x)}{Q(x)}$$

where $x$ ranges over all relation type categories. By construction, $D_{\mathrm{KL}}(U||Q) \geq 0$ and equals zero only when $Q = U$. We also **apply smoothing** for categories where zero instances are observed in the empirical distribution (missing categories) to prevent the divergence from shooting up to infinity for these cases.

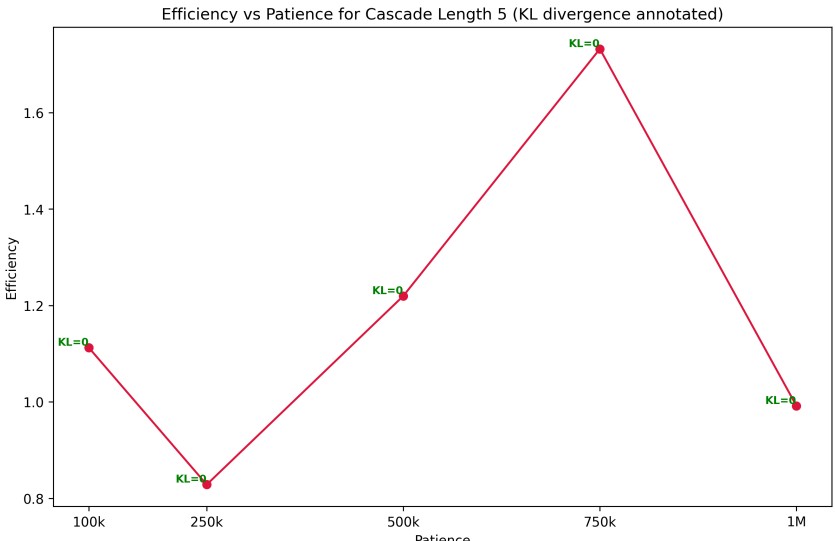

Figure 13: **Efficiency vs Patience:** Efficiency of the rejection sampling process for generating ground truth cascades of length 5 for various values of the patience parameter. The KL divergence from the ideal balanced distrbution is annotated per point with zero corresponding to achieving a perfectly balanced distribution.

Our data generation algorithm (the *problem proposer*) employs rejection sampling to enforce balance across categories. Initially, each sampled datapoint is accepted only if it maintains near-uniform coverage across categories. As sampling progresses, however, this constraint becomes increasingly difficult to satisfy, and the efficiency of rejection sampling deteriorates due to a growing rejection rate. To mitigate this, we introduce a *patience* parameter that limits the number of constrained steps. Once the patience threshold (e.g., 100000) is exhausted, the algorithm relaxes the balancing constraint and accepts datapoints unconditionally. This enables continued large-scale data generation, though at the cost of increased divergence $D_{\mathrm{KL}}(U||Q)$ from the uniform distribution. Importantly, this relaxation applies only to category balancing: the cascade length constraints remain strictly enforced, and each generated program must still achieve the target ground-truth cascade length and modify at least one example.

We visualize the efficiency of our data generation process for different values of the patience parameter $\tau$—100,000; 250,000; 500,000; 750,000; and 1,000,000—across cascade lengths 5, 10, 15, 20, and 25, annotating each point with the KL divergence from the desired uniform distribution over all 16 relation type categories. Points with zero divergence (perfectly balanced distribution) are marked in green, while others are marked in red. The plots of efficiency versus patience for each cascade length are shown in Fig. 13–17. In all plots, the y-axis represents percentage efficiency (so 1 corresponds to 1% or 0.01). For cascade lengths 5 and 10, patience has little effect since perfect KL is always achievable; variations reflect random fluctuations across runs. For cascade lengths 15–25, perfect KL is no longer guaranteed and patience begins to influence efficiency. As expected, efficiency generally decreases with higher patience due to discarding more examples. For lengths 15 and 25, slightly lower KL can be achieved with greater patience, while for length 20, KL remains unchanged, indicating diminishing returns: higher patience does not necessarily reduce divergence from uniform distribution. These results motivate selecting a reasonable, but not excessive, patience value.

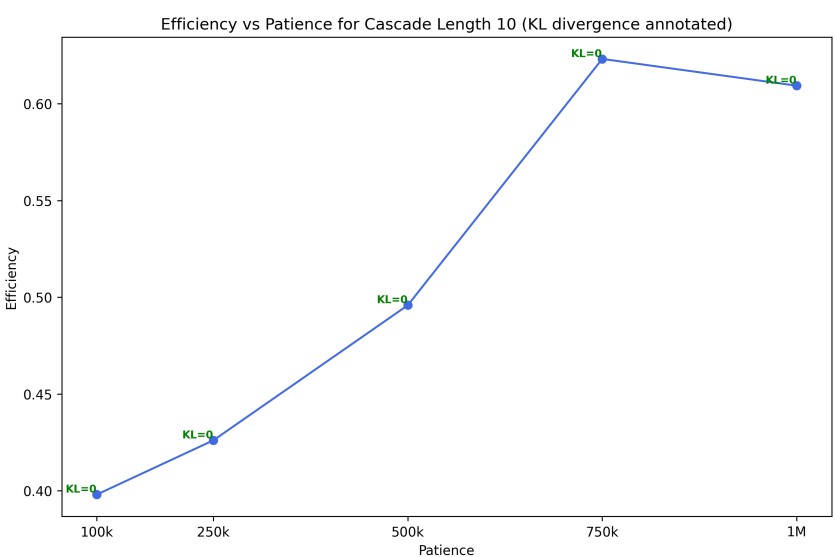

Figure 14: **Efficiency vs Patience:** Efficiency of the rejection sampling process for generating ground truth cascades of length 10 for various values of the patience parameter. The KL divergence from the ideal balanced distrbution is annotated per point with zero corresponding to achieving a perfectly balanced distribution.

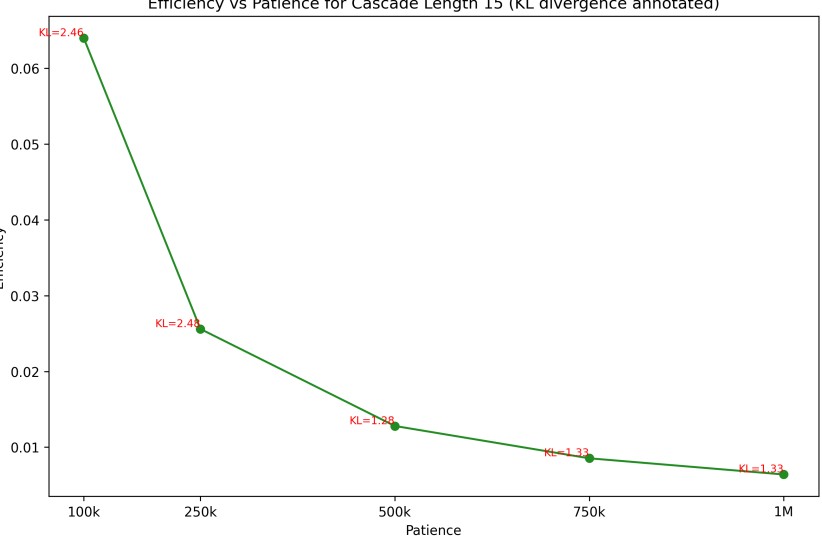

Figure 15: **Efficiency vs Patience:** Efficiency of the rejection sampling process for generating ground truth cascades of length 15 for various values of the patience parameter. The KL divergence from the ideal balanced distrbution is annotated per point with zero corresponding to achieving a perfectly balanced distribution.

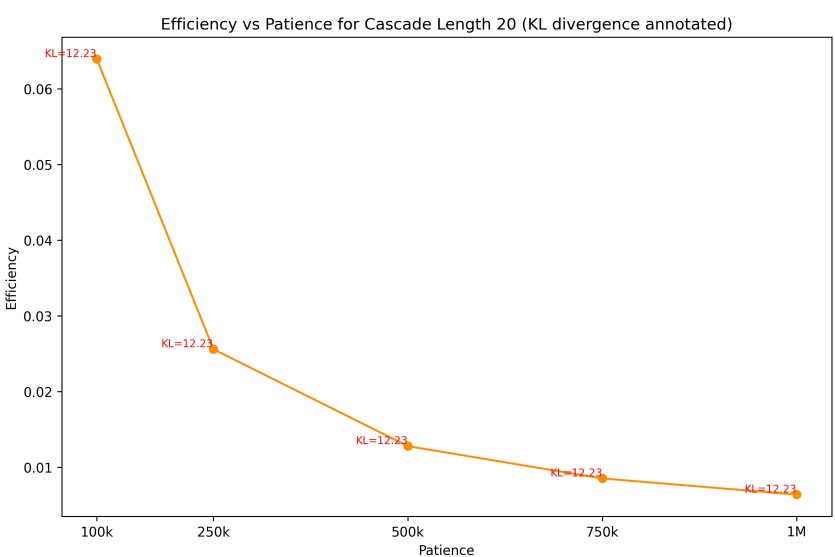

Figure 16: **Efficiency vs Patience:** Efficiency of the rejection sampling process for generating ground truth cascades of length 20 for various values of the patience parameter. The KL divergence from the ideal balanced distrbution is annotated per point with zero corresponding to achieving a perfectly balanced distribution.

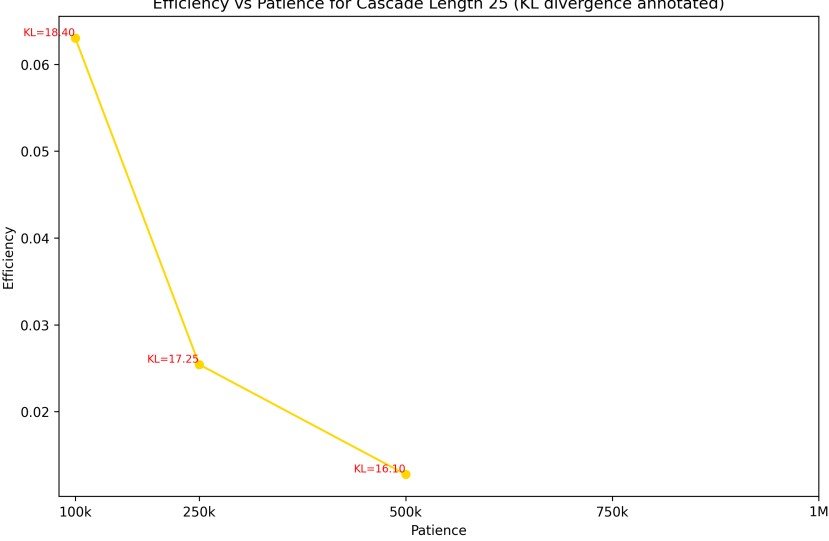

Figure 17: **Efficiency vs Patience:** Efficiency of the rejection sampling process for generating ground truth cascades of length 25 for various values of the patience parameter. The KL divergence from the ideal balanced distrbution is annotated per point with zero corresponding to achieving a perfectly balanced distribution.

### G.3 FACTORIAL ANALYSIS

We conduct a factorial analysis to answer the following questions:

#### G.3.1 EFFECT OF LONG CHAIN OF THOUGHT REASONING

We analyzed the effect of long chain-of-thought (LCoT) reasoning by selecting three models where it can be toggled on or off (Qwen3-32B, Claude-3.7-Sonnet, and Claude-4-Sonnet) on PBEBench-Lite (1008 instances) to evaluate its benefits. Independent variables included `model_id` (Qwen, Claude-3.7, or Claude-4), `reasoning` (LCoT enabled or not), `cascade_len` (ground-truth cascade length), and the presence of BFCC relations: `feeding`, `bleeding`, `counter-feeding`, and `counter-bleeding`. The dependent variable was binary Pass@1 (`passing`) per instance.

We fit a binary logistic regression model with `model_id`, `reasoning`, `feeding`, `bleeding`, `counter-feeding`, and `counter-bleeding` as nominal predictors, and `cascade_len` as a numeric covariate. All pairwise interaction terms were included. The Deviance goodness-of-fit test was non-significant, $\chi^2(6019) = 5398.5$, $p = 1.0$, indicating adequate model fit. The model explained 24.53% of the variance in passing ($R^2_{\text{adj}}$). Wald tests for main effects and significant interactions are summarized in Table 7.

Table 7: **Reasoning Manipulation Analysis on PBEBench-Lite:** Wald test results for main effects and significant interactions in the logistic regression predicting passing.

| Term | df | $\chi^2$ | $p$ | Interpretation |
|------|-----|------|------|----------------|
| Total model | 28 | 1027.68 | .000 | Deviance R^2 (adj) = 24.53% |
| cascade_len | 1 | 415.35 | .000 | Passing is less likely as cascade_len increases |
| model_id | 2 | 128.02 | .000 | The two sonnet models have a higher pass rate than Qwen |
| reasoning | 1 | 18.70 | .000 | Reasoning models are more likely to pass |
| feeding | 1 | 15.69 | .000 | Feeding reduces probability of passing |
| bleeding | 1 | 36.10 | .000 | Bleeding reduces probability of passing |
| counter-feeding | 1 | 21.37 | .000 | Counter-feeding reduces probability of passing |
| counter-bleeding | 1 | 4.67 | .031 | Counter-bleeding increases probability of passing |
| model_id × reasoning | 2 | 189.59 | .000 | For reasoning models, Qwen has a higher probability of passing than the two Sonnet models, which are not different from each other. For non-reasoning models, Claude 4 is better than Clause 3.7, which is better than Qwen. |
| feeding × bleeding | 1 | 33.17 | .000 | Adding either feeding or bleeding reduces the probability of a pass, but both together is better than just feeding. Just bleeding is not worse than having both and not better than just feeding. |
| bleeding × counter-bleeding | 1 | 30.86 | .000 | With counter-bleeding, bleeding loses its effect on probability of a pass. |

### G.3.2 ANALYSIS OF MODELS OF VARYING CAPABILITIES

We analyzed what makes PBEBench-Lite instances difficult for three representative models of different capabilities. Independent variables included `cascade_len` (ground-truth cascade length) and the presence of BFCC relations: `feeding`, `bleeding`, `counter-feeding`, and `counter-bleeding`. The dependent variable was binary Pass@1 (`passing`). All models were fit with binary logistic regression, including all pairwise interaction terms. Results are summarized below.

**Codestral-22B (weakest model).** For Codestral-22B, only the effect of `cascade_len` was significant. Passing was less likely as the cascade length increased. Other predictors showed no detectable effects.

**QwQ-32B (moderately good model).** For QwQ-32B, the model fit the data adequately, $\chi^2(996) = 1946.72$, $p = .129$. The model explained $19.71\%$ of variance ($R^2_{\text{adj}}$). Wald tests are summarized in Table 8. Cascade length, feeding, and bleeding all significantly reduced the probability of passing, while counter-feeding and counter-bleeding showed no effects. An interaction revealed that the joint presence of feeding and bleeding was less detrimental than either alone.

Table 8: **QwQ-32B Analysis on PBEBench-Lite:** Wald test results for main effects and significant interactions in the logistic regression predicting passing.

| Term | df | $\chi^2$ | $p$ | Interpretation |
|---|---|---|---|---|
| Total model | 11 | 197.43 | .000 | Deviance R^2 (adj) = 19.71% |
| cascade_len | 1 | 114.54 | .000 | Passing is less likely as cascade_len increases |
| feeding | 1 | 5.74 | .017 | Feeding reduces the probability of a pass |
| bleeding | 1 | 7.37 | .007 | Bleeding reduces the probability of a pass |
| counter-feeding | 1 | .22 | .637 | No effect |
| counter-bleeding | 1 | .47 | .495 | No effect |
| feeding $\times$ bleeding | 1 | 10.03 | .002 | Though bleeding or feeding alone reduce the probability of a pass, if they are both present, they do not. |

**GPT-5 (strongest model).** For GPT-5, the model fit the data adequately, $\chi^2(996) = 982.16$, $p = .617$. The model explained $17.28\%$ of variance ($R^2_{\text{adj}}$). Wald tests are summarized in Table 9. Cascade length, feeding, and bleeding significantly reduced passing. Counter-feeding and counter-bleeding had no main effects, but an interaction showed that bleeding reduced passing only when counter-bleeding was absent.

### G.4 LOGISTIC REGRESSION ANALYSIS

We conduct a logistic regression analysis on gpt-oss-120b predictions on PBEBench for cascades of length 2 to 20, on the following factors influencing difficulty: ground-truth cascade length (`cascade_len`), the presence of BFCC relations: `feeding`, `bleeding`, `counter_feeding`, and `counter_bleeding`. The dependent variable was binary Pass@1 (passing) per instance. The logistic regression analysis reveals that all the analyzed factors have a statistically significant impact on the success or passing, with feeding, counter feeding, and cascade length having strong negative effects, with cascade length having the strongest impact. However, bleeding has a slight positive effect, and counter-bleeding has a very weak negative effect. This indicates that the presence of bleeding can make the problems easier to solve, while counter bleeding only has a small effect on increasing hardness.

Table 9: **GPT-5 Analysis on PBEBench-Lite:** Wald test results for main effects and significant interactions in the logistic regression predicting passing.

| Term | df | $\chi^2$ | $p$ | Interpretation |
|---|---|---|---|---|
| Total model | 11 | 158.16 | .000 | Deviance R^2 (adj) = 17.28% |
| cascade_len | 1 | 63.76 | .000 | Passing is less likely as cascade_len increases |
| feeding | 1 | 9.48 | .002 | Feeding reduces the probability of a pass |
| bleeding | 1 | 4.95 | .026 | Bleeding reduces the probability of a pass |
| counter-feeding | 1 | 1.16 | .282 | No effect |
| counter-bleeding | 1 | .54 | .464 | No effect |
| bleeding $\times$ counter-bleeding | 1 | 8.50 | .004 | Bleeding only reduces probability of a pass if counter-bleeding is not present |

Table 10: **Logistic Regression Difficulty Analysis of gpt-oss-120b on PBEBench:** Model predictions were analyzed with a sampling budget of 32 and maximum sequence length of 16,384. Each attempt across the 1,216 instances was treated as a datapoint, yielding 38,912 datapoints in total.

| Term | Coefficient | $p$ |
|---|---|---|
| intercept | 2.152 | 0 |
| feeding | -0.254 | 4.77e-19 |
| bleeding | 0.134 | 2.92e-6 |
| counter_feeding | -0.176 | 6.27e-10 |
| counter_bleeding | -0.08 | 0.005 |
| cascade_len | -0.347 | 0 |

### G.5 CONFUSION MATRICES FOR CASCADE LENGTHS

We analyze the length of model-predicted cascades against the ground truth cascades to analyze if the models tend to find more or fewer rules than the ground truth cascade for both successful/passing (Pass@1 = 1) and failure cases. The results on the PBEBench-Lite dataset across all models for successful cases are shown in Fig 19 and Fig. 20. The plots reveal that for successful cases, the models tend to largely find solutions of the correct length for shorter cascades, but for longer ground truth cascades, they tend to find fewer solutions in general, but can surprisingly find shorter solutions as well. For failure scenarios, we note a bigger spread and almost see more cases of the LLMs generating longer programs than the ground truth. This might indicate that for the more complex cases, the LLMs tend to overthink and end up generating more complex cascades that don't work. We also see a large fraction of invalid programs, with this fraction growing more and more for longer ground truth cascades (more complex cases).

### G.6 CONFUSION MATRICES FOR BFCC RELATIONS

We analyze the types of relations present in the model-generated program cascades and compare them against the ground truth relations and visualize the results via confusion matrices. The results are on the PBEBench-Lite dataset across all models and separated by whether the model succeeds or fails (based on Pass@1). We normalize each row (fraction of predicted cases for each possible relation type for a given ground truth type) and show the overall results for successful cases in Fig. 23 and failure cases in Fig. 24. While the per-model results for success cases and failure cases span from Fig 25 to Fig 64. We also plot a simplified version of the confusion matrix that looks at each relation at a time and analyze true positives, false positives, false negatives and true negatives spearately for each relation type for both passing (Fig. 21) and non-passing cases (Fig. 22) aggregated across all the models. These show an interesting pattern where for successful cases for almost all relation types have relatively high false negative rates but it is especially bad for feeding (72% false negatives)

Table 11: **Performance across Cascade Lengths (gpt-oss-120b):** Variation of the performance of gpt-oss-120b across cascades of length 2-20 with sampling budget of 32 and max sequence length of 16384.

| Cascade Length | Pass@1 | |
|---|---|---|
| | First Code Block | Last Code Block |
| 2 | 1 | 1 |
| 3 | 1 | 1 |
| 4 | 0.9688 | 0.9688 |
| 5 | 1 | 1 |
| 6 | 0.9688 | 0.9688 |
| 7 | 0.9688 | 0.9688 |
| 8 | 0.9688 | 0.9688 |
| 9 | 0.8438 | 0.8438 |
| 10 | 0.8438 | 0.8438 |
| 11 | 0.875 | 0.875 |
| 12 | 0.7656 | 0.7656 |
| 13 | 0.6719 | 0.6719 |
| 14 | 0.4531 | 0.4531 |
| 15 | 0.5 | 0.5 |
| 16 | 0.3906 | 0.3906 |
| 17 | 0.2812 | 0.2812 |
| 18 | 0.1719 | 0.1719 |
| 19 | 0.0781 | 0.0781 |
| 20 | 0.0469 | 0.0469 |

Table 12: **PBEBench-Lite-MoreEg Performance:** We compute the Pass@1 and edit similarity as the coarse and fine-grained evaluation, respectively, for each model. ■- indicates mixture-of-experts (or MoE) model and ★- indicates reasoning on for model.

| Model | First Code Block | | | Last Code Block | | |
|---|---|---|---|---|---|---|
| | Pass@1 | Edit Sim | Valid Rate | Pass@1 | Edit Sim | Valid Rate |
| gpt-oss-20b ★■ | 0.3958 | 0.4662 | 0.9748 | 0.3958 | 0.4662 | 0.9748 |
| gpt-oss-120b ★■ | 0.5542 | 0.7018 | 0.9211 | 0.5542 | 0.7018 | 0.9211 |

Table 13: **Performance across Cascade Lengths (GPT-5):** Variation of the performance of GPT-5 across cascades with sampling budget ranging from 1 to 4, medium reasoning effort and 65536 max completion tokens (which includes both reasoning and output tokens).

| Cascade Length | Sampling Budget | Pass@1 |
|---|---|---|
| 20 | 1 | 0.1543 |
| | 2 | 0.2416 |
| | 3 | 0.2974 |
| | 4 | 0.3377 |
| 25 | 1 | 0.0547 |
| | 2 | 0.0911 |
| | 3 | 0.1172 |
| | 4 | 0.1406 |
| 30 | 1 | 0.0117 |
| | 2 | 0.0234 |
| | 3 | 0.0351 |
| | 4 | 0.0469 |

Table 14: **Performance across sampling budget (GPT-5):** Variation of the performance of GPT-5 with sampling budget for cascade length of 20 for medium reasoning effort and 65536 max completion tokens.

| Sample Budget | First Code Block | Last Code Block |
|---|---|---|
| 1 | 0.1543 | 0.1543 |
| 2 | 0.2416 | 0.2416 |
| 3 | 0.2974 | 0.2974 |
| 4 | 0.3377 | 0.3377 |
| 5 | 0.3694 | 0.3694 |
| 6 | 0.3956 | 0.3956 |
| 7 | 0.418 | 0.418 |
| 8 | 0.4375 | 0.4375 |

Table 15: **Performance across reasoning efforts and max completion tokens (GPT-5):** Variation of the performance of GPT-5 with reasoning effort and max completion tokens for cascade length of 10, and sampling budget of 1.

| Max Completion Tokens | Reasoning Effort | Avg Tokens Used | Pass@1 | Null Responses (%) | Non null Correct Responses (%) | Non null Incorrect Responses (%) |
|---|---|---|---|---|---|---|
| 512 | minimal | 194 | 0 | 0 | 0 | 100 |
| 1024 | low | 1024 | 0 | 100 | 0 | 0 |
| 2048 | low | 2046 | 0.0156 | 98.4 | 1.56 | 0 |
| 4096 | low | 3909 | 0.1719 | 72 | 17.19 | 10.81 |
| 4096 | medium | 4096 | 0 | 100 | 0 | 0 |
| 8192 | low | 5041 | 0.6094 | 3.1 | 60.94 | 35.96 |
| 8192 | medium | 7726 | 0.3906 | 61 | 39 | 0 |
| 16384 | medium | 9738 | 0.8125 | 4.7 | 81.25 | 14.05 |
| 16384 | high | 13358 | 0.6875 | 23 | 68.75 | 8.2 |
| 32768 | high | 14045 | 0.9219 | 0 | 92.19 | 7.81 |
| 65536 | high | 14549 | 0.9375 | 0 | 93.75 | 6.25 |

Table 16: **Performance across sampling budget (gpt-oss-120b):** gpt-oss-120b performance with sampling budget for max sequence length of 16384 on the 64 PBEBench instances with ground-truth cascade length 10. The nulls column counts cases (out of $Sampling\ Budget \times 32$) where the chain of thought fails to terminate within the max sequence length.

| Sampling Budget | First Code Block | | | Last Code Block | | | Nulls |
|---|---|---|---|---|---|---|---|
| | Pass@1 | Edit Sim | Valid Rate | Pass@1 | Edit Sim | Valid Rate | |
| 1 | 0.25 | 0.7084 | 0.9776 | 0.25 | 0.7084 | 0.9776 | 1 |
| 2 | 0.5 | 0.7931 | 0.9752 | 0.5 | 0.7931 | 0.9752 | 1 |
| 4 | 0.6094 | 0.9081 | 0.983 | 0.6094 | 0.9081 | 0.983 | 5 |
| 8 | 0.7344 | 0.9477 | 0.9588 | 0.7344 | 0.9477 | 0.9588 | 4 |
| 12 | 0.7812 | 0.9619 | 0.9684 | 0.7812 | 0.9619 | 0.9684 | 10 |
| 16 | 0.8281 | 0.9679 | 0.9706 | 0.8281 | 0.9679 | 0.9706 | 11 |
| 20 | 0.8594 | 0.9691 | 0.9788 | 0.8594 | 0.9691 | 0.9788 | 26 |
| 24 | 0.7969 | 0.9725 | 0.9638 | 0.7969 | 0.9725 | 0.9638 | 3 |
| 28 | 0.8438 | 0.9644 | 0.9797 | 0.8438 | 0.9644 | 0.9797 | 12 |
| 32 | 0.8438 | 0.9739 | 0.9806 | 0.8438 | 0.9739 | 0.9806 | 20 |
| 64 | 0.9062 | 0.9808 | 0.9855 | 0.9062 | 0.9808 | 0.9855 | 27 |

and counter-feeding (62%) showing that even when the models can solve cases with ground truth cascades having these relation types they are highly biased against generating them. Interetingly we

Table 17: **Performance variance per run (gpt-oss-120b):** We analyze the variance exhibited by gpt-oss-120b across all the metrics for a given run. We conduct this experiment on the 64 instances corresponding to cascade length 10 which is also used for all the scaling experiments and use a sampling budget of 32 and max sequence length of 16384, the same parameters used for evaluation of gpt-oss-120b on PBEBench. The nulls column counts cases (out of $Sampling\ Budget \times 32$) where the chain of thought fails to terminate within the max sequence length.

| Sampling Budget | First Code Block | | | Last Code Block | | | Nulls |
|---|---|---|---|---|---|---|---|
| | Pass@1 | Edit Sim | Valid Rate | Pass@1 | Edit Sim | Valid Rate | |
| 32 | 0.8438 | 0.9739 | 0.9806 | 0.8438 | 0.9739 | 0.9806 | 20 |
| 32 | 0.8594 | 0.9716 | 0.9814 | 0.8594 | 0.9716 | 0.9814 | 16 |
| 32 | 0.8281 | 0.9772 | 0.9745 | 0.8281 | 0.9772 | 0.9745 | 32 |

Table 18: **Performance across max sequence length (gpt-oss-120b):** gpt-oss-120b performance with max sequence length at sampling budget 32 on the 64 PBEBench instances with ground-truth cascade length 10. The null column counts cases (out of $64 \times 32 = 2048$) where the chain of thought fails to terminate within the sequence limit.

| Max Seq Length | First Code Block | | | Last Code Block | | | Nulls |
|---|---|---|---|---|---|---|---|
| | Pass@1 | Edit Sim | Valid Rate | Pass@1 | Edit Sim | Valid Rate | |
| 2048 | 0 | 0.0141 | 1 | 0 | 0.0141 | 1 | 2043 |
| 3840 | 0.2812 | 0.5479 | 0.9717 | 0.2727 | 0.4099 | 1 | 1799 |
| 5632 | 0.625 | 0.8447 | 0.9724 | 0.625 | 0.8447 | 0.9724 | 1212 |
| 7424 | 0.7656 | 0.9473 | 0.9705 | 0.7656 | 0.9473 | 0.9705 | 607 |
| 9216 | 0.8594 | 0.9631 | 0.9875 | 0.8594 | 0.9631 | 0.9875 | 305 |
| 12800 | 0.875 | 0.9706 | 0.9788 | 0.875 | 0.9706 | 0.9788 | 80 |
| 14592 | 0.875 | 0.9748 | 0.9825 | 0.875 | 0.9748 | 0.9825 | 28 |
| 16384 | 0.8438 | 0.9739 | 0.9806 | 0.8438 | 0.9739 | 0.9806 | 20 |
| 32768 | 0.9219 | 0.9819 | 0.9838 | 0.9219 | 0.9819 | 0.9838 | 0 |
| 65536 | 0.9062 | 0.9755 | 0.9838 | 0.9062 | 0.9755 | 0.9838 | 0 |

also see low false positives for the success/passing cases which is consistent with the tendencies of these models to try and not predict cascades that incorporate BFCC relations and the false positive rate is highest for feeding the hardest relation type. Finally for failure cases we note that there is a high false negative rate for every relation type ranging between 75% to 80%, consistent with the fact that the evaluated models try to avoid predicting BFCC relations and perhaps for cases where they are needed to find a correct solution, they fail to find one.

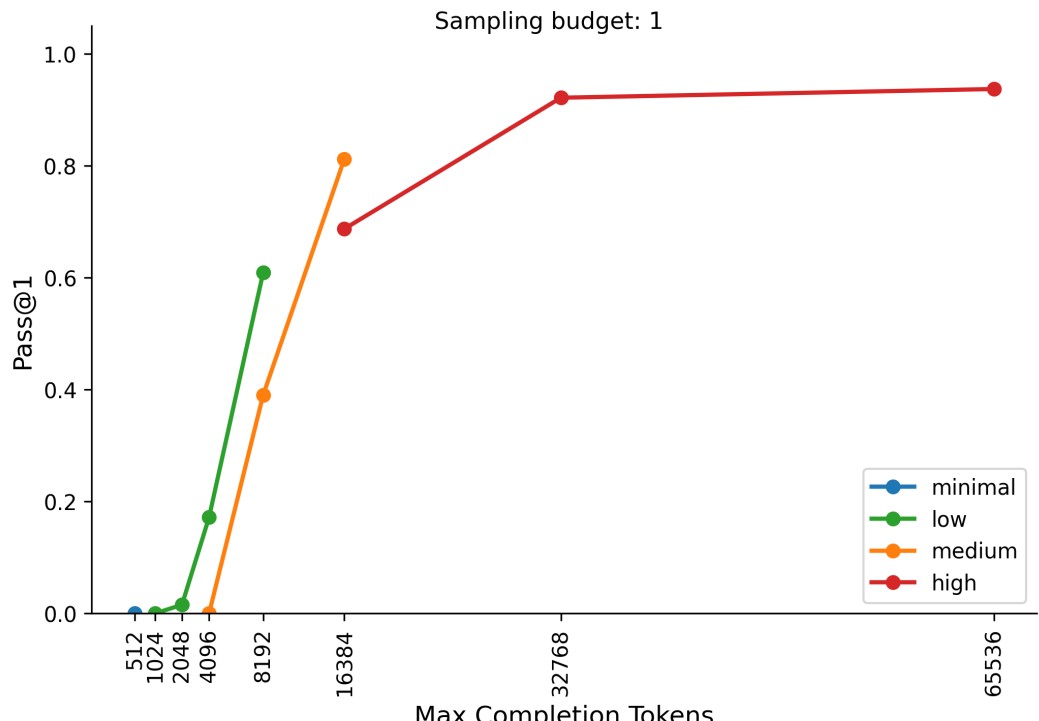

Figure 18: **Pass@1 vs Max Completion Tokens (sampling budget: 1):** Variation of Pass@1 vs Max Completion Tokens for various reasoning modes ranging from minimal, low, medium, to high.

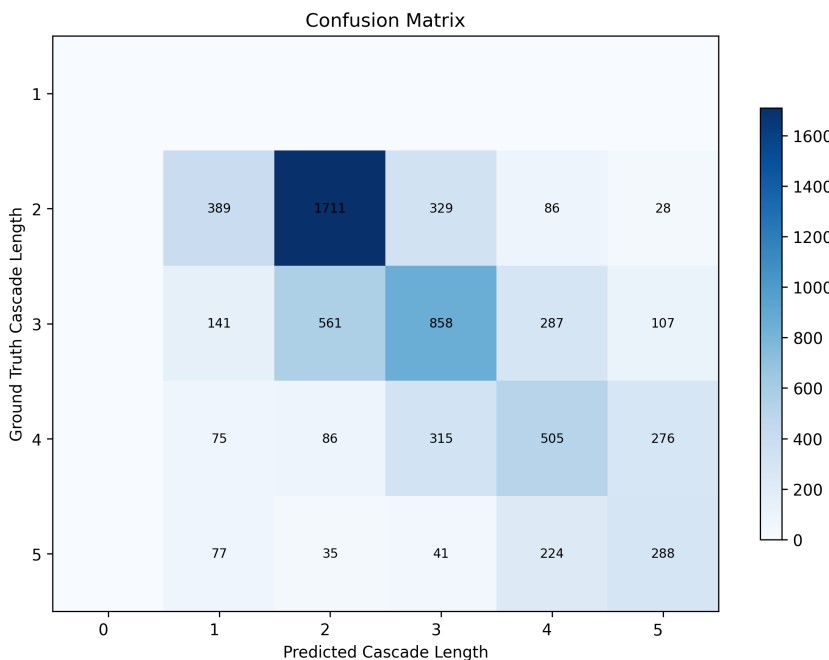

Figure 19: **Cascade Length Confusion Matrix for Success (All Models PBEBench-Lite)**: Confusion matrix showing the distribution of cascade lengths in the model prediction vs ground truth. 0 length on the predicted side corresponds to cases where the model fails to generate a valid cascade at all. The results are averaged across all models on PBEBench-Lite, and failure denotes Pass@1 = 0 with a sampling budget of 1.

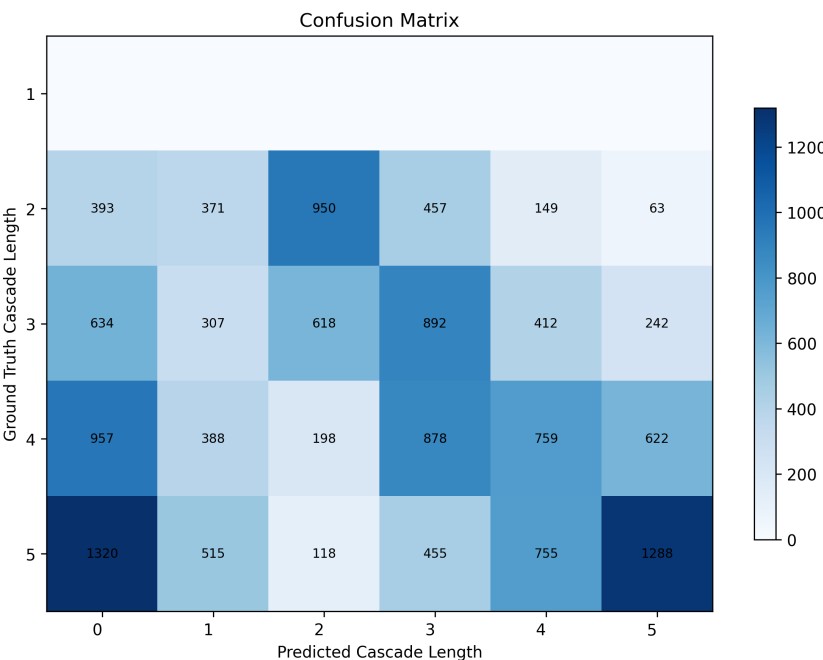

Figure 20: **Cascade Length Confusion Matrix for Failure (All Models PBEBench-Lite)**: Confusion matrix showing the distribution of cascade lengths in the model prediction vs ground truth. 0 length on the predicted side corresponds to cases where the model fails to generate a valid cascade at all. The results are averaged across all models on PBEBench-Lite, and failure denotes Pass@1 = 0 with a sampling budget of 1.

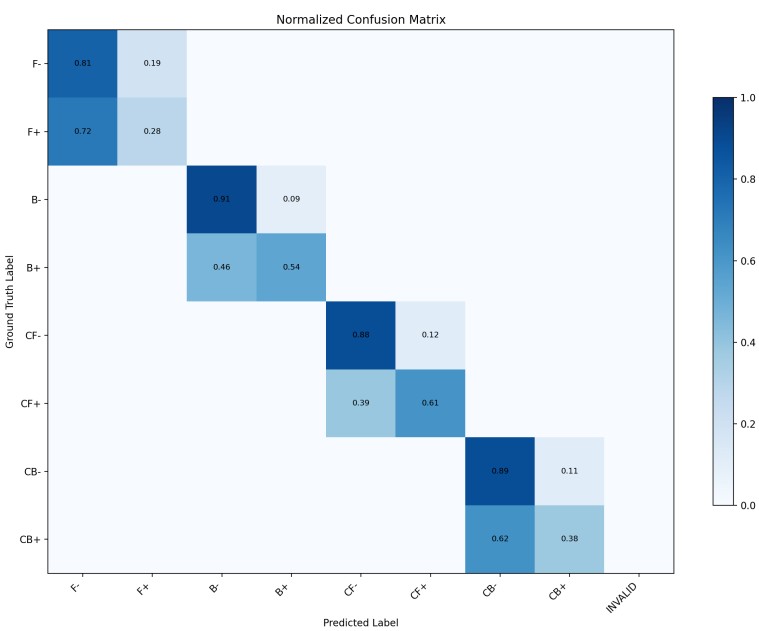

Figure 21: **Simplified Relation Type Confusion Matrix for Success (All Models PBEBench-Lite)**: Confusion matrix showing the distribution of relation types in the model prediction vs ground truth. INVALID category indicates cases where the model fails to generate a valid cascade at all. The results are averaged across all models on PBEBench-Lite, and success denotes Pass@1 = 1 with a sampling budget of 1.

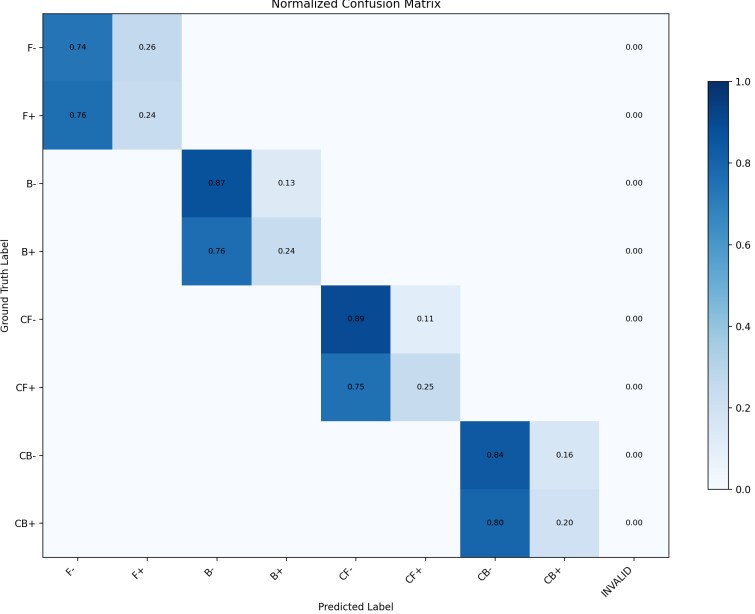

Figure 22: **Simplified Relation Type Confusion Matrix for Failure (All Models PBEBench-Lite)**: Confusion matrix showing the distribution of relation types in the model prediction vs ground truth. INVALID category indicates cases where the model fails to generate a valid cascade at all. The results are averaged across all models on PBEBench-Lite, and failure denotes Pass@1 = 0 with a sampling budget of 1.

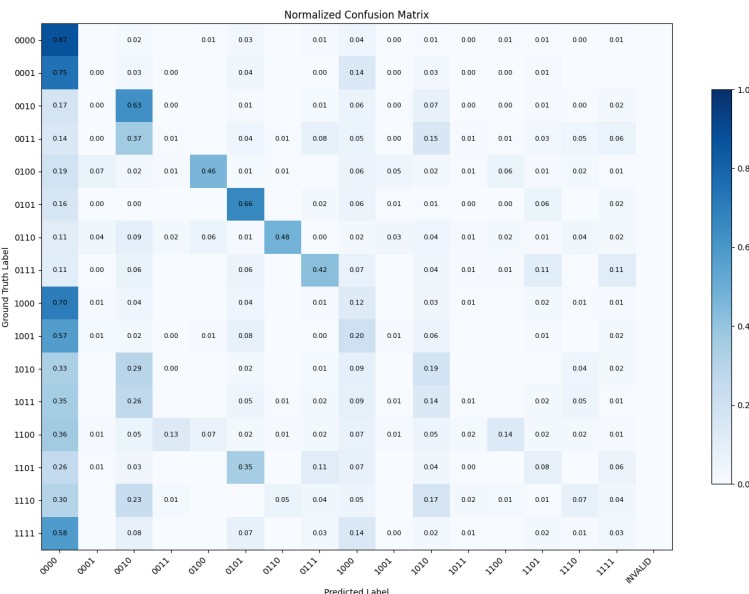

Figure 23: **Relation Type Confusion Matrix for Success (All Models PBEBench-Lite)**: Confusion matrix showing the distribution of relation types in the model prediction vs ground truth. INVALID category indicates cases where the model fails to generate a valid cascade at all. The results are averaged across all models on PBEBench-Lite, and success denotes Pass@1 = 1 with a sampling budget of 1.

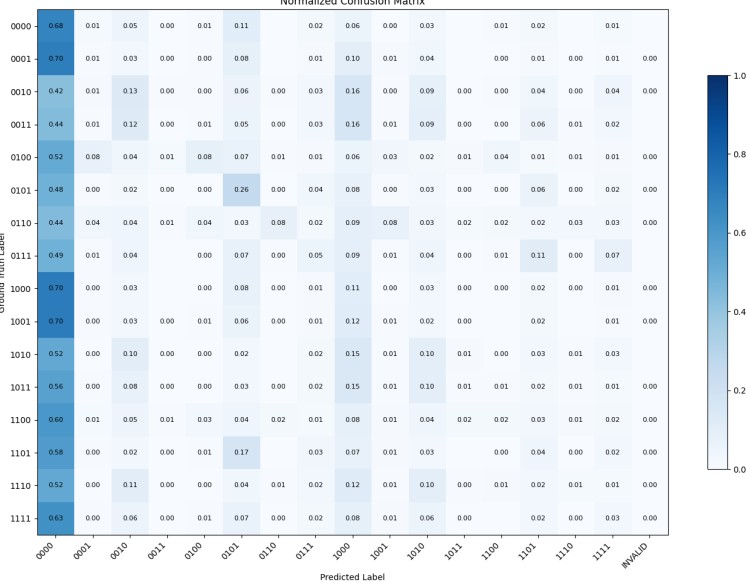

Figure 24: **Relation Type Confusion Matrix for Failure (All Models PBEBench-Lite)**: Confusion matrix showing the distribution of relation types in the model prediction vs ground truth. INVALID category indicates cases where the model fails to generate a valid cascade at all. The results are averaged across all models on PBEBench-Lite, and failure denotes Pass@1 = 0 with a sampling budget of 1.

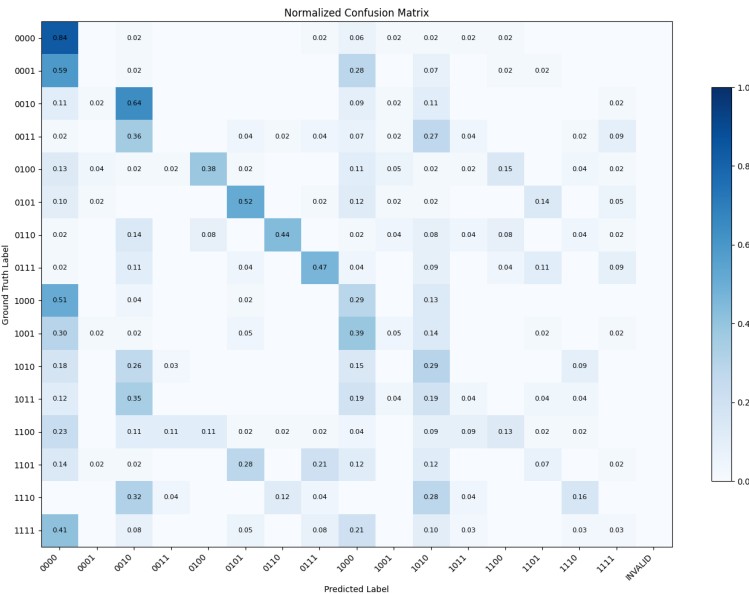

Figure 25: **Relation Type Confusion Matrix for Success (GPT-5 PBEBench-Lite)**: Confusion matrix showing the distribution of relation types in the model prediction vs ground truth. INVALID category indicates cases where the model fails to generate a valid cascade at all. The results are for GPT-5 on PBEBench-Lite, and success denotes Pass@1 = 1 with a sampling budget of 1.

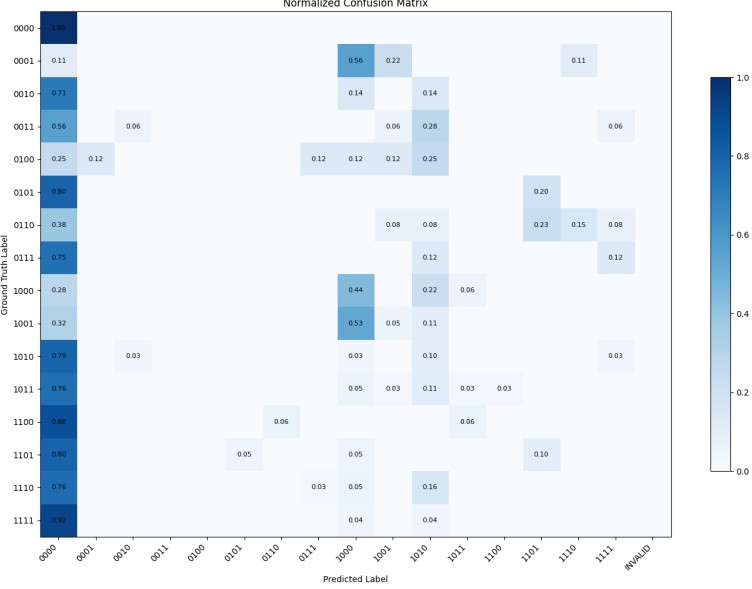

Figure 26: **Relation Type Confusion Matrix for Failure (GPT-5 PBEBench-Lite)**: Confusion matrix showing the distribution of relation types in the model prediction vs ground truth. INVALID category indicates cases where the model fails to generate a valid cascade at all. The results are for GPT-5 on PBEBench-Lite, and failure denotes Pass@1 = 0 with a sampling budget of 1.

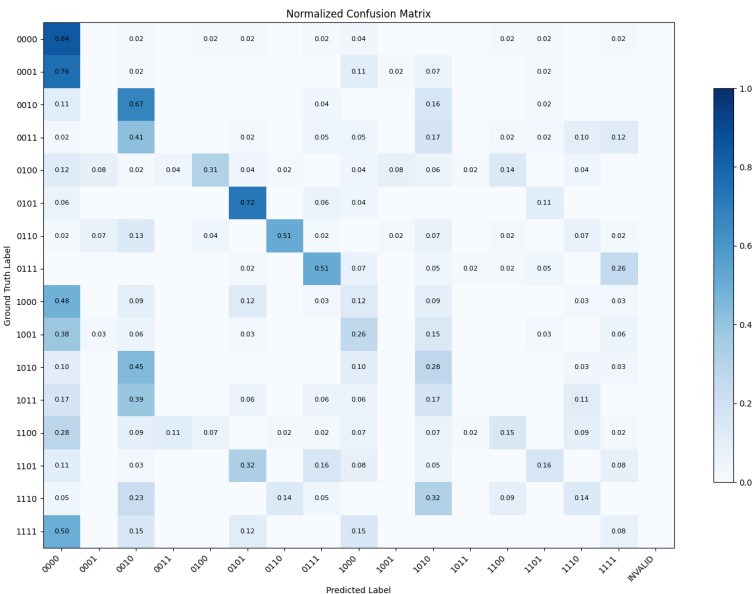

Figure 27: **Relation Type Confusion Matrix for Success (gpt-oss-120b PBEBench-Lite)**: Confusion matrix showing the distribution of relation types in the model prediction vs ground truth. INVALID category indicates cases where the model fails to generate a valid cascade at all. The results are for gpt-oss-120b on PBEBench-Lite, and success denotes Pass@1 = 1 with a sampling budget of 1.

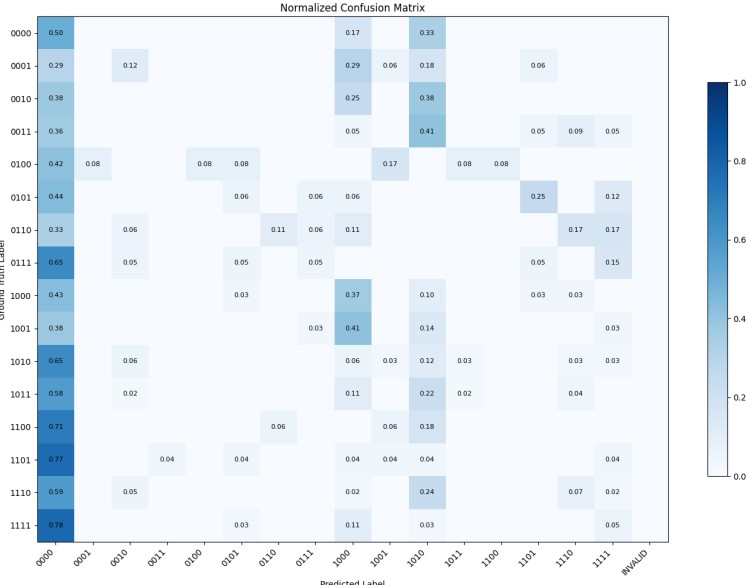

Figure 28: **Relation Type Confusion Matrix for Failure (gpt-oss-120b PBEBench-Lite)**: Confusion matrix showing the distribution of relation types in the model prediction vs ground truth. INVALID category indicates cases where the model fails to generate a valid cascade at all. The results are for gpt-oss-120b on PBEBench-Lite, and failure denotes Pass@1 = 0 with a sampling budget of 1.

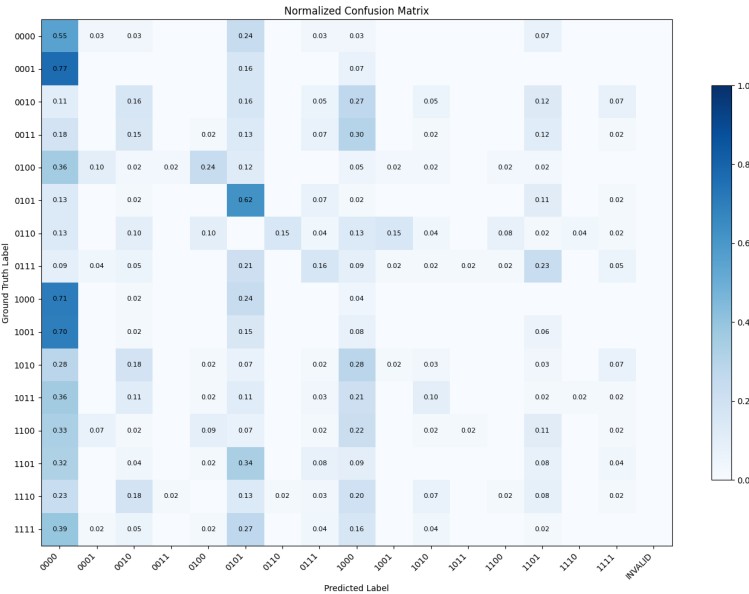

Figure 29: **Relation Type Confusion Matrix for Failure (Claude-3.5-Sonnet PBEBench-Lite)**: Confusion matrix showing the distribution of relation types in the model prediction vs ground truth. INVALID category indicates cases where the model fails to generate a valid cascade at all. The results are for Claude-3.5-Sonnet on PBEBench-Lite, and failure denotes Pass@1 = 0 with a sampling budget of 1.

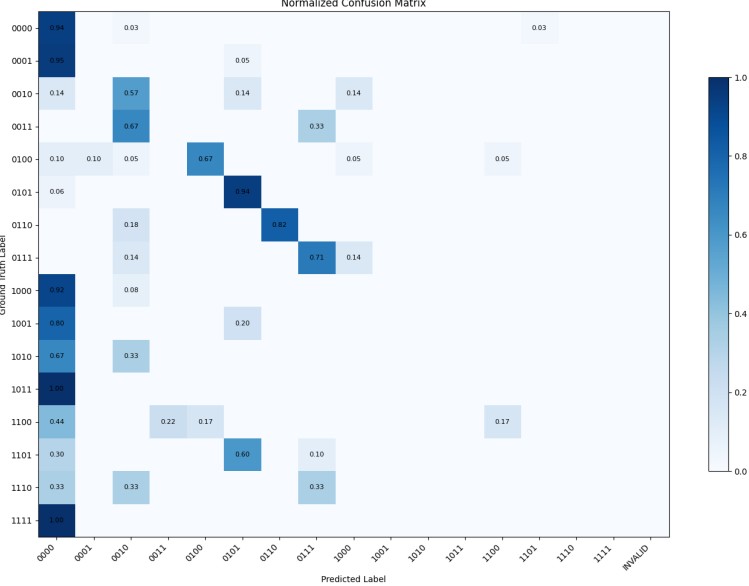

Figure 30: **Relation Type Confusion Matrix for Success (Claude-3.5-Sonnet PBEBench-Lite)**: Confusion matrix showing the distribution of relation types in the model prediction vs ground truth. INVALID category indicates cases where the model fails to generate a valid cascade at all. The results are for Claude-3.5-Sonnet on PBEBench-Lite, and success denotes Pass@1 = 1 with a sampling budget of 1.

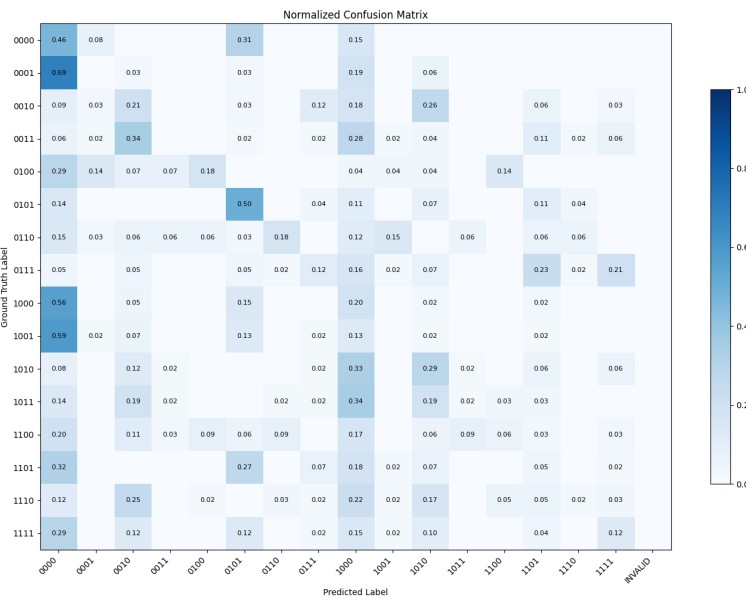

Figure 31: **Relation Type Confusion Matrix for Failure (Claude-3.7-Sonnet (Thinking) PBEBench-Lite)**: Confusion matrix showing the distribution of relation types in the model prediction vs ground truth. INVALID category indicates cases where the model fails to generate a valid cascade at all. The results are for Claude-3.7-Sonnet (Thinking) on PBEBench-Lite, and failure denotes Pass@1 = 0 with a sampling budget of 1.

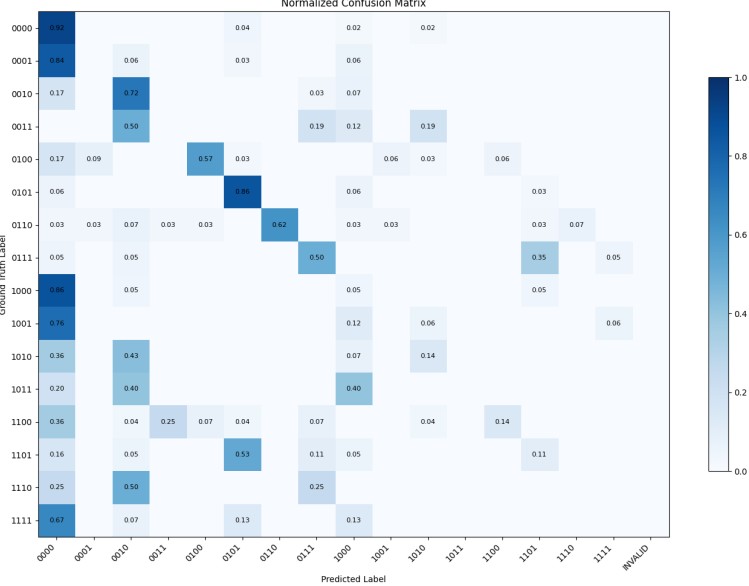

Figure 32: **Relation Type Confusion Matrix for Success (Claude-3.7-Sonnet (Thinking) PBEBench-Lite)**: Confusion matrix showing the distribution of relation types in the model prediction vs ground truth. INVALID category indicates cases where the model fails to generate a valid cascade at all. The results are for Claude-3.7-Sonnet (Thinking) on PBEBench-Lite, and success denotes Pass@1 = 1 with a sampling budget of 1.

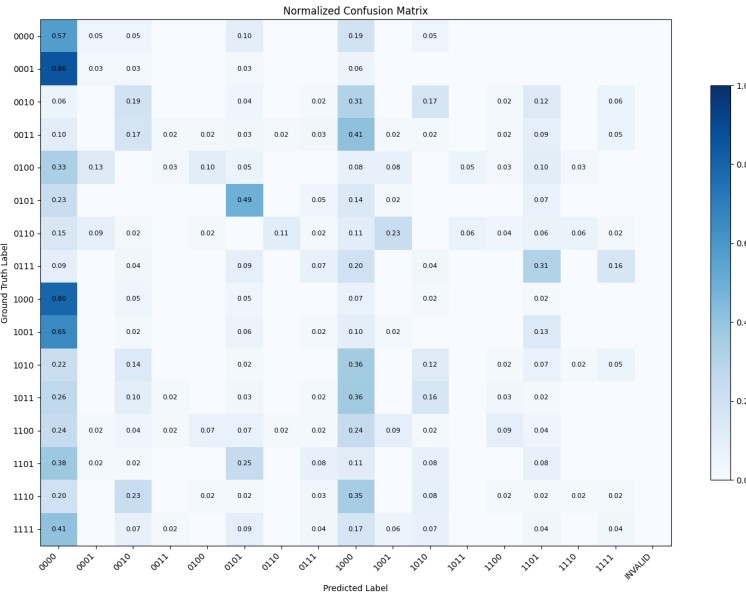

Figure 33: **Relation Type Confusion Matrix for Failure (Claude-3.7-Sonnet PBEBench-Lite)**: Confusion matrix showing the distribution of relation types in the model prediction vs ground truth. INVALID category indicates cases where the model fails to generate a valid cascade at all. The results are for Claude-3.7-Sonnet on PBEBench-Lite, and failure denotes Pass@1 = 0 with a sampling budget of 1.

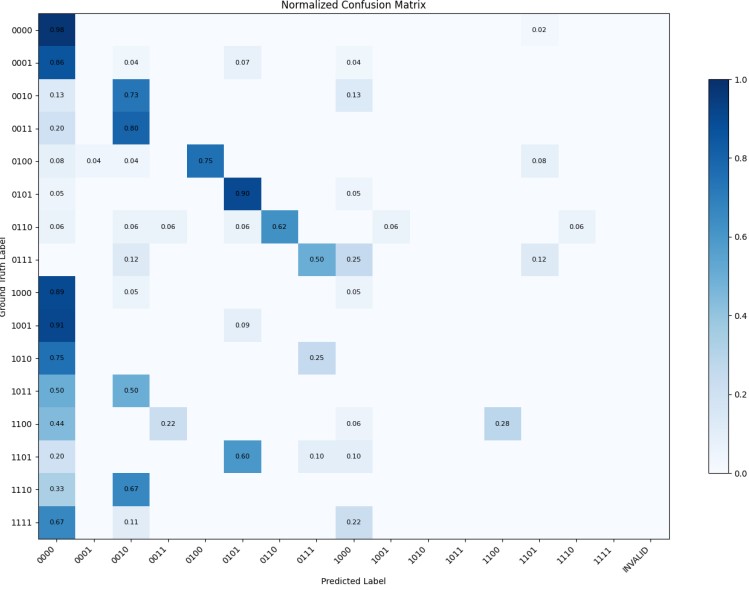

Figure 34: **Relation Type Confusion Matrix for Success (Claude-3.7-Sonnet PBEBench-Lite)**: Confusion matrix showing the distribution of relation types in the model prediction vs ground truth. INVALID category indicates cases where the model fails to generate a valid cascade at all. The results are for Claude-3.7-Sonnet on PBEBench-Lite, and success denotes Pass@1 = 1 with a sampling budget of 1.

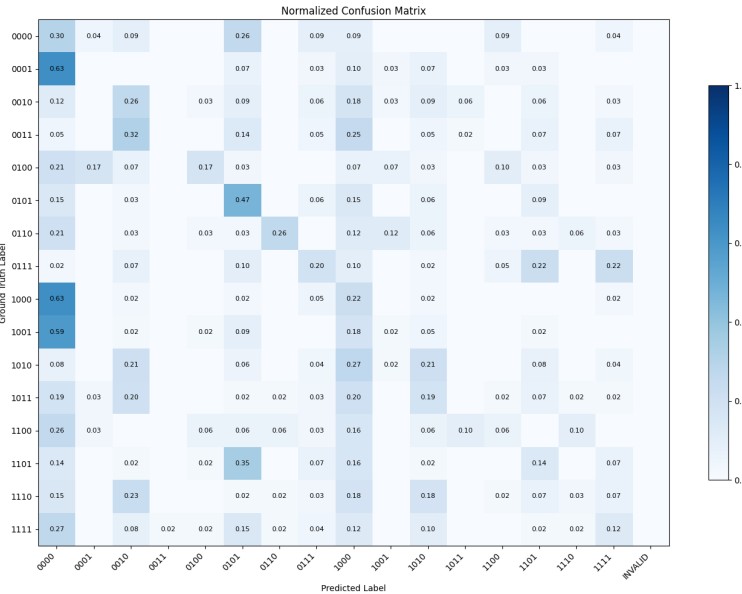

Figure 35: **Relation Type Confusion Matrix for Failure (Claude-4 Sonnet (Thinking) PBEBench-Lite)**: Confusion matrix showing the distribution of relation types in the model prediction vs ground truth. INVALID category indicates cases where the model fails to generate a valid cascade at all. The results are for Claude-4 Sonnet (Thinking) on PBEBench-Lite, and failure denotes Pass@1 = 0 with a sampling budget of 1.

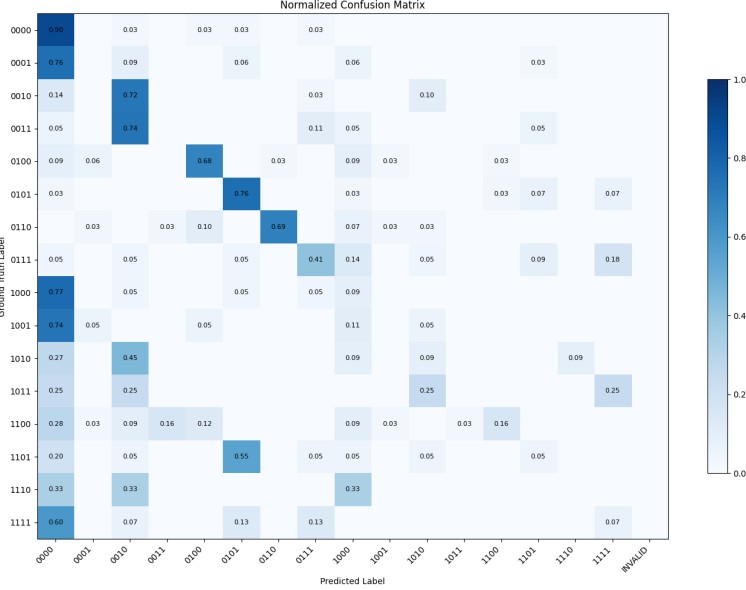

Figure 36: **Relation Type Confusion Matrix for Success (Claude-4 Sonnet (Thinking) PBEBench-Lite)**: Confusion matrix showing the distribution of relation types in the model prediction vs ground truth. INVALID category indicates cases where the model fails to generate a valid cascade at all. The results are for Claude-4 Sonnet (Thinking) on PBEBench-Lite, and success denotes Pass@1 = 1 with a sampling budget of 1.

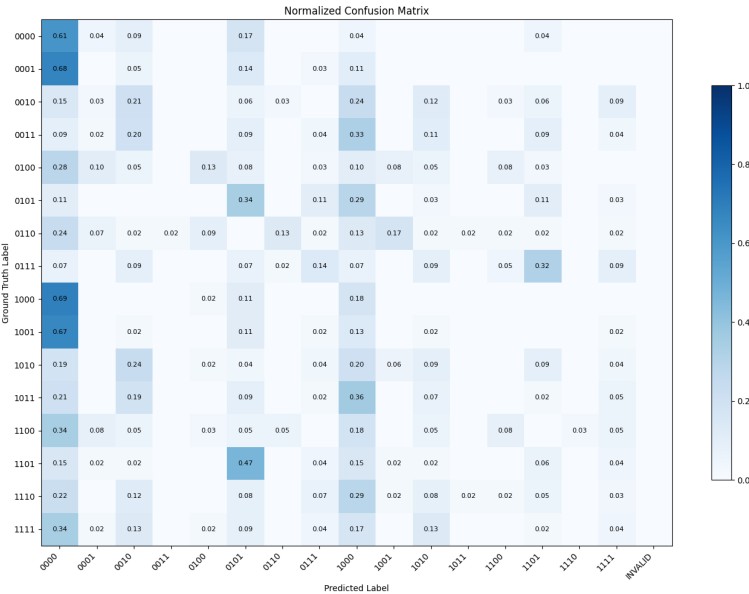

Figure 37: **Relation Type Confusion Matrix for Failure (Claude-4 Sonnet PBEBench-Lite)**: Confusion matrix showing the distribution of relation types in the model prediction vs ground truth. INVALID category indicates cases where the model fails to generate a valid cascade at all. The results are for Claude-4 Sonnet on PBEBench-Lite, and failure denotes Pass@1 = 0 with a sampling budget of 1.

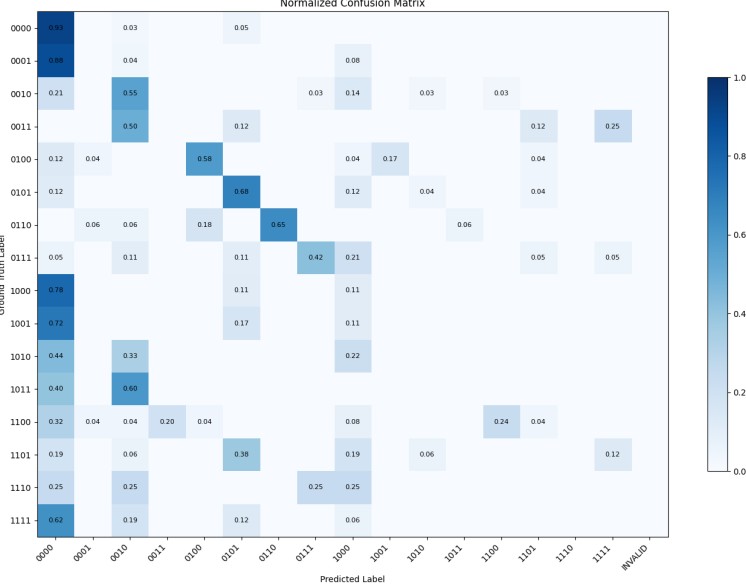

Figure 38: **Relation Type Confusion Matrix for Success (Claude-4 Sonnet PBEBench-Lite)**: Confusion matrix showing the distribution of relation types in the model prediction vs ground truth. INVALID category indicates cases where the model fails to generate a valid cascade at all. The results are for Claude-4 Sonnet on PBEBench-Lite, and success denotes Pass@1 = 1 with a sampling budget of 1.

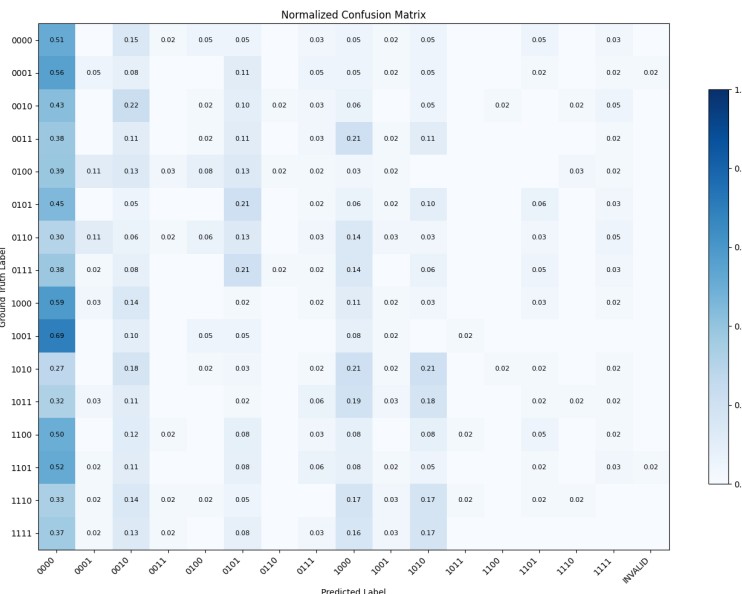

Figure 39: **Relation Type Confusion Matrix for Failure (Codestral-22B PBEBench-Lite)**: Confusion matrix showing the distribution of relation types in the model prediction vs ground truth. INVALID category indicates cases where the model fails to generate a valid cascade at all. The results are for Codestral-22B on PBEBench-Lite, and failure denotes Pass@1 = 0 with a sampling budget of 1.

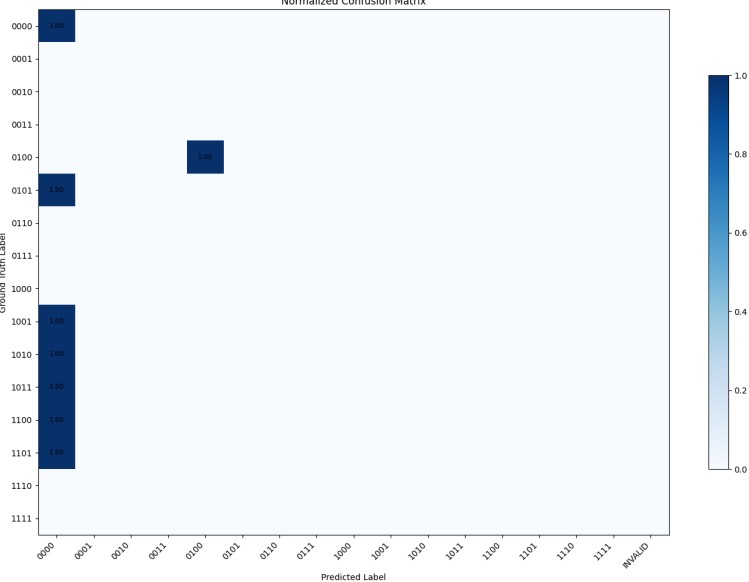

Figure 40: **Relation Type Confusion Matrix for Success (Codestral-22B PBEBench-Lite)**: Confusion matrix showing the distribution of relation types in the model prediction vs ground truth. INVALID category indicates cases where the model fails to generate a valid cascade at all. The results are for Codestral-22B on PBEBench-Lite, and success denotes Pass@1 = 1 with a sampling budget of 1.

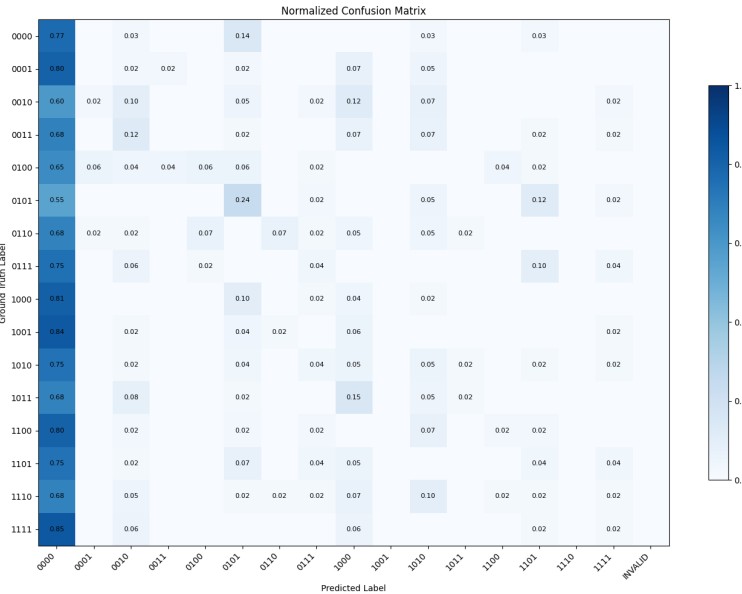

Figure 41: **Relation Type Confusion Matrix for Failure (DeepSeek-R1-Distill-Qwen-32B PBEBench-Lite**): Confusion matrix showing the distribution of relation types in the model prediction vs ground truth. INVALID category indicates cases where the model fails to generate a valid cascade at all. The results are for DeepSeek-R1-Distill-Qwen-32B on PBEBench-Lite, and failure denotes Pass@1 = 0 with a sampling budget of 1.

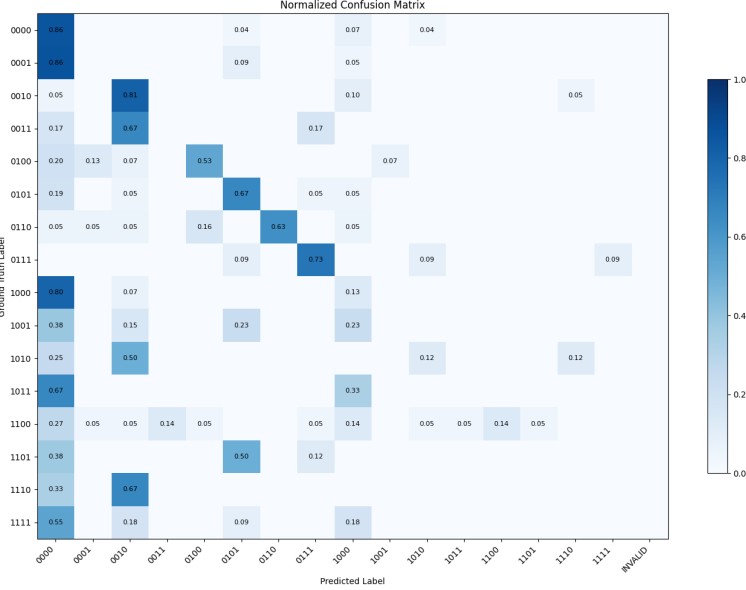

Figure 42: **Relation Type Confusion Matrix for Success (DeepSeek-R1-Distill-Qwen-32B PBEBench-Lite**): Confusion matrix showing the distribution of relation types in the model prediction vs ground truth. INVALID category indicates cases where the model fails to generate a valid cascade at all. The results are for DeepSeek-R1-Distill-Qwen-32B on PBEBench-Lite, and success denotes Pass@1 = 1 with a sampling budget of 1.

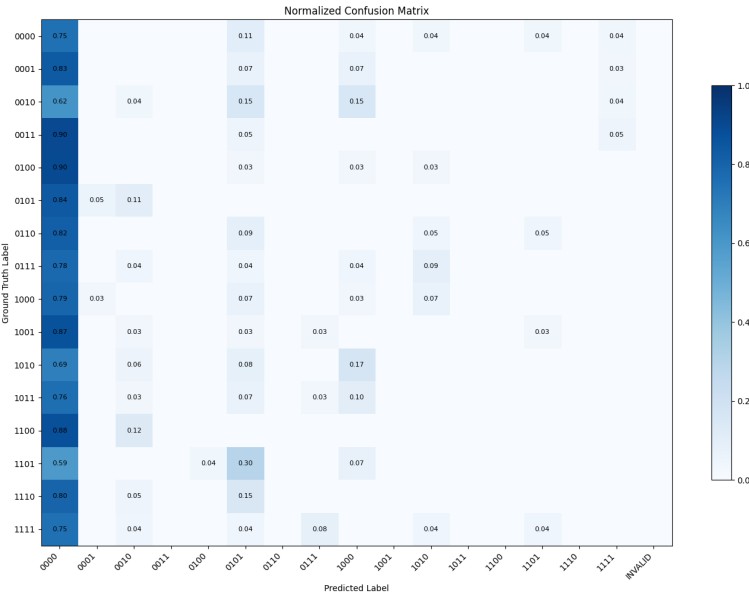

Figure 43: **Relation Type Confusion Matrix for Failure (Gemini 2.5 Flash PBEBench-Lite)**: Confusion matrix showing the distribution of relation types in the model prediction vs ground truth. INVALID category indicates cases where the model fails to generate a valid cascade at all. The results are for Gemini 2.5 Flash on PBEBench-Lite, and failure denotes Pass@1 = 0 with a sampling budget of 1.

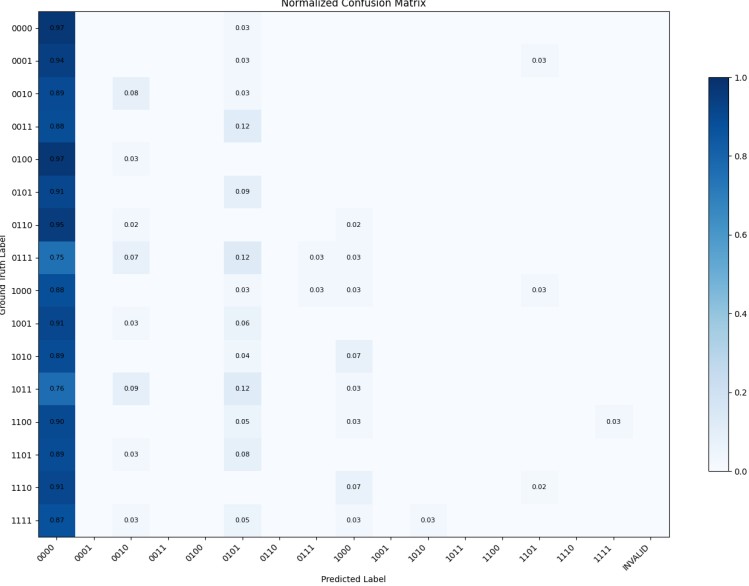

Figure 44: **Relation Type Confusion Matrix for Success (Gemini 2.5 Flash PBEBench-Lite)**: Confusion matrix showing the distribution of relation types in the model prediction vs ground truth. INVALID category indicates cases where the model fails to generate a valid cascade at all. The results are for Gemini 2.5 Flash on PBEBench-Lite, and success denotes Pass@1 = 1 with a sampling budget of 1.

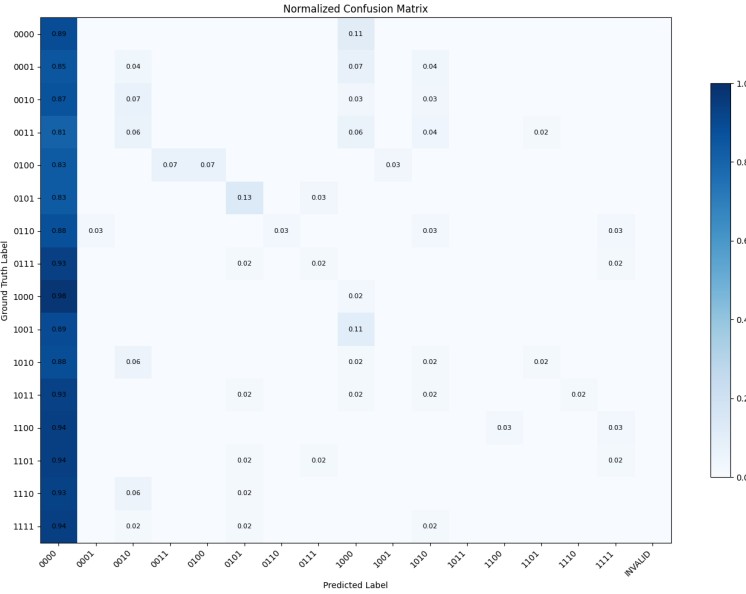

Figure 45: **Relation Type Confusion Matrix for Failure (QwQ-32B PBEBench-Lite)**: Confusion matrix showing the distribution of relation types in the model prediction vs ground truth. INVALID category indicates cases where the model fails to generate a valid cascade at all. The results are for QwQ-32B on PBEBench-Lite, and failure denotes Pass@1 = 0 with a sampling budget of 1.

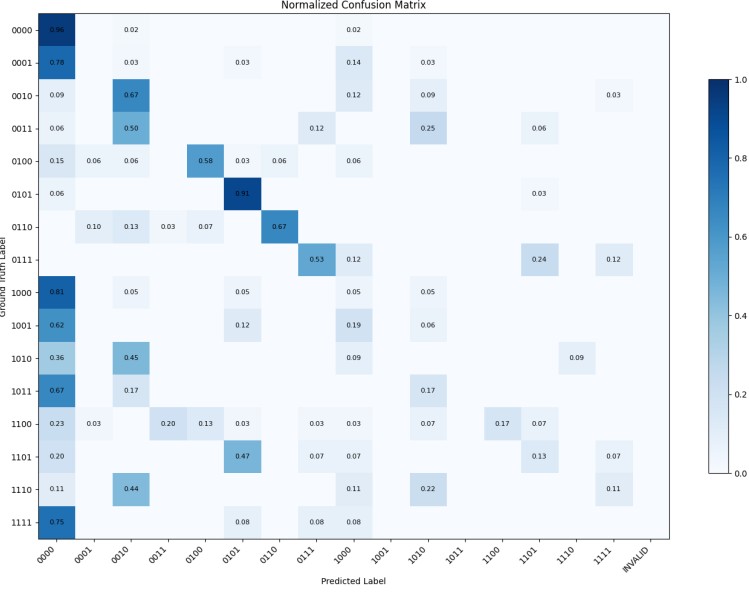

Figure 46: **Relation Type Confusion Matrix for Success (QwQ-32B PBEBench-Lite)**: Confusion matrix showing the distribution of relation types in the model prediction vs ground truth. INVALID category indicates cases where the model fails to generate a valid cascade at all. The results are for QwQ-32B on PBEBench-Lite, and success denotes Pass@1 = 1 with a sampling budget of 1.

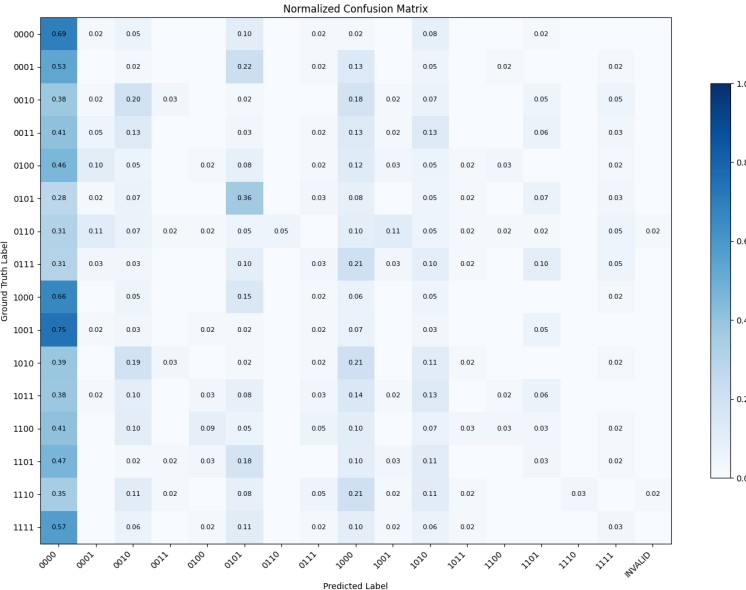

Figure 47: **Relation Type Confusion Matrix for Failure (Qwen2.5-32B-Instruct PBEBench-Lite)**: Confusion matrix showing the distribution of relation types in the model prediction vs ground truth. INVALID category indicates cases where the model fails to generate a valid cascade at all. The results are for Qwen2.5-32B-Instruct on PBEBench-Lite, and failure denotes Pass@1 = 0 with a sampling budget of 1.

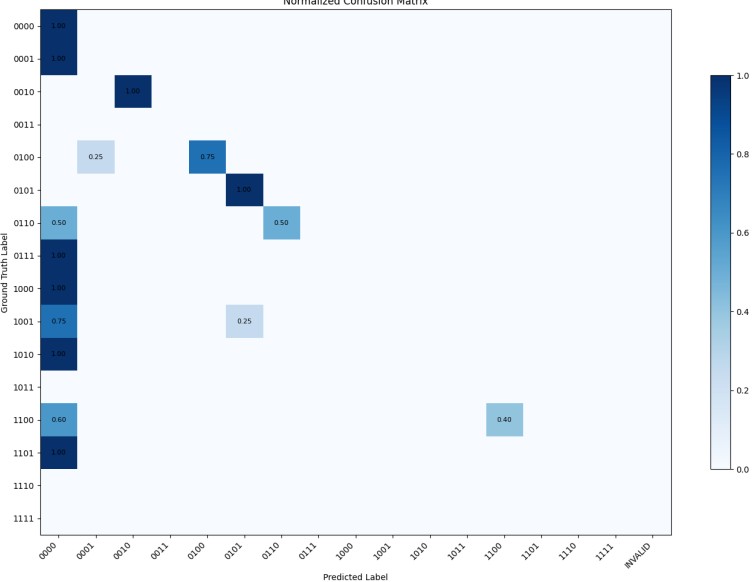

Figure 48: **Relation Type Confusion Matrix for Success (Qwen2.5-32B-Instruct PBEBench-Lite)**: Confusion matrix showing the distribution of relation types in the model prediction vs ground truth. INVALID category indicates cases where the model fails to generate a valid cascade at all. The results are for Qwen2.5-32B-Instruct on PBEBench-Lite, and success denotes Pass@1 = 1 with a sampling budget of 1.

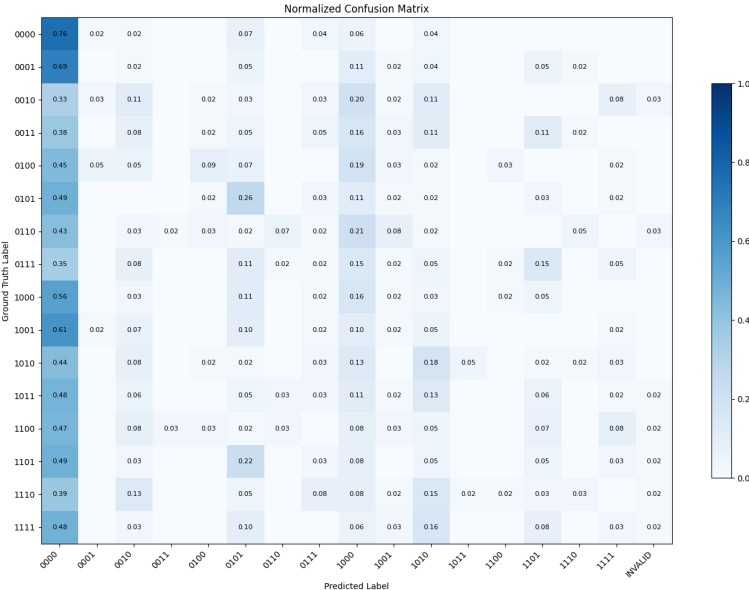

Figure 49: **Relation Type Confusion Matrix for Failure (Qwen2.5Coder-32B-Instruct PBEBench-Lite)**: Confusion matrix showing the distribution of relation types in the model prediction vs ground truth. INVALID category indicates cases where the model fails to generate a valid cascade at all. The results are for Qwen2.5Coder-32B-Instruct on PBEBench-Lite, and failure denotes Pass@1 = 0 with a sampling budget of 1.

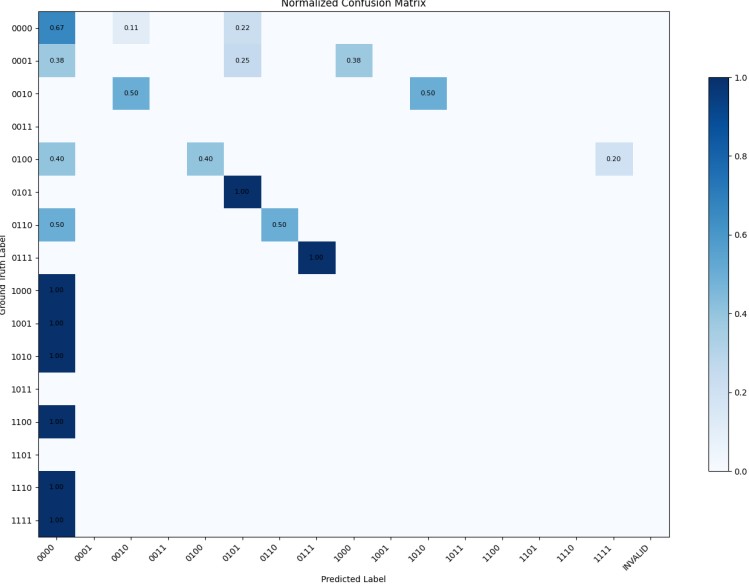

Figure 50: **Relation Type Confusion Matrix for Success (Qwen2.5Coder-32B-Instruct PBEBench-Lite)**: Confusion matrix showing the distribution of relation types in the model prediction vs ground truth. INVALID category indicates cases where the model fails to generate a valid cascade at all. The results are for Qwen2.5Coder-32B-Instruct on PBEBench-Lite, and success denotes Pass@1 = 1 with a sampling budget of 1.

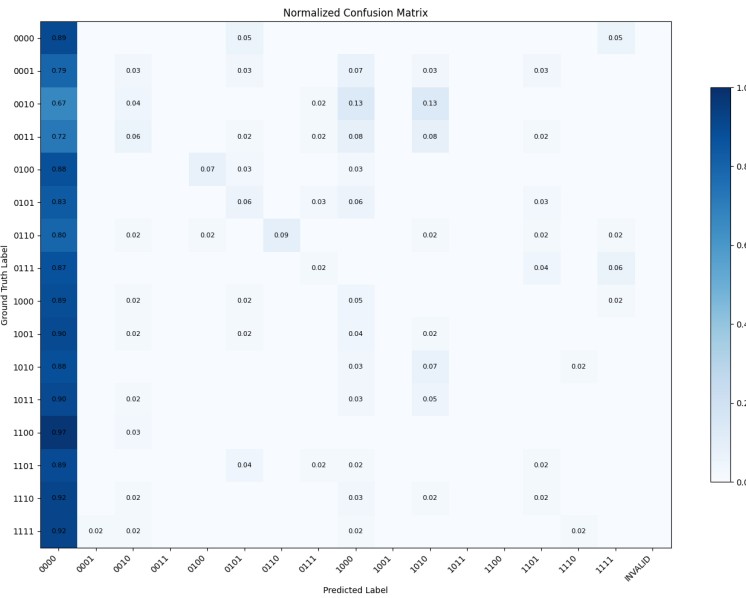

Figure 51: **Relation Type Confusion Matrix for Failure (Qwen3-30B-A3B PBEBench-Lite)**: Confusion matrix showing the distribution of relation types in the model prediction vs ground truth. INVALID category indicates cases where the model fails to generate a valid cascade at all. The results are for Qwen3-30B-A3B on PBEBench-Lite, and failure denotes Pass@1 = 0 with a sampling budget of 1.

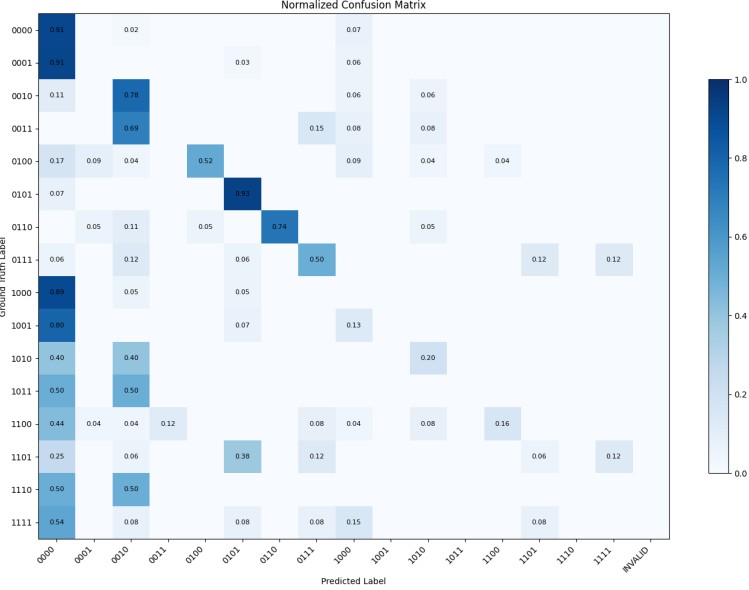

Figure 52: **Relation Type Confusion Matrix for Success (Qwen3-30B-A3B PBEBench-Lite)**: Confusion matrix showing the distribution of relation types in the model prediction vs ground truth. INVALID category indicates cases where the model fails to generate a valid cascade at all. The results are for Qwen3-30B-A3B on PBEBench-Lite, and success denotes Pass@1 = 1 with a sampling budget of 1.

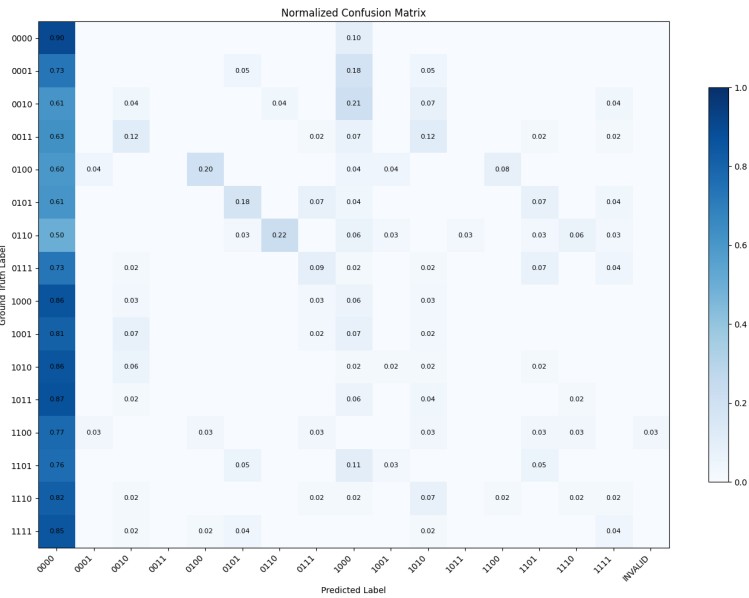

Figure 53: **Relation Type Confusion Matrix for Failure (Qwen3-32B (with CoT) PBEBench-Lite)**: Confusion matrix showing the distribution of relation types in the model prediction vs ground truth. INVALID category indicates cases where the model fails to generate a valid cascade at all. The results are for Qwen3-32B (with CoT) on PBEBench-Lite, and failure denotes Pass@1 = 0 with a sampling budget of 1.

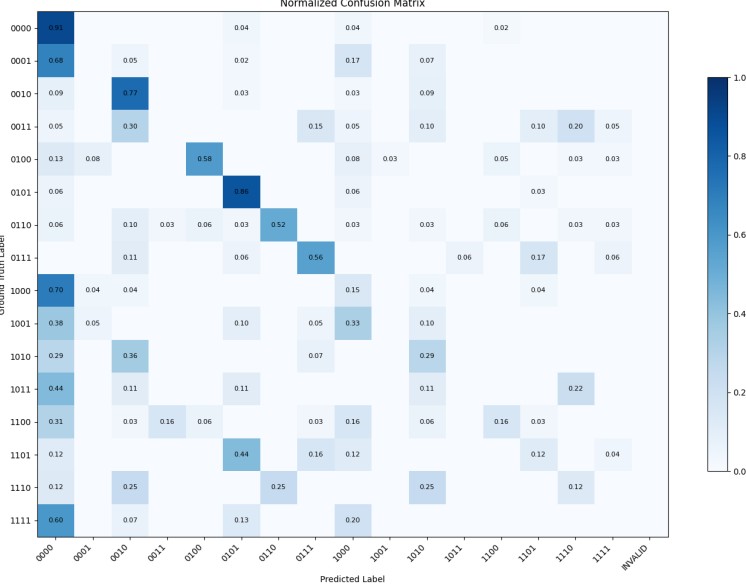

Figure 54: **Relation Type Confusion Matrix for Success (Qwen3-32B (with CoT) PBEBench-Lite)**: Confusion matrix showing the distribution of relation types in the model prediction vs ground truth. INVALID category indicates cases where the model fails to generate a valid cascade at all. The results are for Qwen3-32B (with CoT) on PBEBench-Lite, and success denotes Pass@1 = 1 with a sampling budget of 1.

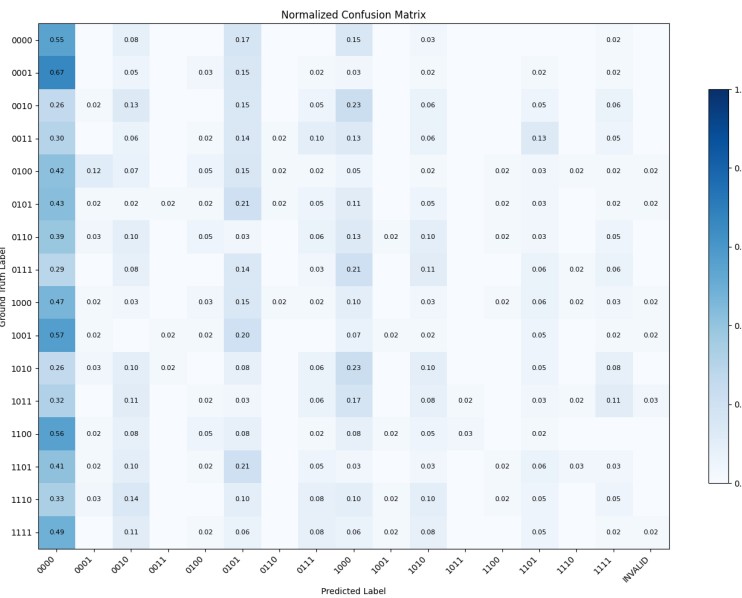

Figure 55: **Relation Type Confusion Matrix for Failure (Qwen3-32B PBEBench-Lite)**: Confusion matrix showing the distribution of relation types in the model prediction vs ground truth. INVALID category indicates cases where the model fails to generate a valid cascade at all. The results are for Qwen3-32B on PBEBench-Lite, and failure denotes Pass@1 = 0 with a sampling budget of 1.

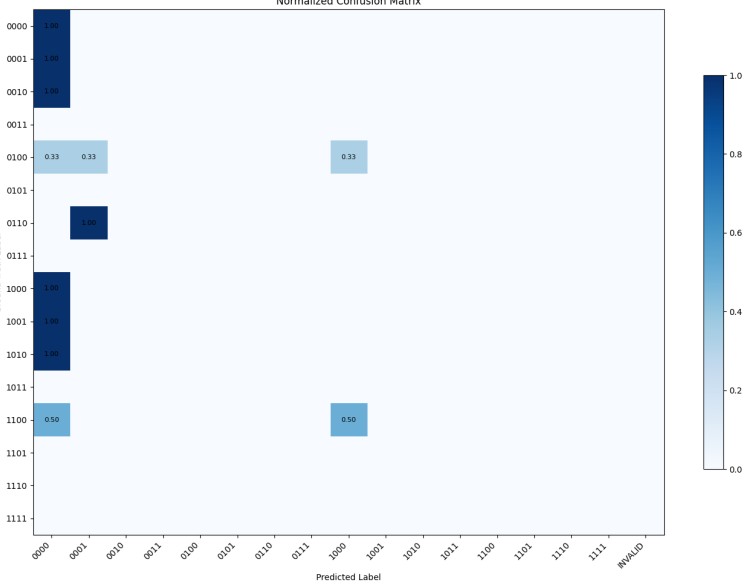

Figure 56: **Relation Type Confusion Matrix for Success (Qwen3-32B PBEBench-Lite)**: Confusion matrix showing the distribution of relation types in the model prediction vs ground truth. INVALID category indicates cases where the model fails to generate a valid cascade at all. The results are for Qwen3-32B on PBEBench-Lite, and success denotes Pass@1 = 1 with a sampling budget of 1.

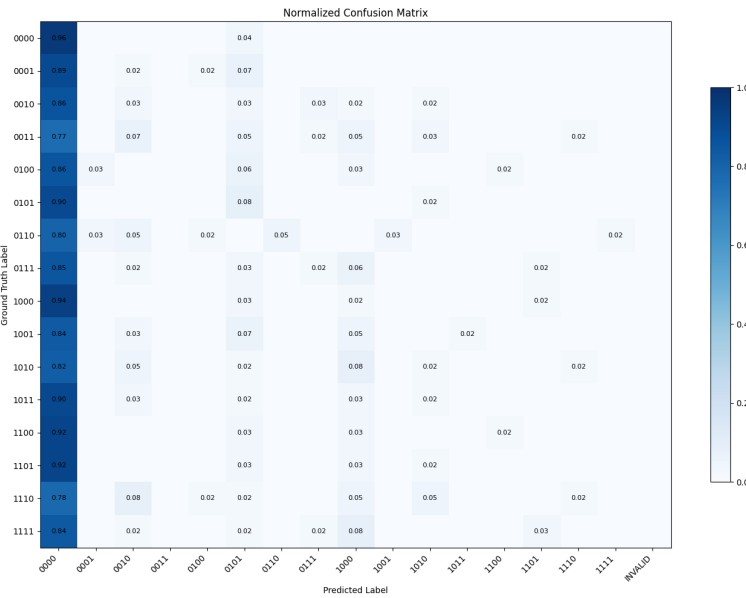

Figure 57: **Relation Type Confusion Matrix for Failure (Qwen3-Coder-30B-A3B-Instruct PBEBench-Lite)**: Confusion matrix showing the distribution of relation types in the model prediction vs ground truth. INVALID category indicates cases where the model fails to generate a valid cascade at all. The results are for Qwen3-Coder-30B-A3B-Instruct on PBEBench-Lite, and failure denotes Pass@1 = 0 with a sampling budget of 1.

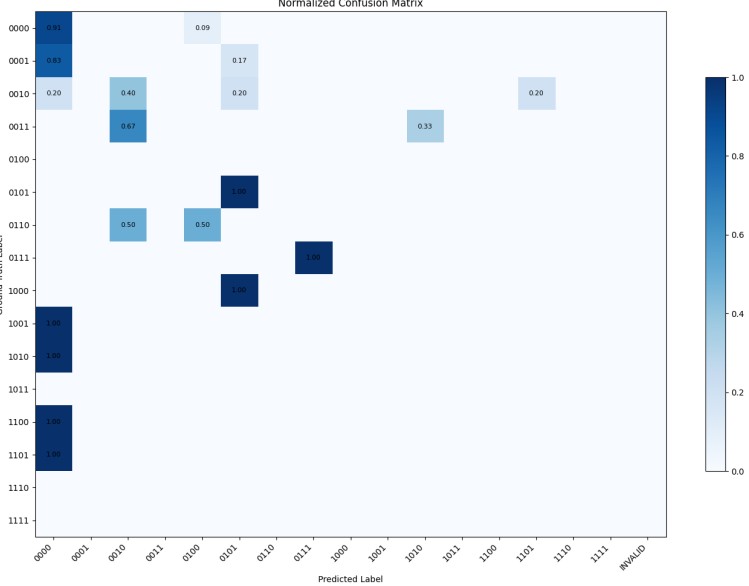

Figure 58: **Relation Type Confusion Matrix for Success (Qwen3-Coder-30B-A3B-Instruct PBEBench-Lite)**: Confusion matrix showing the distribution of relation types in the model prediction vs ground truth. INVALID category indicates cases where the model fails to generate a valid cascade at all. The results are for Qwen3-Coder-30B-A3B-Instruct on PBEBench-Lite, and success denotes Pass@1 = 1 with a sampling budget of 1.

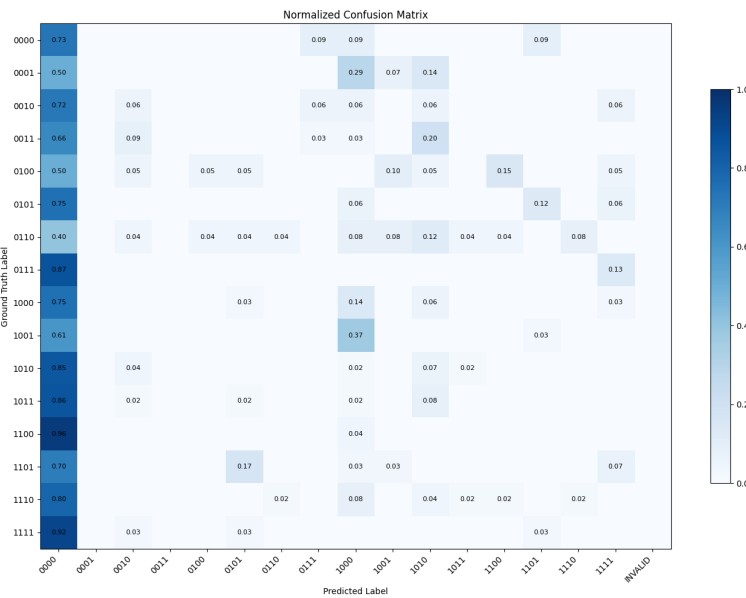

Figure 59: **Relation Type Confusion Matrix for Failure (o3-mini PBEBench-Lite)**: Confusion matrix showing the distribution of relation types in the model prediction vs ground truth. INVALID category indicates cases where the model fails to generate a valid cascade at all. The results are for o3-mini on PBEBench-Lite, and failure denotes Pass@1 = 0 with a sampling budget of 1.

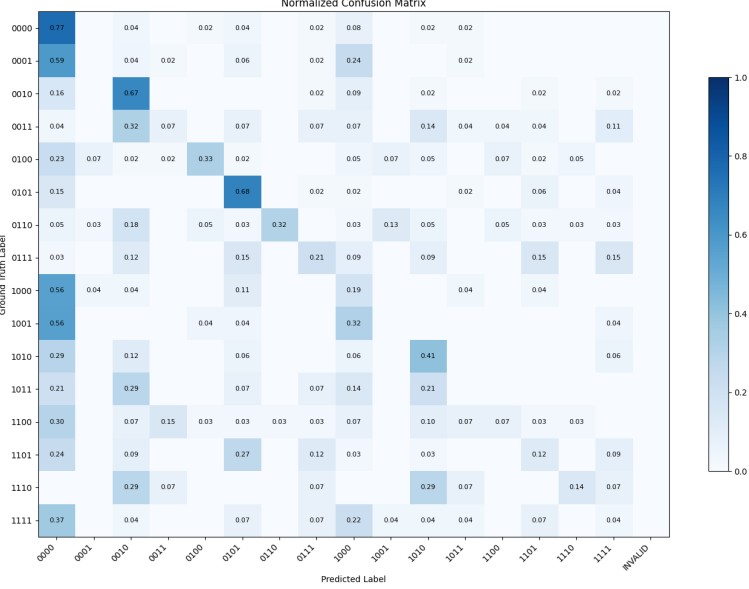

Figure 60: **Relation Type Confusion Matrix for Success (o3-mini PBEBench-Lite)**: Confusion matrix showing the distribution of relation types in the model prediction vs ground truth. INVALID category indicates cases where the model fails to generate a valid cascade at all. The results are for o3-mini on PBEBench-Lite, and success denotes Pass@1 = 1 with a sampling budget of 1.

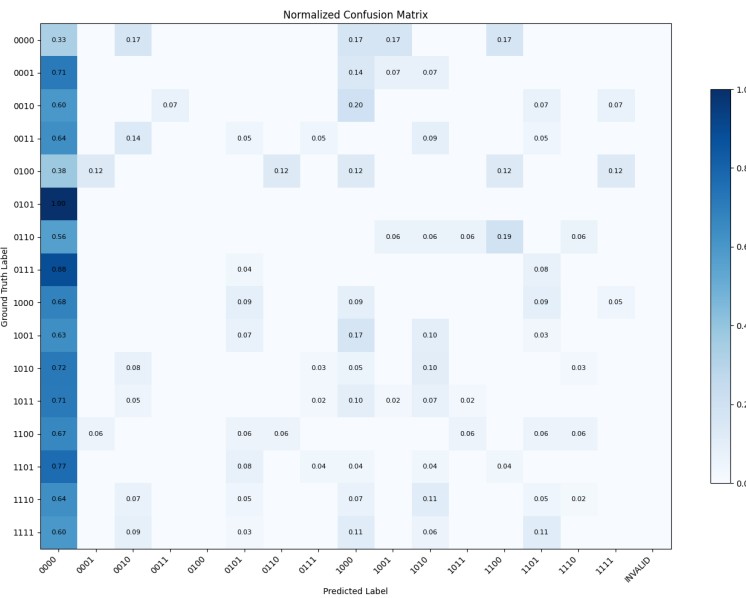

Figure 61: **Relation Type Confusion Matrix for Failure (o4-mini PBEBench-Lite)**: Confusion matrix showing the distribution of relation types in the model prediction vs ground truth. INVALID category indicates cases where the model fails to generate a valid cascade at all. The results are for o4-mini on PBEBench-Lite, and failure denotes Pass@1 = 0 with a sampling budget of 1.

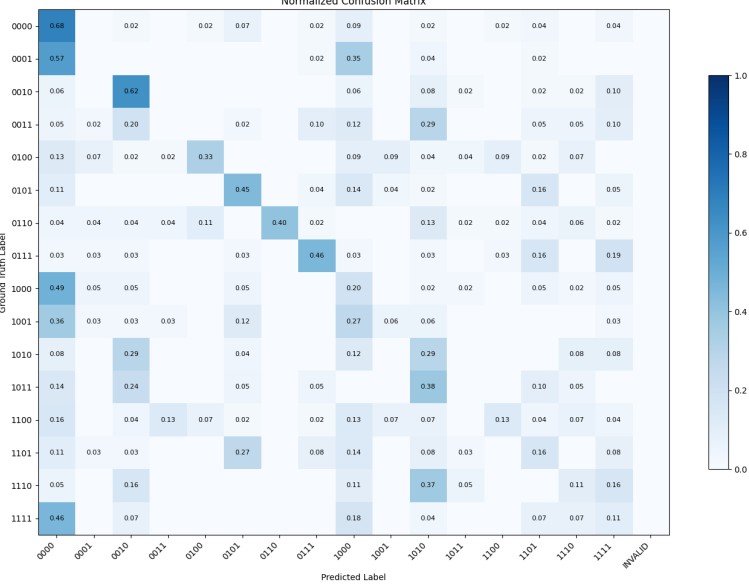

Figure 62: **Relation Type Confusion Matrix for Success (o4-mini PBEBench-Lite)**: Confusion matrix showing the distribution of relation types in the model prediction vs ground truth. INVALID category indicates cases where the model fails to generate a valid cascade at all. The results are for o4-mini on PBEBench-Lite, and success denotes Pass@1 = 1 with a sampling budget of 1.

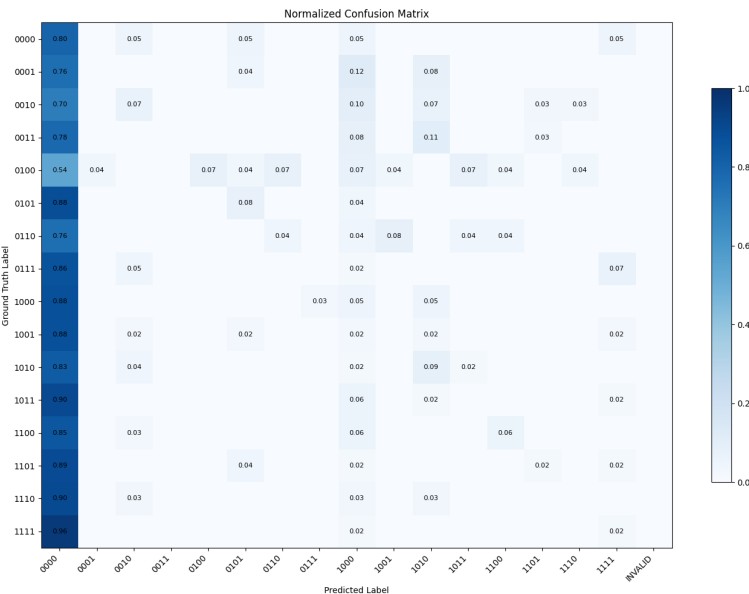

Figure 63: **Relation Type Confusion Matrix for Failure (gpt-oss-120b PBEBench-Lite)**: Confusion matrix showing the distribution of relation types in the model prediction vs ground truth. INVALID category indicates cases where the model fails to generate a valid cascade at all. The results are for gpt-oss-120b on PBEBench-Lite, and failure denotes Pass@1 = 0 with a sampling budget of 1.

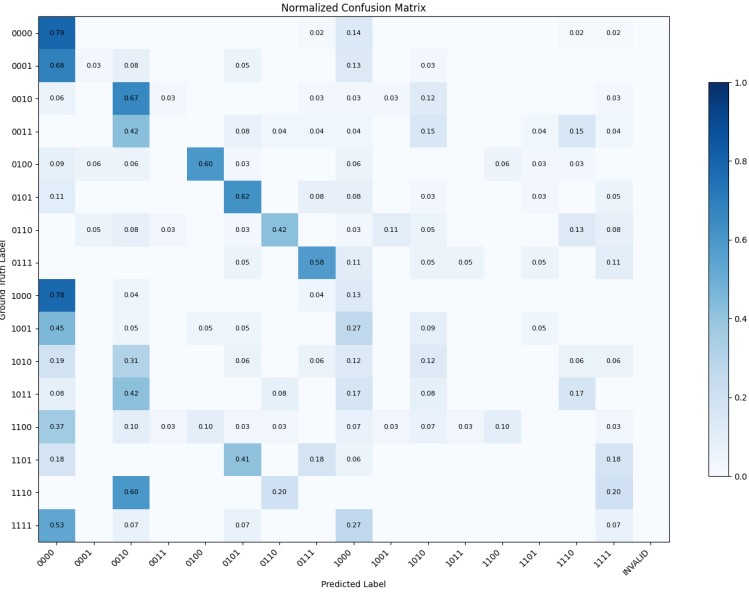

Figure 64: **Relation Type Confusion Matrix for Success (gpt-oss-120b PBEBench-Lite)**: Confusion matrix showing the distribution of relation types in the model prediction vs ground truth. INVALID category indicates cases where the model fails to generate a valid cascade at all. The results are for gpt-oss-120b on PBEBench-Lite, and success denotes Pass@1 = 1 with a sampling budget of 1.

