# OpenReview forum: "PBEBench: A Multi-Step Programming by Examples Reasoning Benchmark inspired by Historical Linguistics"
_ICLR.cc/2026/Conference — ICLR 2026 Conference Withdrawn Submission_

### Official Review · Reviewer_fMHW · 2025-10-24

**Soundness:** 3
**Presentation:** 2
**Contribution:** 3
**Rating:** 6
**Confidence:** 3

**Summary:**

This paper introduces PBEBench, a benchmark for multi‑step Programming by Examples (PBE) that evaluates inductive reasoning in LLMs by asking them to discover and order cascades of replace string‑rewrite rules. Difficulty is controlled by the ground‑truth cascade length. The authors construct two snapshots: PBEBench‑Lite (short cascades) and PBEBench (longer cascades). Experiments across open and closed models show strong gains from long chain‑of‑thought and test‑time compute, yet performance falls sharply as programs lengthen; even top models struggle beyond length 20–30.

**Strengths:**

* Many reasoning benchmarks emphasize deduction; PBEBench explicitly targets inductive program induction, filling a notable gap.
* The problem setup is unique. Bridging forward reconstruction and PBE with a minimal rewrite DSL is conceptually fresh and intellectually appealing.
* Comprehensive evaluation covers diverse LLMs (reasoning vs. non‑reasoning; open vs. closed), a wide range of difficulty levels PBEBench‑Lite through PBEBench with cascades up to 30, and multiple metrics (Pass@1, Edit_Sim, Valid_Rate).

**Weaknesses:**

* The overall setup is not immediately understandable to newcomers. In Figure 1, it would help to state explicitly what the problem proposer takes as input/outputs and what the LLM solver receives/returns.

**Questions:**

* Is the correct solution unique? Can uniqueness of the ground‑truth program cascade be guaranteed—or at least characterized—so that models are not rewarded for producing a different but functionally equivalent cascade?
* Is there a principled “algorithmic” solution? Beyond LLMs, is there a canonical synthesis/search procedure that should solve these instances efficiently? If not, it should be unreasonable to expect LLMs to solve these problems.
* Relatedly, what do the strongest LLMs actually do in their chains of thought—did you observe consistent internal strategies?

---

### Official Review · Reviewer_j3S2 · 2025-10-25

**Soundness:** 3
**Presentation:** 2
**Contribution:** 1
**Rating:** 2
**Confidence:** 4

**Summary:**

The authors introduce PBEBench, a reasoning benchmark where an input string is transformed through a sequence of simple programmatic rewrites. The task for models is to generalize the underlying transformation patterns used to reorganize the string. The benchmark is supported by a generation framework that automatically produces such problems with adjustable complexity. Through extensive evaluation, the authors show that reasoning models achieve stronger performance on these tasks. Also they find that as string length and the number of transformation steps increase, LLMs become notably less reliable in solving them.

**Strengths:**

- The dataset is timely and well-motivated, addressing the need for evaluating and improving inductive reasoning in LLMs.

- The proposed benchmark effectively enables scalable control over reasoning task complexity.

- The writing is clear, and the paper is well structured overall.

**Weaknesses:**

- The overall task is rather narrow in scope, raising questions about its usefulness for downstream applications.

- The lack of experiments showing whether training on PBEBench improves inductive reasoning capabilities within their dataset and downstream strongly limits the significance of the contribution.

- While the paper is well organized and generally clear, Section 3 is very dense and hard to follow. More illustrative examples would help to guide the reader through your paper.

**Questions:**

- Consider fine-tuning a recent LLM (e.g., Qwen/Qwen3-4B-Instruct-2507) on PBEBench. Report how far it can be pushed on your benchmark and contrast it to a recent reasoning model of similar size (I do not expect you to outperform it).

- Downstream inductive reasoning: Evaluate whether training on PBEBench transfers to inductive reasoning capabilities downstream (CLUTRR[1] and SLR[2]).

- Downstream reasoning: To show that capabilities are robust, evaluate that you are able to maintain performance on general LLM reasoning benchmarks, e.g., MMLU[3]

- Add inductive reasoning benchmarks to related work. CLUTRR[1], SLR[2] focus solely on inductive reasoning, while bAbI[4], KOR-Bench[5] do focus on it partially.

- Add a running example in Section 3 to guide the reader through your section.

- In Table 1, the current layout suggests that Claude and OpenAI models are not MoE models, which is likely inaccurate. Consider using an “unknown” tag instead.

- Explain why you evaluated both the first and last code blocks, and what insights these comparisons provide.

- The concepts of feeding and bleeding relationships are core concepts of your work. A brief explanation would help readers unfamiliar with historical linguistics follow your paper.

- Figure 4a: Update the figure title — you likely mean “max completion tokens”.

- The statement “Counterintuitively, more examples reduce performance” is debatable. I would not say that this is necessarily counterintuitive. Long-context tasks often lead to performance degradation in LLMs.

[1] CLUTRR — Sinha et al., Compositional Language Understanding and Text-based Relational Reasoning (EMNLP 2019).
[2] SLR-Bench — Helff et al., SLR: Automated Synthesis for Scalable Logical Reasoning (NeurIPS Wokshop FORLM 2025).
[3] MMLU - Hendrycks et al., Measuring Massive Multitask Language Understanding (ICLR 2021)
[4] bAbI — Weston et. al., "Towards AI-Complete Question Answering: A Set of Prerequisite Toy Tasks" (2015)
[5] KOR-Bench — Ma et al., KOR-Bench: Benchmarking Language Models on Knowledge-Orthogonal Reasoning Tasks (ICLR 2023).

---

### Official Review · Reviewer_p5h3 · 2025-10-28

**Soundness:** 2
**Presentation:** 2
**Contribution:** 2
**Rating:** 4
**Confidence:** 3

**Summary:**

This work introduces a controlled benchmark for evaluating LLM reasoning, built from synthetic data generated through randomly composed chains of simple text replacement functions. The task involves inferring the underlying generative function chain from a given set of input–output string pairs. Constructed automatically from predefined functions, the benchmark addresses common issues in existing reasoning benchmarks, such as data contamination, dependence on domain knowledge, and ambiguous complexity definitions. It is further divided into two subsets by difficulty: PBEBench-Lite (easier) and PBEBench (harder). Zero-shot evaluations across a range of open- and closed-source reasoning models demonstrate that explicit chain-of-thought (CoT) reasoning improves program induction performance, and that increasing reasoning length yields consistent performance gains.

**Strengths:**

1.	The formulation of the proposed benchmark and the corresponding data generation process is structured and well-grounded in formal definitions.
2.	An alternative complexity measure based on the relation types between rule pairs presents an interesting approach, as it captures categorically different forms of compositionality. This diverges from the conventional practice of using composition length (e.g., cascade length) as the sole indicator of complexity.
3.	The experiments cover a broad range of models, including both closed- and open-source, as well as reasoning-oriented (“thinking”) and non-reasoning models, which enhances the validity and generality of the findings.

**Weaknesses:**

1. I find the benchmark’s novelty as an evaluation measure for LLM reasoning somewhat unclear. The five distinctions listed in Section 1 appear conceptually overlapping with existing work or not fully substantiated in their current form:

   a. **Domain-agnostic design**
      It is described as domain-agnostic and independent of non-trivial domain knowledge.
      However, similar synthetic, domain-neutral reasoning benchmarks have been explored in prior work (e.g., [1,2,3,4]), with [2] also cited in this paper.

   b. **Complexity emerging from simple framework**
      Its complexity and difficulty are said to emerge from a simple framework grounded in provable logic.
      This is a sound and elegant design choice, though also a characteristic shared by several earlier benchmarks, including [1].

   c. **Four fundamental reasoning challenges**
      The paper highlights four fundamental reasoning challenges emerging from this framework.
      Yet it is also noted that conventional compositional measures, such as cascade length, have a stronger influence on model performance than the proposed categorical complexity metric, which may weaken the intended distinction.

   d. **Resistance to data contamination and saturation**
      The benchmark is presented as immune to data contamination and resistant to saturation due to the ease of generating new instances with controllable difficulty.
      This is indeed an advantage, though one that is generally shared among synthetic benchmarks.

   e. **Evaluation focus**
      Finally, the benchmark is said to evaluate models on practically and scientifically meaningful tasks rather than human–model comparisons.
      This framing is interesting, though it may appear somewhat at odds with the first claim, since domain-agnostic design typically limits immediate practical relevance.

2. If I understand correctly, the randomized data generation process does not include a mechanism to guarantee that each problem instance has a single correct program. For example, given the Input/Output pairs [ab→ax, ac→ac], multiple valid programs could satisfy them (e.g., [replace(b, x)] or [replace(ab, ax)]). Since each instance includes only a small number of Input/Output pairs (n = 5), such ambiguity is likely to be prevalent throughout the benchmark.

3. Extending from point 2, I suspect that for longer cascade lengths (L = 20), as shown in Figure 2, n = 50 Input/Output pairs may still be insufficient to constrain a single correct program. If the number of valid programs increases with cascade length, this could act as a confounding factor in the complexity evaluation and weaken the validity of the explanation provided in the paper.

4. From Figures 3b and 4b, PBEBench (supposedly the more challenging subset) appears to be nearly saturated (≥90% accuracy) for both open-source and closed-source models. This raises a concern about whether the benchmark still provides meaningful differentiation between models at the upper performance range, or if its difficulty needs to be recalibrated to better capture model-level differences.

5. The abstract and introduction are currently quite confusing. They focus heavily on historical linguistics, which appears to serve only as a motivational background rather than a substantive basis for the paper’s main goal of benchmarking LLM reasoning. In addition, many key concepts are introduced early without sufficient explanation, which hinders readability. I suggest restructuring the introduction to more clearly separate the motivational context from the core technical contributions, and to provide concise definitions or intuitive explanations of essential concepts upfront.

6. This is a minor suggestion and does not affect my rating. I see that a definition of rule pair relations (e.g., *Feeding*, *Bleeding*, etc.) is provided in Section 3.1 (page 5), but this comes too late, as the concepts are referenced multiple times beforehand. This disrupts the paper’s readability. I suggest moving at least part of the explanation from Appendix C (or an intuitive summary of these concepts) to an earlier section of the main text. The ideas could also be introduced in a more accessible manner rather than relying solely on formal definitions; for example, *Feeding* could be described as cases where the output of the first rule influences the second rule, and so on.

7. Very minor point: there are a few formatting issues, such as misaligned column titles in Table 1. Also, the “max sequence length: 16384” and “sampling budgets: 32” indicators within the plots in Figures 3 and 4 have their positions swapped. The underscore (“_”) in line 467 should be replaced with a whitespace.

[1] Sun at el., 2025 (https://arxiv.org/pdf/2506.18880)

[2] Shojaee et al., 2025 (https://ml-site.cdn-apple.com/papers/the-illusion-of-thinking.pdf)

[3] Estermann et al., 2024 (https://arxiv.org/pdf/2407.00401)

[4] Valmeekam et al, 2023 (https://arxiv.org/pdf/2206.10498)

**Questions:**

- Regarding Weakness 2, is there some mechanism or filtering step in place to mitigate this issue; such as ensuring that each generated instance admits only one valid program or discarding ambiguous cases during dataset construction?
- Minor question: is Edit_Sim assigned a value of zero for invalid (non-parsable) cases? Since Edit_Sim can take negative values (as shown in Table 1), handling such invalid cases may be nontrivial. Simply excluding them from aggregation could also bias the results in favor of models that produce invalid outputs on more difficult problems.

---

### Official Review · Reviewer_d4wq · 2025-10-31

**Soundness:** 2
**Presentation:** 3
**Contribution:** 2
**Rating:** 4
**Confidence:** 3

**Summary:**

This paper presents PBEBench, a new benchmark designed to evaluate multi-step inductive reasoning in LLMs, via program synthesis tasks. These tasks are formulated as string rewrite cascades, and are generated programmatically, allowing control over both the complexity and ensuring the training data used for the LLMs are very unlikely to be contaminated.

The authors present two versions of the benchmark, a PBEBench-Lite for quicker evaluations and the full PBEBench for more complex longer chain reasoning.

**Strengths:**

- Generally I find the paper well written, with good structure, clear flow, and all steps are given in sufficient detail that I would be able to reproduce their results.

- I find the formulation of the benchmark novel - many datasets present reasoning tasks that require multiple stages, but this offers a completely synthetic task grounded in historic linguistics. This is aesthetically pleasing, and offers a conceptually different problem to automated maths / logic benchmarks

- The evaluations are fine grained - with metrics full fully correct solutions, and edit distance measures for partial solves.

- The authors find clear empirical evidence that the cascade length is the key factor that degrades performance, and that reasoning models out perform non-reasoning ones.

- The synthetic nature and controllable difficulty also opens an avenue for curriculum learning - possibly adding multi step reasoning.

**Weaknesses:**

- It is unclear how the dataset guarantees that each of the problems are genuinely solvable

- It is unclear if the models are getting the right answer by following the correct set of reasoning steps - it could be partially using correct substitutions and partly de-noising - an important and powerful feature, but not necessarily the property the authors are trying to evaluate here. I would like to see some measure of how well the task path is followed?

- There don’t seem to be baselines for non LLM methods?

- Selecting the best result among samples might over estimate real world applications?

**Questions:**

- There is no evaluation of the intermediate steps the model takes in the reasoning - would it be possible to add a study where you get the model to output the steps which unpick the path and evaluate how well the model is following the path?

- I don’t see any evidence that ensures that the chain of differences is solvable by induction? This is always a problem with generated synthetic / constructed languages (related field) - would it be possible to show some more examples and demonstrate the enough information remains after overlapping substitutions that the problem can be solved?

- Do the authors intend this benchmark to be used by researchers more broadly? And will the dataset be distributed publicly?

- The dataset size feels borderline too small - I appreciate the cost of evaluating these expensive reasoning models against a larger dataset - and as pointed out for some models only a portion of the data was used. How difficult would it be to generate a dataset of say 10x 100x size?

    - Then if the dataset is intended for a broader / wider use - a single team evaluating a single model against the larger dataset is much more attainable.

    - Related point - given the number of examples used do you feel the number of significant figures in results (e.g. Table 1) are justified?

- Would it be possible to add some baseline measures for non LLM methods? (e.g. DFS, or constraint solvers perhaps?

- Given the selecting best result might over estimate performance of a single run - would it be possible to frame this as a TTC option and conduct a small study to evaluate the improvement in the performance?

- My biggest question - suspect completely beyond the scope - but it would be great if the authors could comment. Can you imagine conducting an experiment where you use fine-tuning on multi-step reasoning data such as this, and use it to improve the reasoning ability of the model on other datasets?

    - Get a baseline performance on a reasoning dataset - probably a programming / mathmatical one

    - Baseline on PBEBench

    - Fine tune the model on a larger synthetic dataset generated using this method

    - Track the performance on PBEBench vs the core reasoning task and see if this template offers a reusable pattern to apply to other reasoning tasks?

---

### Note · Authors · 2025-12-23

**Comment:**

We thank the reviewers for their time and thoughtful, detailed feedback. We would like to reiterate that the primary focus of this work is scalable benchmarking of the multi-step inductive reasoning capabilities of Large Language Models, with downstream applications such as sound law induction as a motivating use case [1].

We are grateful for the constructive suggestions provided by the reviewers, including benchmarking on SLRBench [2] and CLUTRR [3] (reviewer j3S2), expanding the related work discussion (reviewer j3S2), comparing LLM-generated solutions against ground-truth solutions (reviewer d4wq), and describing the reasoning strategies employed by different LLMs (reviewer fMHW). We also appreciate the writing and presentation feedback from reviewers p5h3 and j3S2. We plan to incorporate these suggestions, along with additional experimental results, in future resubmissions of this work.

Finally, we would like to clarify that neither solution uniqueness nor exact recovery of a specific ground-truth solution is an objective of this work. In both historical linguistics and the programming-by-example literature more broadly [4], the goal is to synthesize any valid solution that is consistent with the provided examples or input-output pairs. While additional criteria such as solution complexity or generalization to unseen examples may be applied on top of this, the primary objective remains the synthesis of a valid, functionally correct solution.

— PBEBench Authors

[1] Naik, Atharva, et al. "Can large language models code like a linguist?: A case study in low resource sound law induction." arXiv preprint arXiv:2406.12725 (2024).
[2] Helff, Lukas, et al. "SLR: An Automated Synthesis Framework for Scalable Logical Reasoning." arXiv preprint arXiv:2506.15787 (2025).
[3] Sinha, Koustuv, et al. "CLUTRR: A diagnostic benchmark for inductive reasoning from text." arXiv preprint arXiv:1908.06177 (2019).
[4] Li, Wen-Ding, and Kevin Ellis. "Is programming by example solved by LLMs?." Advances in Neural Information Processing Systems 37 (2024): 44761-44790.

**Withdrawal Confirmation:**

I have read and agree with the venue's withdrawal policy on behalf of myself and my co-authors.